# Optical and Geometrical Properties of Cirrus Clouds in Amazonia Derived From 1-year of Ground-based Lidar Measurements

Diego A. Gouveia[1], Boris Barja[1,2,3], Henrique M. J. Barbosa[1], Patric Seifert[4], Holger Baars[4], Theotonio Pauliquevis[5], and Paulo Artaxo[1].

[1]Applied Physics Department. Institute of Physics, University of São Paulo, São Paulo, SP, Brazil.

[2]Atmospheric Optics Group of Camagüey. Meteorological Institute of Cuba, Cuba.

[3]Atmospheric Research Laboratory, University of Magallanes, Punta Arenas, Chile.

[4]Leibniz Institute for Tropospheric Research (TROPOS), Leipzig, Germany

[5]Department of Environmental Sciences, Federal University of São Paulo, Diadema, SP, Brazil.

*Correspondence to:* Boris Barja Gonzalez (bbarja@gmail.com)

**Abstract.** Cirrus clouds cover a large fraction of tropical latitudes and play an important role in Earth's radiation budget. Their optical properties, altitude, vertical and horizontal coverage control their radiative forcing, and hence detailed cirrus measurements at different geographical locations are of utmost importance. Studies reporting cirrus properties over tropical rain forests like the Amazon, however, are scarce. Studies with satellite profilers do not give information on the diurnal cycle, and the satellite imagers do not report on the cloud vertical structure. At the same time, ground-based lidar studies are restricted to a few case studies. In this paper, we derive the first comprehensive statistics of optical and geometrical properties of upper-tropospheric cirrus clouds in Amazonia. We used one year (July 2011 to June 2012) of ground-based lidar atmospheric observations north of Manaus, Brazil. This dataset was processed by an automatic cloud detection and optical properties retrieval algorithm. Upper-tropospheric cirrus clouds were observed more frequently than reported previously for tropical regions. The frequency of occurrence was found to be as high as 88 % during the wet season and not lower than 50 % during the dry season. The diurnal cycle shows a minimum around local noon and maximum during late afternoon, associated with the diurnal cycle of precipitation. The mean values of cirrus cloud top and base heights, cloud thickness and cloud optical depth were $14.3 \pm 1.9$ (std) km, $12.9 \pm 2.2$ km, $1.4 \pm 1.1$ km, and $0.25 \pm 0.46$, respectively. Cirrus clouds were found at temperatures down to $-90\,°C$. Frequently cirrus were observed within the TTL, which are likely associated to slow mesoscale uplifting or to the remnants of overshooting convection. The vertical distribution was not uniform, and thin and subvisible cirrus occurred more frequently closer to the tropopause. The mean lidar ratio was $23.3 \pm 8.0$ sr. However, for subvisible cirrus clouds a bimodal distribution with a secondary peak at about 44 sr was found suggesting a mixed composition. A dependence of the lidar ratio with cloud temperature (altitude) was not found, indicating that the clouds are vertically well mixed. The frequency of occurrence of cirrus clouds

classified as subvisible (τ < 0.03) were 41.6 %, whilst 37.8 % were thin cirrus (0.03 < τ < 0.3) and 20.5 %
opaque cirrus (τ > 0.3). Hence, in central Amazonia not only a high frequency of cirrus clouds occurs, but
also a large fraction of subvisible cirrus clouds. This high frequency of subvisible cirrus clouds may
contaminate aerosol optical depth measured by sun-photometers and satellite sensors to an unknown
extent.
**1.  Introduction**
Clouds cover on average about 50 % of the Earth's surface (Mace et al., 2007) and cirrus alone cover 16.7
% (Sassen et al., 2008), with higher fractions occurring in the Tropics (Sassen et al., 2009). Hence cirrus
are important to understand current climate and to predict future climate (Wylie et al. 2005, Stubenrauch
et al. 2006; Nazaryan et al., 2008). Several studies emphasize the important role that cirrus clouds play in
the Earth's radiation budget (i.e. Liou 1986; Lynch et al. 2002; Yang et al. 2010a).  Their role is twofold.
First, cirrus clouds increase warming by trapping a portion of infrared radiation emitted by the
Earth/atmosphere system. Second, cirrus clouds cool the atmosphere by reflecting part of the incoming
solar radiation back into space. The contribution of each effect and the net effect on the radiative forcing
depends strongly on cirrus cloud optical properties, altitude, vertical and horizontal coverage (Liou 1986,
Kienast-Sjögren et al. 2016). Therefore, understanding their properties is critical to determining their
impact on planetary albedo and greenhouse effects (Barja and Antuña, 2011, Boucher et al., 2013). Also,
tropical cirrus clouds could influence the vertical distribution of radiative heating in the tropical
tropopause layer (e.g., Yang et al., 2010b; Lin et al., 2013). Noticeably, it has been shown that an accurate
representation of cirrus vertical structure in cloud radiative studies improves the results of these
calculations (Khvorostyanov and Sassen, 2002; Hogan and Kew, 2005; Barja and Antuña, 2011). Recent
research also shows that an increase of stratospheric water vapor is linked mainly to the occurrence of
cirrus clouds in the tropical tropopause layer (TTL) (Randel and Jensen, 2013). Finally, measurements of
the properties of cirrus clouds at different geographical locations are of utmost importance, potentially
allowing for improvements in numerical model parameterizations and, thus, reducing the uncertainties in
climatic studies.
Ground-based lidars are an indispensable tool for monitoring cirrus clouds, particularly those cirrus
clouds with very low optical depth, which are undetectable for cloud radars (Comstock et al., 2002) or for
passive instruments (e.g., Ackerman et al., 2008). For this reason, several studies with ground-based
lidars have reported the characteristics of cirrus clouds around the globe during the last decade. There are
some long-term studies reporting climatologies at midlatitudes (eg. Sassen and Campbell, 2001; Goldfarb
et al., 2001; Giannakaki et al., 2007; Hoareau et al., 2013; Kienast-Sjögren et al. 2016) and tropical
regions (eg. Comstock et al., 2002; Cadet et al., 2003; Antuña and Barja, 2006; Seifert et al., 2007;
Thorsen et al., 2011; Pandit et al., 2015). Table 1 shows an overview of these studies with different values
for cirrus clouds characteristics in diverse geographical regions.  There are also some short-term reports
on cirrus clouds characteristics during measurement campaigns at midlatitudes (e.g. Immler and Schrems,
2002a) and tropical latitudes (Immler and Schrems, 2002b, Pace et al., 2003 and references therein).
Additionally, satellite-based lidar measurements have been used to investigate the global distribution of
cirrus characteristics (eg. Nazaryan et al., 2008; Sassen et al., 2009; Sassen et al., 2009; Wang and
Dessler 2012, Jian et al., 2015). Characteristics of tropical and subtropical cirrus clouds have similar
geometrical values and they occur at higher altitudes than those at midlatitudes. However, the frequencies
of occurrence of cirrus cloud types differ significantly between different locations.
Reports on cirrus cloud measurement over tropical rain forests like in Amazonia are scarce. Just a few
global studies with satellite instruments include these regions, and they do not provide information on the
diurnal cycle. There are also a few studies focused on deep convection in Amazonia that report cirrus
clouds (eg. Machado et al., 2002; Hong et al., 2005; Wendisch et al., 2016), but no lidar measurements
were used. Baars et al. (2012) focused on aerosol observations with a ground-based Raman lidar, and thus
report only one cirrus cloud case that was observed between 12 km and 16 km height on 11 September
2008 during an 11-month measurement period in 2008. Barbosa et al. (2014) describe a week of cirrus
cloud measurements performed from 30 August to 6 September 2011 during an intensive campaign for
calibration of the water vapor channel of the UV Raman lidar, which is also used in this study. Cirrus
clouds during that period were present in 60% of the measurements. Average base and top heights were
11.5 km and 13.4 km, respectively, and average maximum backscatter occurred at 12.8 km. Most of the
time, two layers of cirrus clouds were present.
From the above discussion, the importance of continuous and long-term observations of tropical cirrus
clouds is evident. In the present study, we use one year of ground-based lidar measurements (July 2011 to
June 2012) at Manaus, Brazil to investigate the seasonal and daily cycles of geometrical (cloud top and
base altitude) and optical (cloud optical depth and lidar ratio) properties of cirrus over a tropical rain
forest site. In section 2, a description of the Raman lidar system, dataset, processing algorithms and site
are given. The results and discussion are presented in section 3. We close this paper with concluding
remarks in section 4.
**2. Instrumentation, dataset and algorithms.**
**2.1. Site and instrument description**
The ACONVEX (Aerosols, Clouds, cONVection EXperiment) or T0e (nomenclature of the
GoAmazon2014/15 experiment, Martin et al. 2016) site is located up-wind from Manaus-AM, Brazil, at
2.89° S and 59.97° W, in the central part of the Amazon Forest, as shown in the satellite image of Figure
1. Atmospheric observations at this site began in 2011 with the objective to operate a combination of
several instruments for measuring atmospheric humidity, clouds and aerosols as well as processes which
lead to convective precipitation (Barbosa et al., 2014).
As with most tropical continental sites, the diurnal cycle of precipitation is strong with a late afternoon
peak (Adams et al., 2013). The precise definition of the climatological seasons varies among authors (e.g.
Machado et al., 2004, Arraut et al., 2012, Tanaka et al., 2014), however, deep convection is a
characteristic of the region all year.  For our site and period of study, we considered a wet (Jan-Apr),  dry
(Jun-Sep), and transition (Mar, Oct-Dec) season respectively. Convection is more active during the wet
season, when the intertropical convergence zone (ITCZ) influences the region. As the ITCZ moves
northward during the months of the dry month, convective activity decreases.
The lidar system (LR-102-U-400/HP, manufactured by Raymetrics Advanced Lidar Systems) operates in
the ultraviolet (UV) at 355 nm. Three channels detect the elastically backscattered light at 355 nm as well
as the Raman-scattered light of nitrogen (387 nm) and water vapor (408 nm), simultaneously in analog
and photon-counting modes. The system is tilted by 5° from the zenith to avoid specular reflection of
horizontally-oriented ice crystals (e.g., Westbrook et al., 2010). It is automatically operated 7 days a
week, only being closed between 11 am and 2 pm local time (LT is −4 UTC) to avoid the sun crossing the
field of view. Detailed information about the lidar system and its characterization are given by Barbosa et
al. (2014). To retrieve the particle backscatter and extinction profiles from the lidar signal, the
temperature and pressure profiles were obtained from radio soundings launched at 0 and 12 UTC from the
Ponta Pelada Airport, located 28.5 km to the South (3.14°S, 59.98°W) of the experimental site.
**2.2. Datasets**
The lidar dataset used in the present study comprises measurements recorded between July 2011 and June
2012, which were temporally averaged into 5-min profiles (3000 laser shoots at 10 Hz). A total of 36,597
profiles were analyzed corresponding roughly to 1/3 of the maximum possible number of profiles during
1 year.
For the long-term analysis, winds were obtained from the ERA Interim reanalysis (Dee et al., 2011) of
European Center for Midrange Weather Forecast (ECMWF) with spatial resolution of 0.75° and temporal
resolution of 6 h. The tropopause altitudes were calculated using ERA Interim temperature profiles
interpolated to the measurement time of each cirrus layer observation. We followed the definition of the
World Meteorological Organization (IMV WMO, 1966), i.e. *"the lowest level at which the lapse rate*
*decreases to 2 °C km$^{-1}$ or less, provided that the average lapse rate between this level and all higher*
*levels within 2 km does not exceed 2 °C km$^{-1}$"*. We further assumed the lapse rate to vary linearly with
pressure (McCalla, 1981), and the exact altitude where $\Gamma=2$ °C km$^{-1}$ (i.e. the tropopause) was found by
linearly interpolating between the closest available pressure levels. Precipitation was obtained from
TRMM (Tropical Rainfall Measuring Mission) version 7 product 3B42 (Huffman et al., 2007) with 0.25°
and 3 h of spatial and temporal resolution, respectively. Back trajectories were calculated using the
HYSPLIT model (Stein et al., 2015) forced by meteorological fields from the US National Oceanic and
Atmospheric Administration (NOAA) Global Data Assimilation System (GDAS), available at 0.5 degree
resolution.
**2.3. Cirrus cloud detection algorithm.**
We used an automatic algorithm for the detection of the cloud base, the cloud top and the maximum
backscattering heights, based on Barja and Aroche (2001). The algorithm is explained in detail in Barbosa
et al. (2014) and is in here only described briefly. Basically, it assumes a monotonically decreasing
intensity of the lidar signal with altitude in a clear atmosphere and searches for significant abrupt changes.
These abrupt changes are marked as a possible cloud base. Examining the signal noise, each true cloud
base is discriminated. Then, the lowest altitude above cloud base with signal lower than that at cloud base
and corresponding to a molecular gaseous atmosphere is determined as the cloud top. When more than
one layer is present in the same profile, and their top and base are separated more than 500 m, they are
considered as individual clouds.  Figure S.2 gives an example of cloud detection. Barbosa et al. (2014)
also provide information on the discrimination of false positives and the distinguishing of aerosols from
thin cloud layers. After obtaining the base, top and maximum backscatter heights, the corresponding
cloud boundary temperatures are obtained from the nearest radiosonde. A detected high cloud is classified
as a cirrus cloud if the cloud top temperature is lower than –37 $^{\circ}$C (Sassen and Campbell, 2001; Campbell
et al., 2015). These temperatures are typically found at about 10.5 km height over Amazonia.

### 2.4. Frequency of Occurrence and Sampling Issues

In a simplified manner, the frequency of occurrence would just be the ratio of the number of profiles with
cirrus clouds to the total number of profiles. However, while one might be sure when a cirrus cloud was
detected in a given profile, there is no certainty of its presence when the profile has a low signal-to-noise
ratio or when there is no measurement available. Sampling cirrus clouds with a ground-based profiling
instrument can be problematic, particularly for the calculation of the temporal frequency of occurrence,
due to the obscuration by lower clouds, or availability of measurements, which might introduce sampling
biases (Thorsen et al., 2011).
To avoid these sampling issues, we use an approach similar to the conditional sampling proposed by
Thorsen et al. (2011) and Protat et al. (2014). First, we recognize that the presence of cirrus clouds is
rather independent of low-level liquid water clouds that can fully attenuate the laser beam, and
independent of instrumental issues that might restrict measurement time. Hence, the best estimate of the
true frequency of occurrence is the ratio of the number of profiles with cirrus, by the number of profiles
where cirrus could have been detected.
These qualifying profiles are identified as follows. The noise in each clear-sky bin follows a Poisson
distribution and is evaluated as the square root of the signal. The signal-to-noise ratio (SNR) is defined as
the background corrected signal divided by the noise, similar to Heese et al. (2010). Profiles are selected
if a clear-sky SNR higher than 1.0 is found at 16 km, for 7.5 m vertical resolution. Note that this is not the
SNR of the cirrus cloud (cirrus – molecular / noise), which typically ranges from 6 to 36. The threshold
was obtained from a performance evaluation of the detection algorithm. Using simulations, we varied
cloud thickness (15 m to 4.5 km), cloud backscatter coefficient (1 to 10 Mm$^{-1}$ sr$^{-1}$) and SNR (1 to 50). We
found that our algorithm detects 99% of cirrus clouds with COD > 0.005. In other words, given typical
cirrus cloud optical depths, the threshold used implies a sufficiently high SNR at cloud top for applying
the transmittance method (described in section 2.5).
From analysis of the available profiles, 16,025 were found to satisfy these criteria (see Table 2). July,
August and September, the driest months, show the highest fraction of profiles with good SNR, while the
wettest months have the lowest fraction of lidar profiles with good SNR (see figure S.1). To avoid
introducing biases from the different sample sizes in different months, the frequency of occurrence for the
year is calculated as the average frequency of occurrence for each season. The frequency for each season,
in turn, is calculated from the frequency of each month. Finally, the frequency for each month is
calculated by averaging over the mean diurnal cycles (i.e. mean of hourly means), because there are more
profiles with good SNR during night compared to daytime.
### 2.5. Cloud Optical Depth, backscattering coefficient and lidar ratio
Attenuation of the lidar signal by cirrus clouds can be obtained using the ratio of the range-corrected
signal at the top and at the cloud base as described in Young (1995):
$\frac{S(z_t)}{S(z_b)} = \frac{\beta(z_t)}{\beta(z_b)} e^{-2\int_{z_b}^{z_t}\alpha_p(z')dz'} e^{-2\int_{z_b}^{z_t}\alpha_m(z')dz'}$ ,    (1)
where $z_b$ and $z_t$ are the base and top height of a cirrus layer, and $S(z) = P(z)z^2$ is the range corrected
signal. $\beta(z)$ and $\alpha(z)$ are the volumetric backscattering and extinction coefficients, respectively, and each
is the sum of a molecular (sub index m) and a particle (sub index p) contribution. Volumetric
backscattering and extinction profiles from molecules were derived following Bucholtz (1995). Assuming
a negligible aerosol contribution in the atmospheric layers just below and above the cirrus clouds (Young,
1995), we can express the transmittance factor of the lidar equation due to the cirrus layer, $T^{cirrus}$, as
$T^{cirrus} = e^{-2\int_{z_b}^{z_t}\alpha_p(z')dz'} = \frac{S(z_t)}{S(z_b)}\frac{\beta(z_b)}{\beta(z_t)} e^{2\int_{z_b}^{z_t}\alpha_m(z')dz'}$ ,    (2)
and the cirrus optical depth (for an example, see Figure S.2), $\tau^{cirrus}$, as
$\tau^{cirrus} = \int_{z_b}^{z_t}\alpha_p(z')dz' = -\frac{1}{2}\ln(T^{cirrus})$ .    (3)
The accuracy of this calculation depends mainly on the SNR at the cirrus cloud altitude. However, when
the lidar signal is completely attenuated by the cirrus cloud (i.e. the transmission factor approaches zero)
it is impossible to obtain the true values of the cirrus top altitude and optical depth. The retrievals, in these
cases called apparent values, are necessarily underestimated and were not included in our analysis (see
Table 2).
The backscattering coefficients of cirrus clouds were determined by the Fernald-Klett-Sasano method
(Fernald et al., 1972; Klett, 1981; Sasano and Nakane, 1984) for each 5-min averaged profile having
cloud and satisfying the conditions discussed in the previous section. For retrieving extinction, the Klett
method requires a predetermined value for the layer-mean lidar ratio (LR), which is the ratio between the
extinction and backscattering coefficients. Then, integrating the extinction coefficient from the cloud base
to cloud top, the cirrus cloud optical depth is obtained ($\tau_{Klett}^{cirrus}$). Following Chen et al. (2002), we
estimated the value of LR for every cloud profile by iterating over a range of values of LR and comparing
the values of $\tau_{Klett}^{cirrus}$ with the independent value of the cirrus optical depth obtained from the
transmittance method described above ($\tau^{cirrus}$). The cirrus mean lidar ratio is the one that minimizes the
residue: $R(S) = (\tau_{Klett}^{cirrus} - \tau^{cirrus})^2$. We use the approach of Chen et al. (2002) instead of the Raman
method (Ansman et al., 2002) because our instrument can only detect the Raman scattered light at
nitrogen during nighttime as Raman scattering is very weak compared to the elastic scattering. Moreover,
the Raman results are very noisy even during nighttime and, by analyzing simulated lidar profiles (not
shown), we found that for the given setup of our study (24/7 analysis of 5-min profiles) a more precise
and accurate cirrus layer-mean LR can be obtained with the Chen et al. (2002) method.
The Klett method assumes single scattering, but eventually the received photons could have been
scattered by other particles multiple times before reaching the telescope. This effect, named multiple
scattering, increases the apparent laser transmittance and decreases the corresponding extinction
coefficient values. Inversion of uncorrected signals could bias the extinction, and hence the COD and LR,
typically by 5-30% (Thorsen and Fu, 2015). This is particularly important at UV wavelengths, for which a

much stronger forward scattering and therefore larger amounts of multiple scattering occur compared to the visible or infrared wavelengths. For this reason, we refrain from applying empirical correction formulas (e.g. such as eq. 10 in Chen et al., 2002), and instead perform a full treatment of multiple scattering following the model of Hogan (2008). The correction is found iteratively, similar to Seifert et al. (2007) and Kienast-Sjögren et al. (2016). The forward model is initialized with the originally retrieved, uncorrected extinction profile, and the model output is used to correct the extinction profile iteratively, until it converges. In our case, we assumed the effective radius of ice crystals to vary with temperature according to a climatology of aircraft measurements of tropical cirrus data (Krämer et al., 2016a, 2016b), which includes the recent ACRIDICON field campaign with the German aircraft HALO in the Amazon region (Wendisch et al., 2016). The full treatment corrects the retrieved LR by about 40%, from $16.8 \pm 5.8 \, sr$ (uncorrected) to $23.6 \pm 8.1 \, sr$, while Chen's approach would only correct it to $20.2 \pm 7.0 \, sr$. In the following sections, all cirrus optical properties (lidar ratio, extinction coefficient, and optical depth) derived in the frame of this study were corrected for multiple-scattering.

## 3. Results and discussion.

### 3.1. Frequency of cirrus cloud occurrence.

A total of 11,252 lidar profiles were recorded with the presence of cirrus clouds, yielding an average temporal frequency of cirrus cloud occurrence of 73.8 % from July 2011 to June 2012. Figure 2 shows the monthly frequency of cirrus cloud occurrence, with statistical error, and precipitation in central Amazonia. There is a well-defined seasonal cycle, with maximum values from November to April, reaching 88.1 % during the wet season, and a minimum value in August during the dry season (59.2 %), but with frequencies not lower than 50 % (see Table 2). Moreover, the mean monthly cirrus cloud frequency follows the same seasonal cycle as accumulated precipitation, which responds to the seasonal changes of the ITCZ, and is higher from January to April and lower from June to September (Machado et al., 2002; 2014). Mean cirrus frequencies during the wet months are higher by a statistically significant amount than during dry months (notice the small standard deviation of the mean despite the high variability). This result and the lack of the other possible formation mechanisms proposed in the literature (Sassen et al., 2002) suggest that deep convection is the main formation mechanism for cirrus clouds in central Amazonia. Deep convective clouds generate cirrus clouds when winds in the upper troposphere remove ice crystals of the top of the large convective column, generating anvil clouds. Anvil clouds remain even after the deep convective cloud dissipates and persists from 0.5 to 3.0 days (Seifert et al, 2007).

To further investigate the role of deep convection as the main local formation mechanism, the high-altitude circulation and spatial distribution of precipitation were studied. The mean wind field at 150 hPa, approximately the mean cirrus top-cloud altitude (14.3 km, see Table 3), and accumulated precipitation are shown in Figure 3. The study period was divided into wet (January, February, March and April), dry (June, July, August and September) and transition (May, October, November and December) periods, based on accumulated precipitation. During the wet months, the South American monsoon is prevalent, and associated rain amounts range from 8 to 14 mm/day, with monthly totals of about 300 mm. Winds at

150 hPa blow from the southeast at about 6 m/s. During the dry period, convective activity moved to the
north toward Colombia and Venezuela and the 150 hPa air flow is from the west, also at about 6 m/s, thus
allowing cirrus clouds to be advected by 520 km or 4.5° per day. As previous studies reported that
tropical cirrus could be transported by thousands of kilometers (e.g. Fortuin et al., 2007), 24-h back-
trajectories were calculated to investigate the possible origin of the observed clouds. These are shown in
the right panels of Figure 3, where one trajectory was calculated for each cirrus layer detected, with the
arrival height set to the height of top of the cirrus layer. Most of the trajectories are directed to the regions
of maximum accumulated precipitation (left panel), which are much closer to the site during the wet (~
5°) than dry (~ 10°) season. This gives further evidence that cirrus clouds observed in central Amazonia
are likeliest detrained anvils from tropical deep convection.
The backward trajectories also reveal that the high-altitude circulation is quite variable. Indeed, many
backward trajectories do not follow the average wind pattern and seem to point in the opposite direction
of precipitation, particularly during the dry season. One should note, however, that central Amazonia still
receives about 100 mm per month of precipitation in the dry season (reddish colors around the site, Figure
3) and most of it comes from mesoscale convective systems (Machado et al., 2004; Burleyson et al.,
2016). Hence, during the dry season, there is a mixture of locally produced and long-range transported
cirrus, in contrast to the wet season when there is always near-by convection.
The diurnal cycle of cirrus cloud frequency, shown in Figure 4, also has a close relation with the
convective cycle. The frequency of occurrence, for the overall period or any season, exhibits a minimum
between 10 and 14 hours local time (LT). Maximum values are found between 17 and 18 LT, in the late
afternoon, when values are slightly higher than in the morning. This diurnal variation follows the diurnal
cycle of convection documented in the literature (e.g. Machado et al., 2002; Silva et al., 2011, Adams et
al. 2013), as also shown in Figure 4 as the diurnal cycle of precipitation averaged over an area of 2° x 2°
centered on the experimental site. Maximum precipitation occurs between 13 and 18 LT, during both the
dry and the wet seasons, which coincides with the increase in cirrus frequency. In Figure 4, a smaller
amplitude in cirrus frequency during the wet season versus the dry season months is seen. This can be
reconciled by analyzing the maximum precipitation rates and the upper-altitude circulation (see Figure 3).
When the frequency of deep convection is greater (3 times more in the wet season) and closer to the site
(~5° in the wet and ~10° in the dry), the cirrus clouds, which are long-lived, presumably get more evenly
distributed during the day.
To verify that the lower cirrus cloud cover around noon was not related to a decrease in SNR and, hence,
a decrease in detection efficiency, we analyzed the frequency of occurrence for different cirrus types
(following Sassen and Cho, 1992). Opaque (COD > 0.3), thin (0.3 > COD > 0.03) and sub-visual cirrus
(SVC) clouds (COD < 0.03) were considered. Their diurnal variation is also shown in Figure 4. The
frequency of occurrence of opaque cirrus has the larger amplitude, during both dry and wet seasons.
During the dry (wet) season, it increases from less than 5 % (20 %) to about 30 % (50 %) in the hours
following the precipitation maximum, 15 h to 19 h LT. The second larger diurnal variation corresponds to
the occurrence frequency of thin cirrus, which decreases after the sunrise from 30 % (50 %) to 20 % (30
%) during the dry (wet) season, and increase again during night time, when the opaque cirrus clouds are
dissipating. The SVC, whose detection could be biased by lower SNR, do not show a clear diurnal cycle.
Hence, the diurnal cycle of the frequency of occurrence of cirrus clouds in central Amazonia is likely a
result of the diurnal cycles of opaque and thin cirrus, which have a sufficiently high COD to not be
missed by the detection algorithm.

**3.2. Geometrical, optical and microphysical properties of cirrus clouds.**

Table 2 shows column-integrated statistics of the properties of cirrus clouds during the one-year
observational period, also distinguished by season. Column-integrated COD varies from $0.25 \pm 0.45$ in
the dry season to $0.47 \pm 0.65$ in the wet season. The frequency of occurrence of opaque, thin and SVC
column-integrated COD is 11.8 % (31.3 %), 23.9 % (37.9 %) and 23.3 % (18.3 %) respectively in the dry
(wet) season. The maximum backscattering altitude does not show a seasonal cycle, and is on average
$13.4 \pm 2.0$ km (or $-60 \pm 15$ $^{o}$C). The average number of simultaneous layers of cirrus present in each
cloudy profile is 1.4 (1.25 during the dry, and 1.62 during the wet season), and hence geometrical
properties, in a column-integrated sense, are not discussed.
As cirrus at different altitudes might have different origins or microphysical properties, it is more
important to analyze the statistics based on each layer detected, as shown in Table 3. The overall mean
value for the cloud layer base altitude is $12.9 \pm 2.2$ km, for the cloud layer top altitude, $14.3 \pm 1.9$ km, and
for the cloud layer geometrical thickness, $1.4 \pm 1.1$ km. The mean value of the cloud layer maximum
backscattering altitude is $13.6 \pm 2.0$ km. The differences between the mean values of the geometrical
properties in the dry and wet seasons are not statistically significant, except for the thickness, which
changes from 1.3 km to 1.5 km, respectively. These values are similar to those reported by Seifert et al.
(2007) for the Maldives (4.1 °N, 73.3 °E): $11.9 \pm 1.6$ km (base), $13.7 \pm 1.4$ km (top), $1.8 \pm 1.0$ km
(thickness), $12.8 \pm 1.4$ km (max. backscatter) and $-58 \pm 11$ °C (temperature at max. backscatter). Reports
from subtropical regions also show similar values. Cadet et al. (2003) report for the Reunion Island (21°S,
55°E) cirrus cloud base and top altitudes of 11 km and 14 km, respectively. Antuña and Barja (2006)
report for a subtropical experimental site (Camagüey, Cuba, 21.4° N, 77.9° W) cirrus cloud base and top
altitudes of 11.63 km and 13.77 km, respectively. On the other hand, Sassen and Campbell (2001) show
mean values for midlatitude cirrus cloud base and top of 8.79 km and 11.2 km, respectively, which is
lower than for tropical cirrus, and an average geometrical thickness of 1.81 km. Some cirrus cloud
characteristics reported around the globe are shown in Table 1 for comparison.
The geometrical characteristics of the detected cirrus clouds were further examined by means of
normalized histograms. Figure 5 shows the results for cloud base and top height, thickness and cloud
optical depth. Histograms for the wet and dry season months reveal differences. The cloud base
distribution (Figure 5a) is wider during the wet season. There are relatively more cirrus layers with cloud
base below 12 km and above 16.5 km during the wet than during the dry season. Particularly, there is a
peak centered at 16.5 km during wet months, which does not exist during the dry season months. The
distribution of geometrical thickness (Figure 5b) shows more cirrus layers thicker than 2 km (and less
thinner than that) in the wet season. The normalized histogram of COD (Figure 5d) shows relatively more
cirrus layers with COD > 0.1 in the wet season, and more with COD < 0.1 in the dry season. The largest
differences, however, are seen in the cirrus cloud top altitude distribution (figure 5c). It shows two peaks
in the wet months, one centered at 14.25 km and second centered at 17.75 km. On the other hand, for dry

months, there is only one peak centered at 15.75 km. These differences suggest different cirrus types with different origins.

Comstock et al. (2002) proposed two different types of cirrus clouds at Nauru Island in the tropical western Pacific with oceanic conditions: one type (laminar thin cirrus) with cloud base altitudes above 15 km and the other (geometrically thicker and more structured cirrus) with base altitudes below this height, with different characteristics. Liu and Zipser (2005) used TRMM Precipitation Radar (PR) dataset to trace the deep convection and precipitation throughout the tropical zone, including oceans and continents. The authors showed that only 1.38 % and 0.1 % of tropical convective systems, and consequently their generated cirrus clouds, reached 14 km and 16.8 km of altitude, respectively.

Considering these previous results, we suggest that the highest peak in wet months in cloud top distribution originates from convection penetrating the tropopause, located at about 15.9-16.5 km, while the lowest peak is the ceiling of most tropical convection. The single peak observed during the dry months, in turn, originates from cirrus clouds transported by large distances. Clouds generated by convective systems can persist in the atmosphere from hours to days if they are slowly lifted (Ackerman et al., 1988; Seifert et al., 2007). Clouds that ascended and are horizontally transported by long distances are, in general, optically and geometrically thinner and found at higher altitudes in the troposphere. This also explains why the geometrical thicknesses and optical depth are lower during the dry season months.

To investigate if the higher cirrus layers were indeed geometrically and optically thinner, a more in-depth analysis of the vertical distribution was performed. Figure 6 shows two-dimensional histograms of cloud optical depth and cirrus occurrence vertical distribution for the wet season months (top) and dry season months (bottom). The right panels show the vertical distribution of the frequency of occurrence for the three cirrus categories. During the wet months, there is more dispersion (wider range of COD for a fixed altitude, and vice-versa) than in the dry months, which we speculate might be associated with the well-documented variability in the intensity of deep convection in Amazonia (Machado et al., 2002; Adams et al., 2009, 2013, 2015). Indeed, it is only during the wet season that a significant fraction of cirrus is found above 16 km height, and they have a COD ranging from 0.001 to 0.02. Moreover, while the distribution of opaque cirrus peaks at 12 km height in both seasons, thin cirrus and SVC shows a bimodal distribution only in the wet season, with the highest maxima above 14 km and 16 km respectively. This is presumably associated with the overshooting convection discussed above, which occurs mostly during the wet season (Liu and Zipser, 2005). Moreover, ice detrainment directly into the tropical tropopause layer (TTL) is one of the main mechanisms of TTL cirrus formation; the other is in-situ formation by supersaturation promoted by mesoscale uplift (Cziczo et al., 2013), which can occur above tropical convective systems (Garret et al., 2004), a very common feature of the Amazon hydrological cycle.

To investigate the role of the tropopause capping on the cirrus vertical development, its altitude was calculated from the ERA Interim dataset for the observation time of each cirrus profiles (see section 2.2 and Figures S.3a and S.3b). The tropopause mean altitudes during the wet, transition and dry periods are $16.5 \pm 0.2$ km, $16.3 \pm 0.3$ and $15.9 \pm 0.4$, respectively. Therefore, a non-negligible fraction of the observed cirrus during the wet and dry seasons (Figure 6) occurred likely above the tropopause. Figure 7 shows the distribution of the distance from the cloud top and bottom to the tropopause. About 7 % (19 %) of the detected cirrus clouds have their cloud base (top) above the tropopause during the wet season, and

5 % (13 %) during the dry season. Most of the cirrus cloud tops are found right below the tropopause
inversion, except during the wet season when they are uniformly distributed from -2 km to +0.5 km,
which is associated with the variability in deep convection intensity as discussed above. During the dry
season, on the other hand, deep convection overshooting occurs primarily north of the equator (Figure 2
from Liu and Zipser, 2005). These cirrus that form around the tropopause cannot last for a long time
(typically less than a day; Jensen et al., 1996), as they cannot be lifted above the tropopause inversion.
Therefore, they cannot be transported over long distances and do not reach the measurement site, hence
there is only one maximum near 15 km in the distribution of cloud tops, which is just below the
tropopause.
The classification of cirrus clouds following Sassen and Cho (1992) shows that 41.6 % of the cirrus
clouds measured in our experimental site are subvisible ($\tau < 0.03$), 37.8 % are thin cirrus ($0.03 < \tau < 0.3$)
and 20.5 % are opaque cirrus ($\tau > 0.3$). Table 3 shows these values for each season. SVC clouds have the
highest (lower) fraction during dry (wet) months. Opaque clouds have the highest (lowest) fraction during
wet (dry) months, which is expected, as there is a predominance of newly-generated clouds by deep
convection.
This large fraction of optically-thin and subvisible cirrus clouds over Amazonia present a challenge for
using passive remote sensing from space, such as MODIS. As mentioned by Ackerman et al. (2010), thin
cirrus clouds are difficult to detect because of insufficient contrast with the surface radiance. MODIS only
detects cirrus with optical depth typically higher than 0.2 (Ackerman et al., 2008). Therefore, the
MODIS's cloud-mask does not include 71 % of cirrus clouds over Amazonia, and likewise, their
estimation of aerosol optical depth might be contaminated with these thin cirrus. Aerosol optical depth
measurements from AERONET can also be contaminated with thin cirrus clouds. Chew et al. (2011), for
instance, estimated that the fraction of contaminated measurements of AERONET AOD in Singapore
(1.5° N, 103.7° E) is about 0.034 to 0.060. The determination of the actual contamination of MODIS and
AERONET aerosol products for Amazonia by thin cirrus will be the subject of a forthcoming study.
The different types of cirrus clouds measured in central Amazonia, with different formation mechanisms,
optical depths and altitude ranges are expect to be composed of ice crystals of different shapes. One way
to gain information on the crystal habits is to compute the lidar-ratio (Sassen et al., 1989). As explained in
section 2, we are able to estimate the average lidar ratio for the detected cirrus cloud layers in each profile
using an interactive approach instead of explicitly calculating the extinction from the Raman signal,
which would be available only during night-time.
Average values are given in Table 3 for all cirrus, and for each category. A mean value of $23.9 \pm 8.0$ (std)
sr was obtained for the whole period and the variation is less than 1.5 sr for the different seasons (i.e., it
does not show a seasonal cycle). For opaque, thin and SV cirrus the means are $25.7 \pm 6.3$ sr, $22.8 \pm 7.9$ sr
and $21.6 \pm 8.4$ sr, respectively. Pace et al., (2003) found a mean value of lidar ratio of 19.6 sr for the
tropical site of Mahé, Seychelles. Seifert et al.(2007), also for tropical regions, report values close to
32 sr. Platt and Diley (1984) reported the value of 18.2 sr with an error of 20%. For the other latitudes,
examples are given in Table 1. We note, however, that the lidar ratio may vary greatly depending on the
altitude and composition of cirrus clouds (Goldfarb et al., 2001), but also on the correction for multiple
scattering (Platt, 1981; Hogan, 2008). The latter depends on the ice crystals effective radius, and the
associated uncertainty can range from 20 to 60 % (Wandinger, 1998).
Although the mean LR for all seasons and categories are similar, their statistical distribution might yet
reveal differences. Figure 8 shows the histograms of lidar ratio corrected for multiple-scattering for the
different seasons (top) and for the different categories (bottom). For all seasons, the most frequent lidar
ratios are between 18 sr and 28 sr. There are notable differences only for different cirrus categories. The
opaque cirrus distribution has a peak at 25 sr, while thin cirrus has its peak at about 21 sr, and SVC at
about 15 sr, with a secondary peak at 44 sr.
As cirrus microphysical properties are expected to depend on altitude (e.g., Goldfarb et al., 2001), we
examine the dependence of the lidar ratios with the cirrus cloud top temperature (Figure 9). The plots
show the mean, the median, and the interquartile distance. A slight increase in the lidar ratio values from
20 sr to 28 sr for a decrease in temperature from -40 to -55 °C can be noticed during the dry period.
During the wet period, the lidar ratio values are between 18 sr and 28 sr in all temperature intervals.
Seifert et al. (2007) and Pace et al. (2003) both show the same temperature dependence of the lidar ratio,
but with different mean values of the lidar ratio. This behavior is an indication of a slight variation in the
microphysical characteristics of the observed clouds.
**4. Conclusions.**
One year of ground-based lidar measurements collected between July 2011 and June 2012 were used to
investigate the geometrical and optical properties of cirrus clouds in central Amazonia. An algorithm was
developed to search through this dataset with high vertical and temporal resolution and to automatically
find clouds, calculate particle backscatter, and derive optical depth and lidar ratio. The frequency of cirrus
cloud occurrence during the observation period was 73.8 %, which is higher than reported previously in
the literature for other tropical regions. Cirrus frequency reached 88.1 % during the wet months (January,
February, March and April), but decreased to 59.2 % during the dry months (June, July, August, and
September). Analysis of high-level circulation and precipitation during the wet months indicate that near-
by deep convection was likely the main source of these cirrus. Whilst during the dry period, there was a
mixture of locally produced and transported clouds. Moreover, we found that the diurnal cycle of the
frequency of occurrence of opaque and thin cirrus shows a minimum around 12h LT and a maximum
around 18h LT, following the diurnal cycle of the precipitation for both seasons.
The geometrical and optical characteristics of cirrus clouds measured in the present study were consistent
with other reports from tropical regions. The mean values were 12.9 ± 2.2 km (base), 14.3 ± 1.9 km (top),
1.4 ± 1.1 km (thickness), and 0.25 ± 0.46 (optical depth). Cirrus clouds were found at temperatures down
to −90 °C and maximum backscatter altitude was 13.6 ± 2.0.
By simultaneously analyzing cloud altitude and COD, it was found that cirrus clouds observed during the
dry season months are optically thinner and lower in altitude than those during the wet period. The
vertical distribution of frequency of occurrence is mono-modal, and 13 % of the observed cirrus had top
within the TTL. During the wet season months, there is a wider range of COD for a fixed altitude, and
vice-versa, which is associated with the variability in the intensity of deep convection in Amazonia. The
vertical distribution of the frequency of occurrence of the detected clouds shows a bimodal distribution
for thin and SV cirrus, and 19 % of the observed cirrus had top within the TTL, which are likely
associated to slow mesoscale uplifting or to the remnants of overshooting convection.
For the first time, the lidar ratio of cirrus clouds was obtained for the Amazon region. The mean lidar
ratio, corrected for multiple-scattering, was $23.6 \pm 8.1$ sr, in agreement with other reports from the
tropical regions. The statistical distribution of lidar ratios measured during the different seasons is the
same, and they also do not vary with temperature (altitude) of the clouds, indicating that these are well
mixed vertically. It was observed, however, that the distributions of the lidar ratio for different cirrus
categories are quite different. They are more skewed towards lower lidar ratios for smaller COD. From all
cirrus clouds observed, 41.6 % were classified as subvisible (COD < 0.03), 37.8 % as thin (0.03 < COD <
0.3) and 20.5 % as opaque (COD > 0.3). During the dry months, subvisible cirrus clouds reached a
maximum frequency of occurrence of 46 %, while opaque cirrus have their maximum during the wet
season months (25.2 %). These values are characteristic for the region under study and somewhat
different from other tropical regions. Thus, central Amazonia has a high frequency of cirrus clouds in
general, and a large fraction of subvisible cirrus clouds. Therefore, the aerosol optical depth determined
by Sun photometers and satellite based sensors in this region might be contaminated by the presence of
these thin clouds. Future work must be conducted in order to evaluate how large this contamination might
be over Amazonia.
**5. Acknowledgements**
We thank our colleague David K Adams from UNAM and two reviewers for reading the manuscript and
giving valuable comments. We thank Martina Krämer for sharing the aircraft data on tropical cirrus.
D.A.G. acknowledges the support of the CNPq fellowship program. B.B. acknowledges the financial
support of CAPES project A016_2013 on the program Science without Frontiers and the SAVERNET
project. H.M.J.B. and P.A. acknowledge the financial support from FAPESP Research Program on Global
Climate Change under research grants 2008/58100-1, 2009/15235-8, 2012/16100-1, 2013/50510-5, and
2013/05014-0. Maintenance and operation of the instruments at the experimental site would not have
been possible without the institutional support from EMBRAPA. We thank INPA, The Brazilian Institute
for Research in Amazonia, and the LBA Central office for logistical support. Special thanks to Marcelo
Rossi, Victor Souza and Jocivaldo Souza at Embrapa, and to Ruth Araujo, Roberta Souza, Bruno Takeshi
and Glauber Cirino from LBA. The authors gratefully acknowledge the NOAA Air Resources Laboratory
(ARL) for the provision of the HYSPLIT transport and dispersion model used in this publication.

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

Background Diurnal Cycle of Deep Convection around the GoAmazon2014/5 Field Campaign
Sites. *J. Appl. Meteor. Climatol.,* **55**, 1579–1598, doi: 10.1175/JAMC-D-15-0229.1, 2016
Campbell, J. R., Vaughan, M. A., Oo, M., Holz, R. E., Lewis, J. R., and Welton, E. J., Distinguishing
cirrus cloud presence in autonomous lidar measurements, Atmos. Meas. Tech., 8, 435-449,
doi:10.5194/amt-8-435-2015, 2015.
Campbell, J., S. Lolli, J. Lewis, Y. Gu, and E. Welton, Daytime Cirrus Cloud Top-of-the-Atmosphere
Radiative Forcing Properties at a Midlatitude Site and Their Global Consequences. J. Appl.
Meteor. Climatol., 55, 1667–1679, doi: 10.1175/JAMC-D-15-0217.1, 2016
Cadet, B., Goldfarb, L., Faduilhe, D., Baldy, S., Giraud, V., Keckhut, P., and Réchou, A., A sub-tropical
cirrus clouds climatology from Reunion Island (21°S, 55°E) lidar data set, Geophys. Res. Lett.,
30(3), 1130, doi:10.1029/2002GL016342, 2003.
Chen, W.; Chiang, C.; Nee, J.: Lidar ratio and depolarization ratio for cirrus clouds. Applied Optics, v.
41, n. 30, p. 6470 6476, 2002.
Chew B, Campbell J, Reid J, Giles D, Welton E, Salinas S, Liew S.: Tropical cirrus cloud contamination
in sun photometer data. Atmospheric Environment;45 (37):6724-6731, 2011.
Comstock, J. M., Ackerman, T. P., and Mace, G. G.: Ground-based lidar and radar remote sensing of
tropical cirrus clouds at Nauru Island: Cloud Statistics and radiative impacts, J. Geophys.
Res.,107, 4714, doi:10.1029/2002JD002203, 2002.
Cziczo, D. J., Froyd, K. D., Hoose, C., Jensen, E. J., Diao, M., Zondlo, M. A., Smith, J. B., Twohy, C. H.,
and Murphy, D. M.: Clarifying the dominant sources and mechanisms of cirrus cloud formation.,
Science, 340, 1320–1324, doi:10.1126/science.1234145, 2013.
Fernald, F. G., Herman, B. M. and Reagan, J. A.: Determination of aerosol height distribution by lidar,
Appl. Opt., 11, 482–489, 1972.
Fortuin, J. P. F., Becker, C. R., Fujiwara, M., Immler, F., H. M. Kelder, Scheele, M. P., and Schrems, O.,
Verver, G. H. L.: Origin and transport of tropical cirrus clouds observed over Paramaribo,
Suriname (5.8°N, 55.2°W), J. Geophys. Res., 112, D09107, doi:10.1029/2005JD006420, 2007.
Garrett, T. J., A. J. Heymsfield, M. J. McGill, B. A. Ridley, D. G. Baumgardner, T. P. Bui, and C. R.
Webster (2004), Convective generation of cirrus near the tropopause, J. Geophys. Res., 109,
D21203, doi:10.1029/2004JD004952.
Giannakaki, E., Balis, D. S., Amiridis, V., and Kazadzis, S.: Optical and geometrical characteristics of
cirrus clouds over a Southern European lidar station, Atmos. Chem. Phys., 7, 5519–5530,
doi:10.5194/acp-7-5519-2007, 2007.
Goldfarb, L., Keckhut, P., Chanin, M.-L., and Hauchecorne, A.: Cirrus climatological results from lidar
measurements at OHP (44° N, 6° E), Geophys. Res. Lett., 28, 1687–1690, 2001.
Hoareau, C., Keckhut, P., Noel, V., Chepfer, H., and Baray, J.-L.: A decadal cirrus clouds climatology
from ground-based and spaceborne lidars above the south of France (43.9° N–5.7° E), Atmos.
Chem. Phys., 13, 6951–6963, doi:10.5194/acp-13-6951-2013, 2013.
Hogan, R. J., and Kew, S. F.: A 3D stochastic cloud model for investigating the radiative properties of
inhomogeneous cirrus clouds. Q. J. R. Meteorol. Soc., 131, 2585-2608, 2005.
Hong, G., Heygster, G., Miao, J., and Kunzi, K.: Detection of tropical deep convective clouds from
AMSU-B water vapor channels measurements, J. Geophys. Res., 110, D05205,
doi:10.1029/2004JD004949, 2005.
Huffman, G.J., Adler, R.F., Bolvin, D.T., Gu, G., Nelkin, E.J., Bowman, K.P., Hong, Y., Stocker, E.F.,
and Wolff, D.B.,: The TRMM multi-satellite precipitation analysis: quasi-global, multi-year,
combined-sensor precipitation estimates at fine scale. J. Hydrometeorol. 8 (1), 38–55, 2007.
Immler, F. and Schrems, O.: LIDAR measurements of cirrus clouds in the northern and southern
midlatitudes during INCA (55° N, 53° S): A comparative study, Geophys. Res. Lett., 29, 1809,
doi:10.1029/2002GL015076, 2002a.
Immler, F., and Schrems, O.: Determination of tropical cirrus properties by simultaneous LIDAR and
radiosonde measurements, Geophys. Res. Lett., 29/23, 4, doi:10.1029/2002GL015076, 2002b.
International meteorological vocabulary. WMO, No. 182. TP. 91. Geneva (Secretariat of the World
Meteorological Organization) 1966. Pp. xvi, 276. Sw. fr. 40. Q.J.R. Meteorol. Soc., 93: 148.
doi:10.1002/qj.49709339524
Jensen, E. J., Toon, O. B., Selkirk, H. B., Spinhirne, J. D., and Schoeberl, M. R.: On the formation and
persistence of subvisible cirrus clouds near the tropical tropopause, J. Geophys. Res., 101(D16),
21361–21375, doi:10.1029/95JD03575, 1996.
Khvorostyanov, V. I., and Sassen, K.: Microphysical processes in cirrus and their impact on radiation A
Mesoscale Modeling Perspective, in Cirrus ed D Lynch, K Sassen, D O C Starr and G Stephens
(Oxford: Oxford University Press) pp 397–432, 2002.
Kienast-Sjögren, E., Rolf, C., Seifert, P., Krieger, U. K., Luo, B. P., Krämer, M., and Peter, T.,
Climatological and radiative properties of midlatitude cirrus cloud derived by automatic evaluation
of lidar measurements, Atmos. Chem. Phys., 16, 7605-7621, doi:10.5194/acp-16-7605-2016,
593     2016.

Kim, Y., Kim, S.-W., Kim, M.-H. and Yoon, S.-C.: Geometric and optical properties of cirrus clouds
inferred from three-year ground-based lidar and CALIOP measurements over Seoul, Korea,
Atmospheric Research, 139, 27-35, 2014.
Klett, J.D.: Stable analytical inversion solution for processing lidar returns. Appl. Opt. 20(2), 211–220,
598     1981.

Krämer, M., Rolf, C., Luebke, A., Afchine, A., Spelten, N., Costa, A., Meyer, J., Zöger, M., Smith, J.,
Herman, R. L., Buchholz, B., Ebert, V., Baumgardner, D., Borrmann, S., Klingebiel, M., and
Avallone, L.: A microphysics guide to cirrus clouds – Part 1: Cirrus types, Atmos. Chem. Phys.,
16, 3463-3483, doi:10.5194/acp-16-3463-2016, 2016a.
Martina Krämer, Armin Afchine, Linnea Avallone, Darrel Baumgardner, Stephan Borrmann, Bernhard
Buchholz, Anja Costa, Volker Ebert, David Fahey, Robert Herman, Eric Jensen, Marcus
Klingebiel, P. Lawson S. Woods, Anna Luebke, Jessica Meyer, Christian Rolf, A. Rollins, T.
Thornberry, Jessica Smith, Nicole Spelten, Martin Zöger, Microphysical properties of cirrus
clouds between 75N and 25S derived from extensive airborne in-situ observations, In.: XVII
International Conference on Clouds & Precipitation, Manchester, 25-29 July, 2016b.
Lakkis, G.S., Lavorato, M., and Canziani, O.P.: Monitoring cirrus clouds with lidar in the Southern
Hemisphere: a local study over Buenos Aires. 1. Tropopause heights. Atmos. Res. 92 (1), 18–26,
2009.

Lin, L., Fu, Q., Zhang, H., Su, J., Yang, Q., and Sun, Z.: Upward mass fluxes in tropical upper
troposphere and lower stratosphere derived from radiative transfer calculations, J. Quant.
Spectrosc. Radiat. Transfer, 117, 114–122, 2013.
Liou, K. N.: Influence of cirrus clouds on weather and climate processes: A global perspective. Mon.
Wea. Rev., 114, 1167–1199, 1986.

Liu, C., and Zipser, E. J., Global distribution of convection penetrating the tropical tropopause, J.
Geophys. Res., 110, D23104, doi:10.1029/2005JD006063, 2005

Liu, C., and Zipser, E. J.: Implications of the day versus night differences of water vapor, carbon
monoxide, and thin cloud observations near the tropical tropopause, J. Geophys. Res., 114,
D09303, doi:10.1029/2008JD011524, 2009.

Lynch, D. K., Sassen, K., Starr, D. O., and Stephens, G.: Cirrus. Oxford University Press, 480 pp., 2002.
Mace, G. G., R. Marchand, Q. Zhang, and G. Stephens, Global hydrometeor occurrence as observed by
CloudSat: Initial observations from summer 2006, Geophys. Res. Lett., 34, L09808,
doi:10.1029/2006GL029017, 2007.

Machado, L.A.T., Laurent, H., and Lima, A.A.: Diurnal march of the convection observed during
TRMM-WETAMC/LBA, J. Geophys. Res., 107(D20), 8064, doi:10.1029/2001JD000338, 2002.

Machado, L.A.T.; Laurent, H.; Dessay, N.; Miranda, I.: Seasonal and diurnal variability of convection
over the Amazonia - A comparison of different vegetation types and large scale forcing.
Theoretical and Applied Climatology, 78, 61-77, doi: 10.1007/s00704-004-0044-9. 2004.

Machado, L.A.T., Silva Dias, M.A.F., Morales, C., Fisch, G., Vila,D., Albrecht, R., Goodman, S.J.,
Calheiros, A.J.P., Biscaro, T., Kummerow, C.,Cohen, J., Fitzjarrald, D., Nascimento, E.L.,
Sakamoto, M.S., Cunningham, C., Chaboureau, J. –P., Petersen, W.A., Adams, D.K., Baldini, L.,
Angelis, C.F., Sapucci, L.F., Salio, P., Barbosa, H.M.J., Landulfo, E., Souza, R.A.F., Blakeslee,
R.J., Bailey, J., Freitas, S., Lima, W.F.A., Tokay, A.: THE CHUVA PROJECT: how does
convection vary across Brazil? Bull. Am. Meteor. Soc., 1365–1380, doi:10.1175/BAMS-d-13-
00084.1, 2014.

Martin, S. T., Artaxo, P., Machado, L. A. T., Manzi, A. O., Souza, R. A. F., Schumacher, C., Wang, J.,
Andreae, M. O., Barbosa, H. M. J., Fan, J., Fisch, G., Goldstein, A. H., Guenther, A., Jimenez, J.
640        L., Pöschl, U., Silva Dias, M. A., Smith, J. N., and Wendisch, M.: Introduction: Observations and
Modeling of the Green Ocean Amazon (GoAmazon2014/5), Atmos. Chem. Phys., 16, 4785-4797,
doi:10.5194/acp-16-4785-2016, 2016.

McCalla, C., 1981: Objective Determination of the Tropopause Using WMO Operational Definitions,
Office Note 246, U.S. Department of Commerce, NOAA, NWS, NMC, 18pp, October 1981.

Nazaryan, H., McCormick, M. P., and Menzel, W. P.: Global characterization of cirrus clouds using
CALIPSO data, J. Geophys. Res., 113, D16211, doi:10.1029/2007JD009481, 2008.

Pace, G., Cacciani, M., di Sarra, A., Fiocco, G., and Fuà, D.: Lidar observations of equatorial cirrus
clouds at Mahé Seychelles, J. Geophys. Res., 108(D8), 4236, doi:10.1029/2002JD002710, 2003.

Pandit, A. K., Gadhavi, H. S., Venkat Ratnam, M., Raghunath, K., Rao, S. V. B., and Jayaraman, A,:
Long-term trend analysis and climatology of tropical cirrus clouds using 16 years of lidar data set
over Southern India. Atmos. Chem. Phys., 15, 13833–13848, doi:10.5194/acp-15-13833-2015,
2015

Platt, C. M. R., Remote sounding of high clouds. III: Monte Carlo calculations of multiple scattered lidar
returns. J. Atmos. Sci., 38, 156–167, 1981

Randel, W. J. and Jensen, E. J.: Physical processes in the tropical tropopause layer and their roles in a
changing climate, Nat. Geosci, 6, 169–176, doi:10.1038/ngeo1733, 2013.

Sasano Y., and Nakane H.: Significance of the extinction/backscatter ratio and the boundary value term in
the solution for the two-component lidar equation", Appl. Opt., vol. 23, 11–13, 1984.

Sassen, K., and B. S. Cho, B. S., Subvisual/thin cirrus dataset for satellite verification and climatological
research, *J. Appl. Meteorol.*, 31, 1275–1285, 1992

Sassen, K., Starr, D. O'C., and Uttal, T.: Mesoscale and Microscale Structure of Cirrus Clouds: Three
Case Studies, J of the Atmos. Sci. 46:3, 371-396, 1989.

Sassen, K. and Campbell, J. R.: A midlatitude cirrus cloud climatology from the facility for atmospheric
remote sensing. Part I: Macrophysical and synoptic properties, J. Atmos. Sci., 58, 481–496, 2001.

Sassen, K.: Cirrus Clouds. A Modern Perspective, In Cirrus D. Lynch, K. Sassen, D. O´C Starr, and G.
Stephens Eds., Oxford University Press, 136-146, 2002.

Sassen, K., Wang, Z., and Liu, D.: Global distribution of cirrus clouds from CloudSat/Cloud-Aerosol
Lidar and Infrared Pathfinder Satellite Observations (CALIPSO) measurements, J. Geophys. Res.,
113, D00A12, doi:10.1029/2008JD009972, 2008.

Sassen, K., Wang, Z., and Liu, D.: Cirrus clouds and deep convection in the tropics: Insights from
CALIPSO and CloudSat, J. Geophys. Res., 114, D00H06, doi:10.1029/2009JD011916, 2009.

Seifert, P.; Ansmann, A.; Muâller, D.; Wandinger, U.; Althausen, D.; Heymsfield, A. J.; Massie, S. T.;
Schmitt, C.: Cirrus optical properties observed with lidar, radiosonde and satellite over the tropical
indian ocean during the aerosol-polluted northeast and clean maritime southwest monsoon. J.
Geophys. Res., v. 112, p. D17205, 2007.

Silva, V. B. S., Kousky, V. E., and Higgins, R. W.: Daily Precipitation Statistics for South America: An
Intercomparison between NCEP Reanalyses and Observations. J. Hydrometeorol., 12, 101-117.
DOI: 10.1175/2010JHM1303.1, 2011.

Stein, A.F., Draxler, R.R, Rolph, G.D., Stunder, B.J.B., Cohen, M.D., and Ngan, F., NOAA's HYSPLIT
atmospheric transport and dispersion modeling system, Bull. Amer. Meteor. Soc., 96, 2059-2077,
doi:10.1175/BAMS-D-14-00110.1, 2015

Stubenrauch, C. J., Chédin, A., Rädel, G., Scott, N. A., and Serrar, S.: Cloud Properties and Their
Seasonal and Diurnal Variability from TOVS Path-B. J. Climate, 19, 5531–5553, 2006.

Tanaka, L. M. d. S., Satyamurty, P., and Machado, L. A. T.: Diurnal variation of precipitation in central
Amazon Basin. Int. J. Climatol. 34, 3574–3584, DOI: 10.1002/joc.3929, 2014.

Thorsen, T. J., Qiang, F., and Comstock, J. M.: Comparison of the CALIPSO satellite and ground-based
observations of cirrus clouds at the ARM TWP sites, J. Geophys. Res., 116, D21203,
doi:10.1029/2011JD015970, 2011.

Thorsen, T. and Q. Fu, Automated Retrieval of Cloud and Aerosol Properties from the ARM Raman
Lidar. Part II: Extinction. J. Atmos. Oceanic Technol., 32, 1999–2023, doi: 10.1175/JTECH-D-14-
00178.1, 2015.

Wandinger, U., Multiple-scattering influence on extinction- and backscatter-coefficient measurements
with Raman and high-spectral-resolution lidars, Appl. Optics, 37, 417–427, 1998.

Wang, T., and Dessler, A. E.: Analysis of cirrus in the tropical tropopause layer from CALIPSO and MLS
data: A water perspective, J. Geophys. Res., 117, D04211, doi:10.1029/2011JD016442, 2012.

Wendisch, M., et al.: The ACRIDICON–CHUVA campaign: Studying tropical deep convective clouds
and precipitation over Amazonia using the new German research aircraft HALO. Bull. Am. Met.
Soc., accepted, doi:10.1175/BAMS-D-14-00255.1, 2016

Westbrook, C. D., Illingworth, A. J., O'Connor, E. J. and Hogan, R. J., Doppler lidar measurements of
oriented planar ice crystals falling from supercooled and glaciated layer clouds. Q.J.R. Meteorol.
Soc., 136: 260–276, 2010.

Wylie, D. P., Jackson, D. L., Menzel, W. P., andBates, J. J.: Trends in global cloud cover in two decades
of HIRS observations. J. Climate, 18, 3021–3031, 2005.

Yang, P., Hong, G., Dessler, A. E., Ou, S. C., Liou, K. N., Minnis, P., and Hashvardhan,: Contrails and
induced cirrus: Optics and radiation. Bull. Amer. Meteor. Soc., 91, 473–478, 2010a.

Yang, Q., Fu, Q., and Hu, Y.: Radiative impacts of clouds in the tropical tropopause layer, J. Geophys.
Res., 115, D00H12, doi:10.1029/2009JD012393, 2010b.

Young, S.: Analysis of lidar backscatter profiles in optically thin cirrus, Appl. Opt., 34, 7019–7031, 1995.

Tables:

Table 1. Summary of some recent cirrus cloud studies based on at least a few months of ground-based lidar observations in the tropics and mid-latitudes. The first columns show the period of study and laser wavelength (nm) for each site location, for which more than one study might be available. The cirrus characteristics are those reported by the different authors, which might include: base and top height (km), thickness (km), base and top temperature (°C), frequency of occurrence (%) and lidar-ratio (sr).

| Measurement site | Location | Period of study | Wavelength [nm] | Average values | | | | | | | | |
|---|---|---|---|---|---|---|---|---|---|---|---|---|
| | | | | Height [km] | | | Temp. [°C] | | Frequency [%] | | LR[sr] | |
| | | | | Base | Top | Thick. | Base | Top | SVC | Thin | | |
| Salt Lake City, Utah, USA | 40.8°N 111.8°W | 1986 to 1996 | 694 | 8.8 | 11.2 | 1.8 | −34.4 | −53.9 | 50 | - | | Sassen and Campbell (2001) |
| Haute Provence, France | 43.9°N 5.7°E | 1997 to 2007 | 532/1064 | 9.3 | 10.7 | 1.4 | | | 38 | | 18.2 | Goldfarb et al. (2001) Hoareau et al. (2013) |
| Thessaloniki, Greece | 40.6°N 22.9°E | 2000 to 2006 | 355/532 | 8.6 | 11.7 | 2.7 | −38 | −65 | | 57 | 30 | Giannakaki et al. (2007) |
| Seoul, South Korea | 37°N, 127°E | 2006 to 2009 | 532/1064 | 8.8 | 10.6 | | | | | | 20 | Kim et al. (2009) |
| Buenos Aires, Argentina | 34.6 °S, 58.5 °W | 2001 to 2005 | 532 | 9.6 | 11.8 | 2.4 | | −64.5 | | | | Lakkis et al.(2008) |
| Reunion Island | 21°S, 55°E | 1996 to 2001 | 532 | 11 | 14 | | | | 65 | | 18.3 | Cadet et al. (2003) |
| Camagüey, Cuba | 21.4° N, 77.9° W | 1993 to 1998 | 532 | 11.6 | 13.8 | | | | 25 | | 10 | Antuña and Barja, (2006) |
| Gadanki, India | 13.5 N, 79.2 E | 1998 to 2013 | 532 | 13.0 | 15.3 | 2.3 | | −65 | 52 | 36 | 25 | Pandit et al., (2015) |
| Hulule, Maldives | 4.1°N, 73.3°E | 1999, 2000 | 532 | 11.9 | 13.7 | 1.8 | −50 | −65 | 15 | 49 | 32 | Seifert et al. (2007) |
| Mahé, Seychelles | 4.4 °S, 55.3 °E | Feb-Mar 1999 | 532 | | | 0.2-2.0 | | | | | 19 | Pace et al., (2003) |
| Nauru Island | 0.5 °S, 166.9 °E | Apr-Nov 1999 | 532 | ~14 | ~16 | | | | | | | Comstock et al. (2002) |

Table 2. Summary of column-integrated statistics for the total time of observation, as well as for the wet, transition and dry seasons. Frequency of occurrence is calculated using a conditional sampling to avoid biases (session 2.4). Mean cirrus cloud properties and standard deviation of the sample (in parenthesis) are shown. The standard deviations of the mean were calculated and used to determine if seasonal differences (wet-dry) of the mean values are statistically significant to the 95% confidence level (indicated as *) using a 2-sample t-test. Geometrical properties are not given because most cloud profiles have more than one layer of cirrus. Lidar ratio is calculated as a column average.

|  | Total | Wet | Transition | Dry |
|---|---|---|---|---|
| Observation time [%] [a] | 37.4 | 41.5 | 21.9 | 48.9 |
| N. prof. measured [b] | 36844 | 13828 | 7423 | 15593 |
| N. prof. used in analysis [c] | 16025 | 3458 | 2099 | 10468 |
| N. prof. discarded for apparent top [d] | 476 | 223 | 148 | 105 |
| Frequency of Occurrence [%]* | 73.8 | 88.1 | 74.2 | 59.2 |
| N. prof. w/ cirrus | 11252 | 3145 | 1706 | 6397 |
| Frequency of Occurrence, Opaque [%]* | 22.6 | 31.3 | 24.6 | 11.8 |
| N. prof. w/ cirrus, Opaque | 3327 | 1316 | 610 | 1401 |
| Frequency of Occurrence, Thin [%]* | 32.8 | 37.9 | 36.5 | 23.9 |
| N. prof. w/ cirrus, Thin | 4577 | 1224 | 798 | 2555 |
| Frequency of Occurrence, SVC [%]* | 18.3 | 18.7 | 13.0 | 23.3 |
| N. prof. w/ cirrus, SVC | 3322 | 603 | 296 | 2423 |
| Cloud Optical Depth* | 0.35 (0.55) | 0.47 (0.65) | 0.40 (0.57) | 0.25 (0.45) |
| Max Backscatter Altitude [km]* | 13.4 (2.0) | 13.4 (2.2) | 13.3 (2.2) | 13.6 (1.7) |
| Temperature Max. Back. Alt. [°C]* | -60 (15) | -60 (16) | -59 (17) | -62 (13) |
| Lidar Ratio [sr]* [e] | 23.6 (8.1) | 22.8 (8.0) | 22.8 (7.8) | 24.6 (7.7) |
| Num. of cirrus layers per cloud prof. | 1.41 (0.63) | 1.62 (0.77) | 1.61 (0.67) | 1.25 (0.48) |

[a] Fraction of observation time to total possible time (21h per day)
[b] Total number of profiles measured, i.e. not screened for low clouds or precipitation
[c] Refers to the number of 5-min profiles with high enough SNR (section 2.4)
[d] Number of profiles with apparent cirrus top, considering only good profiles
[e] All layers in the same profile share the same average LR

Table 3. Summary of layer-statistics for the total time of observation, as well as for the wet, transition and dry seasons. Mean cirrus cloud properties and standard deviation of the sample (in parenthesis) are shown. The standard deviations of the mean were calculated and used to determine if seasonal differences (wet-dry) are statistically significant to the 95% confidence level (indicated as *) using a 2-sample t-test. Lidar ratio is calculated as a column average.

| *All Layers* | Total | Wet | Transition | Dry |
|---|---|---|---|---|
| Num. of cirrus layers | 15824 | 5096 | 2739 | 7989 |
| Base Altitude [km]* | 12.9 (2.2) | 12.8 (2.4) | 12.6 (2.3) | 13.0 (1.9) |
| Top Altitude [km] | 14.3 (1.9) | 14.3 (2.0) | 14.1 (2.0) | 14.3 (1.6) |
| Thickness [km]* | 1.4 (1.1) | 1.5 (1.2) | 1.5 (1.1) | 1.3 (1.0) |
| Cloud Optical Depth* | 0.25 (0.46) | 0.30 (0.52) | 0.26 (0.47) | 0.20 (0.40) |
| Max Backscatter Altitude [km] | 13.6 (2.0) | 13.7 (2.3) | 13.5 (2.2) | 13.6 (1.8) |
| Lidar Ratio [sr]* | 23.3 (8.0) | 22.6 (8.1) | 22.8 (7.9) | 24.4 (7.9) |
| Relative freq. opaque cirrus [%]* | 20.5 | 25.2 | 21.0 | 17.4 |
| Relative freq. thin cirrus [%] | 37.8 | 37.0 | 43.2 | 36.5 |
| Relative freq. SVC [%]* | 41.6 | 37.8 | 35.8 | 46.0 |
| Base above the tropopause [%]* | 5.9 | 6.9 | 5.5 | 5.3 |
| Top above the tropopause [%]* | 15.7 | 18.7 | 16.1 | 12.9 |
| | | | | |
| *Opaque Layers* | | | | |
| Num. of opaque layers | 3251 | 1283 | 574 | 1394 |
| Base Altitude [km]* | 10.7 (1.5) | 10.6 (1.6) | 10.4 (1.5) | 10.8 (1.2) |
| Top Altitude [km] | 13.4 (1.6) | 13.5 (1.7) | 13.1 (1.6) | 13.6 (1.4) |
| Thickness [km]* | 2.76 (1.02) | 2.84 (1.07) | 2.65 (1.04) | 2.73 (0.94) |
| Cloud Optical Depth* | 0.93 (0.64) | 1.00 (0.66) | 0.90 (0.66) | 0.86 (0.59) |
| Max Backscatter Altitude [km] | 12.0 (1.7) | 12.1 (1.9) | 11.6 (1.7) | 12.1 (1.5) |
| Lidar Ratio [sr]* | 25.7 (6.3) | 26.0 (6.7) | 25.8 (6.6) | 25.3 (5.7) |
| | | | | |
| *Thin Layers* | | | | |
| Num. of thin layers | 5985 | 1888 | 1183 | 2914 |
| Base Altitude [km]* | 12.9 (1.7) | 13.1 (1.9) | 12.9 (1.8) | 12.8 (1.4) |
| Top Altitude [km]* | 14.4 (1.7) | 14.6 (2.0) | 14.4 (1.8) | 14.3 (1.4) |
| Thickness [km]* | 1.46 (0.78) | 1.42 (0.82) | 1.49 (0.78) | 1.47 (0.74) |
| Cloud Optical Depth | 0.12 (0.07) | 0.12 (0.07) | 0.12 (0.07) | 0.11 (0.07) |
| Max Backscatter Altitude [km]* | 13.7 (1.7) | 13.9 (1.9) | 13.7 (1.9) | 13.5 (1.5) |
| Lidar Ratio [sr]* | 22.8 (7.9) | 21.8 (7.7) | 21.6 (7.4) | 24.3 (8.1) |
| | | | | |
| *SVC Layers* | | | | |
| Num. of SVC layers | 6581 | 1924 | 980 | 3677 |
| Base Altitude [km]* | 14.4 (1.9) | 14.7 (2.1) | 14.4 (2.1) | 14.2 (1.6) |
| Top Altitude [km]* | 14.9 (1.9) | 15.2 (2.1) | 15.0 (2.1) | 14.7 (1.6) |
| Thickness [km] | 0.51 (0.37) | 0.50 (0.38) | 0.53 (0.38) | 0.51 (0.36) |
| Cloud Optical Depth | 0.011 (0.008) | 0.011 (0.008) | 0.012 (0.009) | 0.011 (0.008) |
| Max Backscatter Altitude [km]* | 14.6 (1.9) | 14.9 (2.1) | 14.7 (2.1) | 14.4 (1.6) |
| Lidar Ratio [sr]* | 21.6 (8.4) | 19.9 (7.6) | 21.5 (8.1) | 23.5 (9.0) |

Figures:

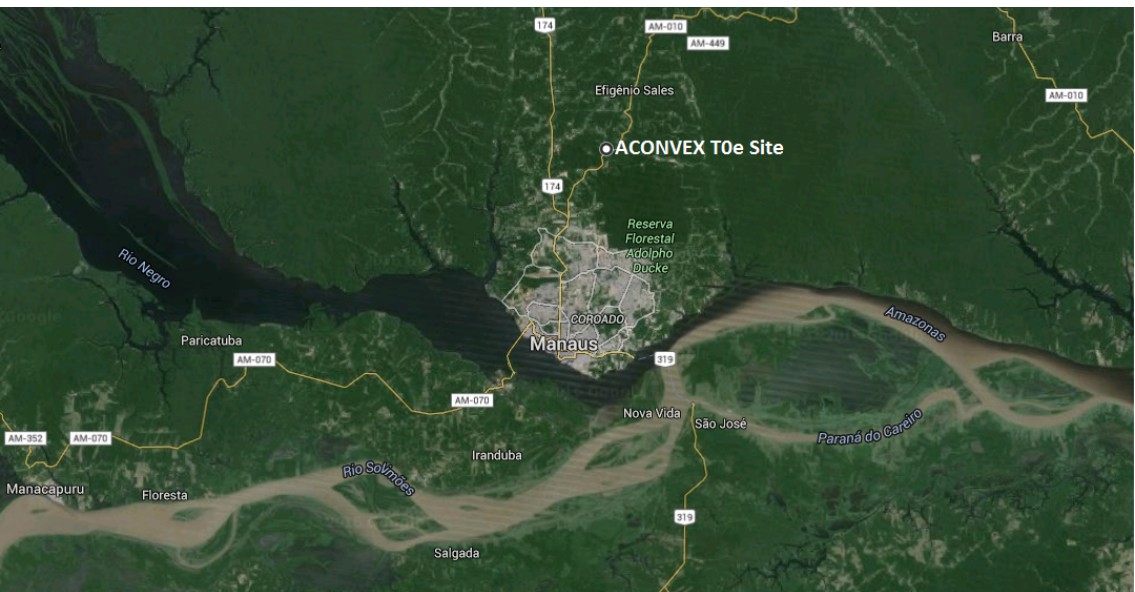

Figure 1. Satellite-based map (Google Earth) showing the location of the lidar site (ACONVEX T0e, 2.89$^{\circ}$S 59.97$^{\circ}$W), 30 km upwind (north) from downtown Manaus-AM, Brazil.

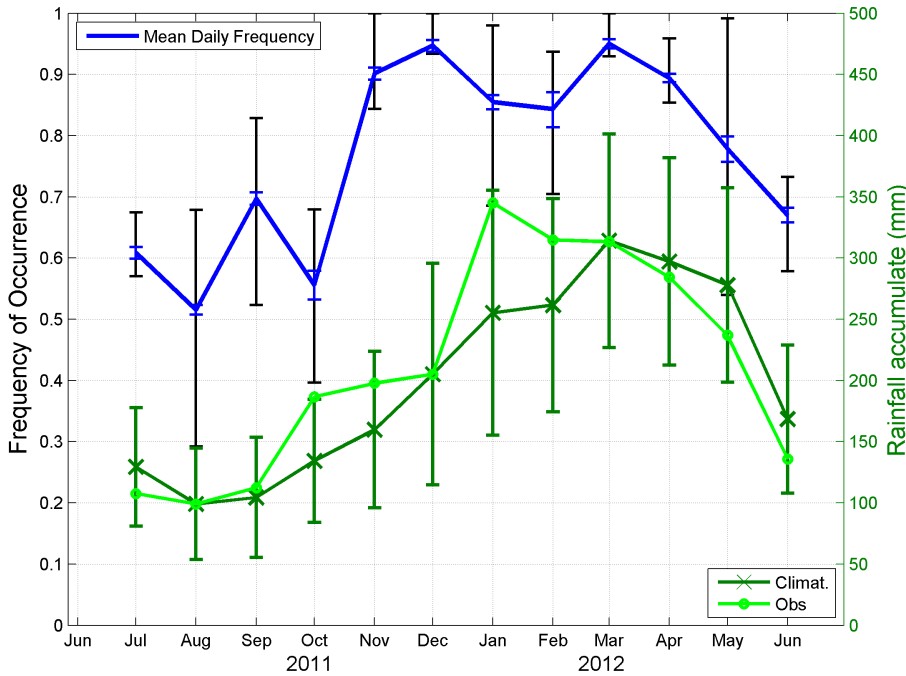

Figure 2. Monthly frequency of occurrence of cirrus clouds from July 2011 to June 2012 (blue line) with the associated statistical error (black). Accumulated (light green) and climatological (dark green) rainfall, shown on the right axis, were obtained from the TRMM 3B42 version 7 dataset averaged over an area of 10°x10°.

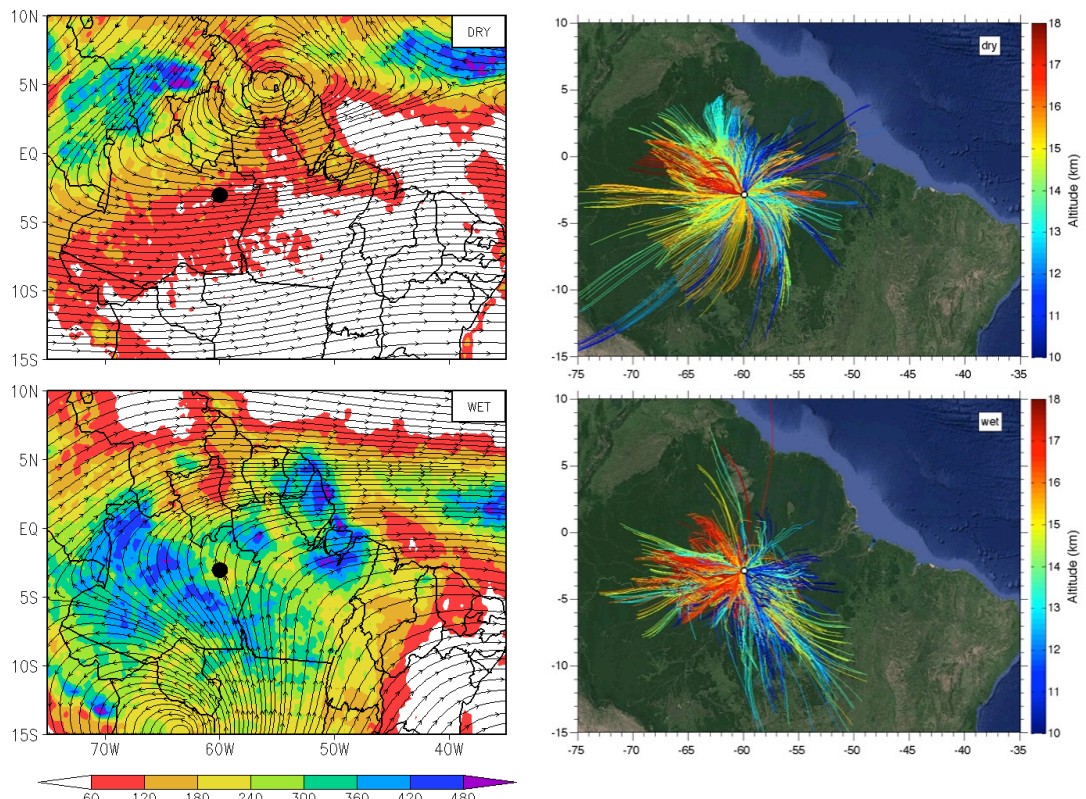

Figure 3. Left panels show mean precipitation (colors, mm month$^{-1}$) from the TRMM 3B42 version 7 and mean wind field (vectors, m/s) at 150 hPa (~ 14.3 km) from ECMWF ERA Interim reanalysis. Right panels show 24 h back trajectories of air masses arriving at the site at the time and altitude that cirrus layers were detected. Results are shown separately for the dry (JJAS, top) and wet months (JFMA, bottom). Backward trajectories were computed using HYSPLIT model with 0.5° resolution winds from GDAS/NOAA. The experimental site location is indicated in all panels with a circle.

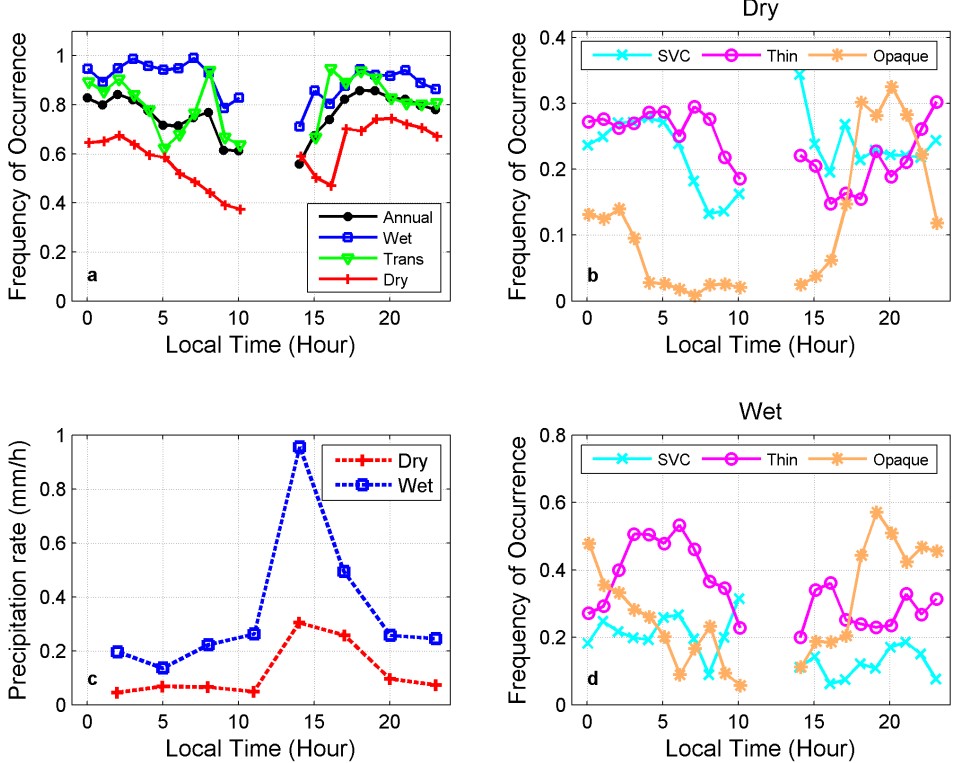

Figure 4. Panel (a) shows the daily cycles of the hourly frequency of occurrence of cirrus clouds for the annual, wet, transition and dry periods. The same is shown for SVC, thin and opaque cirrus clouds during the dry (b) and wet (d) seasons. Mean observed precipitation rate (mm/h) from TRMM version 7 over an area of 2° × 2° centered on the site, for the dry and wet periods, is given in panel (c).

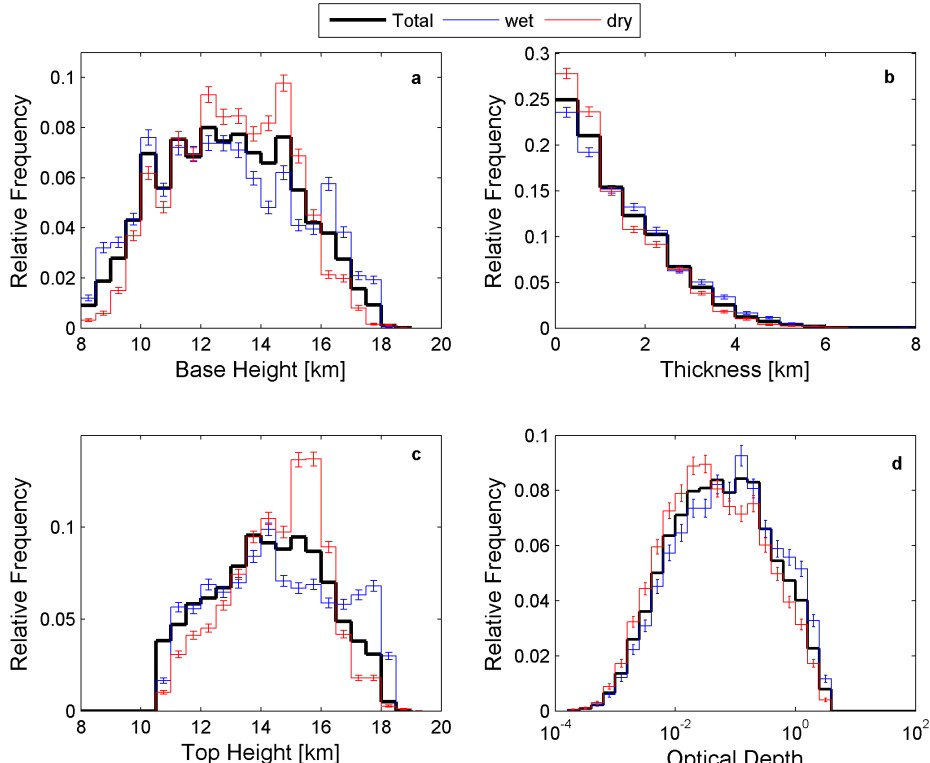

Figure 5. Panels show the normalized histograms of (a) cirrus cloud base, (b) cloud geometrical thickness, (c) cirrus cloud top, and (d) optical depth, for the overall period (black), wet season (JFMA, red) and dry season (JJAS, blue). Error bars indicate the counting statistics uncertainty.

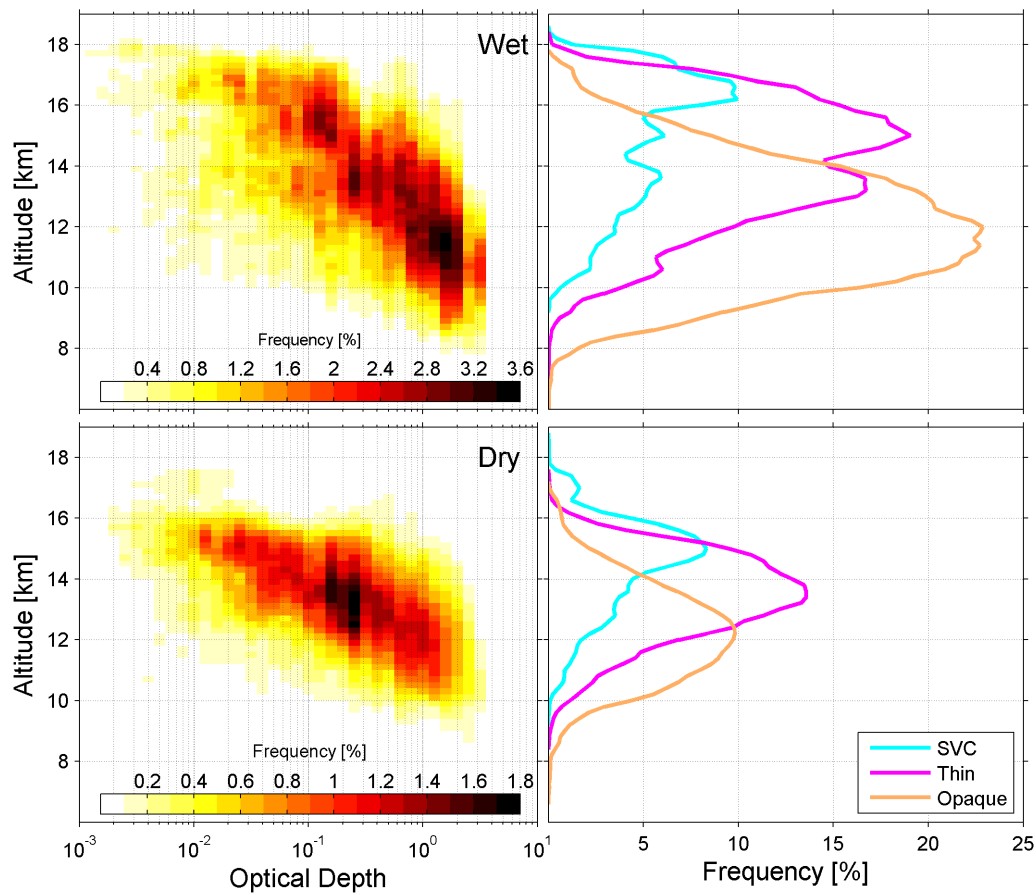

Figure 6. Two-dimensional histograms of cirrus frequency of occurrence with altitude as a function of optical depth during the wet (top) and dry (bottom) season months are shown on the left. The same is shown on the right but integrated for SVC, thin and opaque cirrus clouds optical depths.

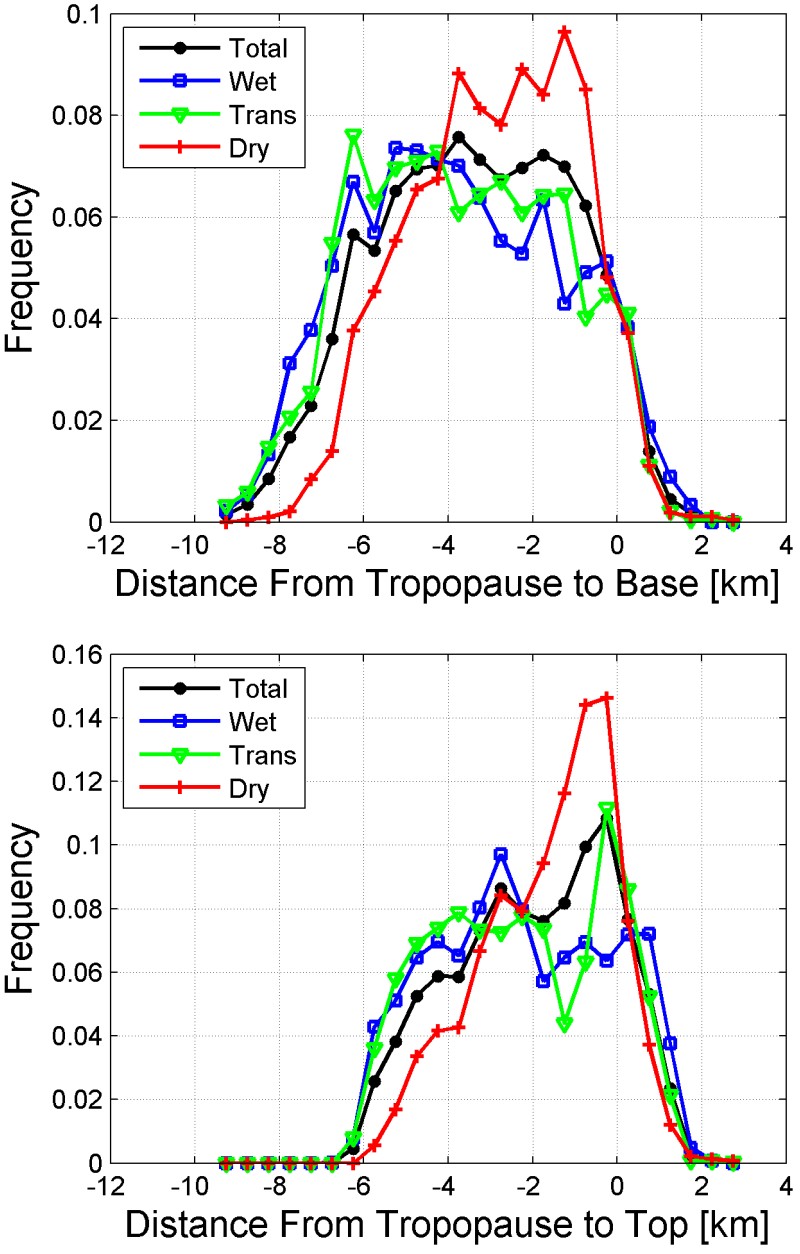

Figure 7. Normalized histograms of the distance of the tropopause to the cirrus base and top are shown for overall period (black) and each season (colors). Negative values mean that clouds are below tropopause. The average tropopause altitude was 16.2±0.4 km.

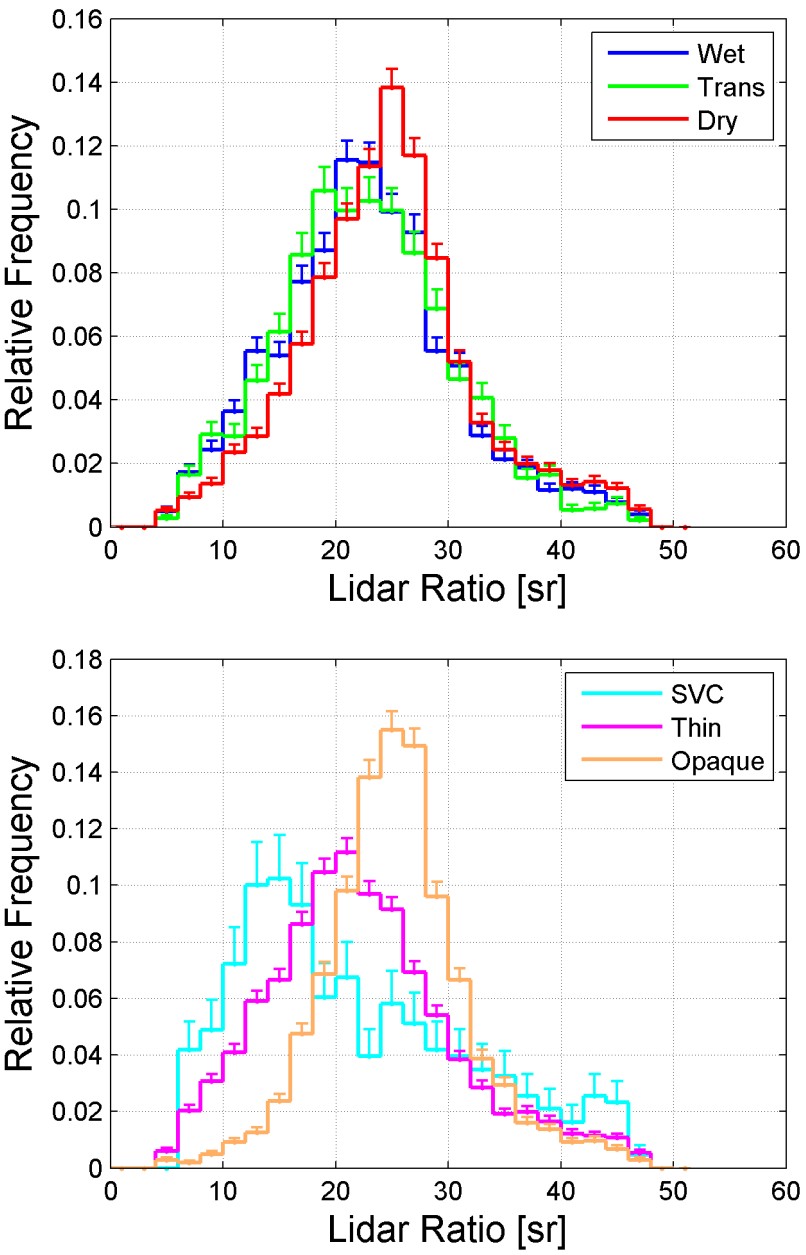

Figure 8. Normalized histograms of the lidar ratio, already corrected for multiple-scattering, for the different seasons (top) and for SVC, thin and opaque cirrus (bottom) are shown. Error bars indicate the counting statistics uncertainty.

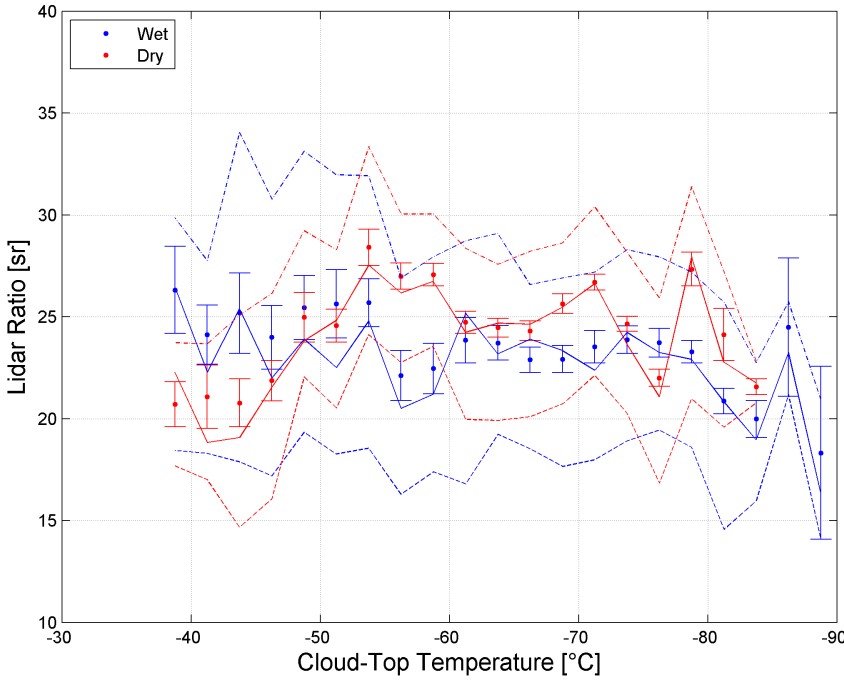

Figure 9. Dependence of the corrected lidar ratio with cloud-top temperature is shown for the wet (blue) and dry (red) seasons. The markers give mean and standard deviation of the mean. The continuous and dashed lines give median and interquartile distance. Temperature is divided in 2.5 °C intervals.