# Peer review of "Optical and Geometrical Properties of Cirrus Clouds in"

_Atmospheric Chemistry and Physics, 2016_

## Referee Comment (RC1) · Anonymous Referee #1 · 26 Jul 2016

The authors present cirrus statistics from a ground based Raman lidar. Retrievals are performed using only the elastic signal of the lidar. The authors use previous studies to speculate on physical reasons for seasonal and diurnal difference they report. Overall, the paper is well written, organized and easy to follow. However, the authors should be less aggressive when speculating on their statistics and reading into differences that are not statistically significant.

Much of the paper is spent discussing season and diurnal differences in the cirrus statistics. However, little attenuation is paid to whether these differences are actually statistically significant. Some figures/tables give the standard deviations, but little discussion of them is given in the text leaving the reader to determine significance themselves. In Table 2, it appears that non of the statistics differ significantly: i.e. one cannot say that the cirrus differ at all from season to season. Similarly in Figure 2, the box plot reveals that there is no significant seasonal cycle in frequency of occurrence either. In other figures where histograms are given, a statistical test should be applied to ensure difference among distributions are statistical significant before they are discussed. It is only appropriate/worthwhile to discuss differences that are statically significant.

I would also caution the authors against extrapolating too much from their relatively limited data. An example of this is using the lidar ratio to infer the ice crystal habit. The lidar ratio alone cannot be used to identify the ice crystal habit since it also depends on the particle orientation relative to the laser beam. In addition, theoretical studies of ice crystal phase functions vary wildly so there is no real consensus on what the lidar ratio even is for different ice crystals.

The authors note that this ground-based site is unique compared to others reported in previous work, yet rely heavily on previous work to explain their results. The paper would be greatly enhanced by making a more quantitative effort the explain their data. For example, instead of speculating on the sources of moisture for the cirrus in different seasons, a more convincing approach would be to run back trajectories to show the reader where the air came from.

Is there reason the authors don't use the nitrogen signal to retrieve extinction? Not doing so doesn't completely discount the data presented, but it does devalue it somewhat since this paper is just another in a long-line of elastic lidar cirrus studies. In addition, the transmission method is really only accurate for mid-range optical depths. Too thin and there isn't enough transmission signal to get a reliable optical depth. Too thick and there isn't enough molecular signal above the cloud. I encourage the authors to go beyond just checking the SNR above/below the cloud when doing the optical depth retrieval and to fully derive the uncertainty in the optical depth values they report. Figures 5 and 6 show optical depths down to 0.001, which I expect to be extremely uncertain when using the transmission method to retrieval optical depth.
* * *
Interactive
comment

The treatment and discussion of multiple scattering could be improved. Although, not explicitly stated, I'm guessing the authors use Eq. (10) from Chen et al. (2002) where eta depends on the optical depth of the cloud layer. I'd would encourage against using this equation. Chen et al. provide no physical justification for this equation and the values for larger optical depths quickly approach the wide angle scattering limit of eta=0.5 which is unrealistic for the geometry of a ground based lidar. In addition, for optical depth greater than about 1.2, eta<0.5 which is unphysical. The authors should also keep in mind that the shorter wavelength of 355nm (compared to 532nm as is used in Chen et al. 2002 and many other studies) means much stronger forward scattering and therefore larger amounts of multiple scattering. Typical extinction biases could range from 5-30% and sometimes even larger (see Thorsen and Fu, JTECH 2015 Fig. 13). I would suggest the authors make clear to the reader that their optical depth may contain significant biases due to multiple scattering unless some type of explicit treatment of multiple scattering is performed.

---

## Referee Comment (RC2) · J. Campbell (Referee) · 4 Aug 2016

This paper describes ground based lidar measurements of upper tropospheric clouds over one year collected at a site in Brazil. Physical and optical cloud properties are described for a series of queries that characterize the state of upper tropospheric clouds for an equatorial locale where few datasets have been collected and reported in the literature. The paper is well organized. The narrative is well written. The figures have been well designed, and are clear/legible. The subject matter is wholly appropriate for ACP, as ground-based lidar observations remain critical context for evaluating cloud and aerosol properties from satellite observation. Long-term records of clouds and aerosols from such sites have been reported for years, and remain a critical fixture of the peer reviewed literature.

This is the first time that this reviewer has considered this manuscript.

My summary recommendation for the Editor is that this paper should undergo a major revision for both scientific and technical content. I'm attaching my technical notes, which include a series of editing recommendations and minor questions. I will list my major concerns here in this portion of the review.

My primary scientific concern relates to the definition of "cirrus" clouds in the manuscript. I will sign this review, so its important that I reconcile my concern with existing literature. 45 years after the first long term reports of cirrus clouds began appearing in the literature (Platt), the community has reached a point where we should and need to be much more diligent with how we characterize our observations for peer review. Cirrus clouds are a phenomenological classification based on ground-apparent observations of ice-phase clouds in the upper troposphere. In a recent paper that I authored (Campbell et al. 2015), we went to significant length to demonstrate a practical and viable definition for cirrus clouds in autonomous long-term datasets like this one, and in particular for those that lack a polarized backscatter measurement. Whereas I am a primary advocate of papers just like this, and have participated in multiple studies documenting cirrus in a manner consistent with the narrative here, I cannot advocate for a paper that used a simple thermal threshold like -25 C as being a practical delineator for cirrus cloud presence. This absolutely has to be revised. I recognize that this is a serious request, and I raise this point very respectful of the work that has been put into the manuscript, the statistics and the analysis. However, I question every number you have in here, again respectfully, because of such a simple and non-physical definition applied for discriminating these clouds.

I would respect if the authors were to disagree with our conclusions/recommendations in Campbell et al. (2015). But, in response, they'd better come up with a physically-based reason for doing so. Cloud top temperatures of -37 makes physical sense for the class of clouds that we call cirrus. Its a practical and defendable threshold. The lidar community, as we argue in our paper, cannot continue to produce datasets with haphazard classifications and expect anyone in the climate community to take our work seriously.

This point must be addressed for clarity and consistency.

MINOR SCIENTIFIC POINTS (in order of the manuscript)

- Its unclear what the authors are saying about the presence of SNR > 3 in the upper troposphere with respect to cloud observation. Do they mean clear-sky? Or, do they mean within particulate scattering layers?

- Its unclear how the authors define the tropopause, and thus accommodate the potential for resolving the bottom/top of the tropopause transition layer, in Section 2. This hurts the discussion later on where context is necessary for understanding where the clouds are with respect to this boundary.

- Since the sample size is stated to relative to the ability to measure SNR > 3 in the upper troposphere, all of the samples appear to be relative occurrence frequencies and not absolute ones. This is HIGHLY confusing. There is no way that you're resolving an absolute cloud frequency of 67%. In a new paper that we have in Early Online Release in JAMC (Campbell et al. 2016), we show in a year's worth of MPLNET observations at Greenbelt, MD an absolute frequency near 16%, which owes to the attenuation of the beam from low-level clouds and undersampling of the upper troposphere. There are multiple places in the narrative where serious confusion arises and the speculative discussion becomes meaningless because of this confusion.

- Speaking of this issue, nothing is said of the work of Thorsen et al. (2011) and Protat et al. (2014) and undersampling issues relating to ground-based profiling, attenuation, and the relative cloud samples that we have to analyze. This is a serious weakness that leads to three other points of concern.

- It is discussed that the lowest cloud observations occur around solar noon (10-12 LT). This leads me to believe that your instrument is suffering from issues with SNR from the bright background, even at 355 nm. Whereas it is introduced that this is potentially a real artifact, I see no reason to take such a claim at face value. As I cannot evaluate your algorithm or its performance, and with the practical understanding that you are willing to deal with cloud samples in the algorithm at an SNR as low as 3, I cannot help but conclude that you're dealing with sampling issues due to background noise.

- Furthermore, all of the speculation about the transport of clouds vs. near-source convective generation is very weak. The authors are forgetting that if the clouds are being generated at/near or on top of them, the lidar will not be profiling the clouds. You are *always* dealing with transport of some kind, as such. I recognize what they are trying to say, but recommend they be much more circumspect about how they are delineating source/transport with respect to the limited information that they have.

- The distribution of clouds as a function of COD also relates to sample bias and attenuation effects. Yes, there is an exponential distribution of cirrus cloud occurrence with respect to COD (again, see what we have in Campbell et al. 2015). However, the distributions that you have with respect to subvisible, optically-thin and opaque clouds is absolutely not consistent with other studies. There should rougly be a 50-60%/40-50% distribution between transluscent and opaque clouds. In Campbell et al. (2016), we see a very similar distribution as yours that we fully attribute to sampling bias. I see no reason to think this sample is not subject to the same effects.

- Although there is a point where the authors show a correlation between COD and cloud base, cloud base is a nearly useless parameter for such vigorous study. As myriad Sassen papers discuss and describe, cloud top is the most important layer because this is where cirrus cloud nucleate, grow and begin falling. Cloud base, as such, is redundant. Its simply the boundary where evaporation/sublimation is complete in falling crystals. So much effort in the narrative is spent on cloud base and drawing physical correlation, where it seems to have no physical meaning. Cloud top should be the focal point.

- As such, there is absolutely no physical basis for evaluting lidar ratio versus mid-cloud temperature. It makes absolutely no physical sense. Now, I recognize that the CALIPSO team has done this very thing with their analyses. I don't agree with them either. But, they are dealing with a downward looking dataset, at least, and this offers other challenges that the authors are not dealing with in the zenith. Whereas I would accept if the authors referenced Garnier et al. (2015) and wanted to leave this as is, I still wouldn't think that it made much physical sense. In particular, as with CALIPSO, you're never actually going to know for certain what the mid-cloud temperature is (or unfortunately the cloud top temperature is) because of attenuation. For CALIOP, this is actually a bigger issues, since they can attenuate working downward with clouds that ground-based lidars would likely never reach. But, the comment still remains. I recommend sticking with what you can physically interpret, and particle effective size and habit are likelier in the long run to relate with available water vapor and temperature found at cloud top than somewhere within the cloud.

- No uncertainty analysis is provided for the lidar ratio analysis. This concerns me, again, because of the low SNR environments that you claim to be working with. As such, its unclear to me that you can actually develop meaningful correlative relationships, like Garnier, with a relatively low number of cases that the SNR would be sufficient and uncertainty suppressed. The uncertainty term presented appears to me to be a standard deviation, which again seems misrepresentative in context.

- Please see my note about how you interpreted Chew et al. (2011). Its not correct. 34% of Level 2 AERONET observations were found biased by unscreened cirrus.

I recognize that this is a lot of stuff. I offer this with full respect to what you are trying to do, because its in my direct interest working so many years with MPLNET to see this sort of work get published. I present these thoughts in detail with the sincere hope of helping resolve what I believe to be significant scientific shortcomings in the narrative as it is. I wish you the best.

J. Campbell Monterey, CA USA

Please also note the supplement to this comment:
http://www.atmos-chem-phys-discuss.net/acp-2016-458/acp-2016-458-RC2-supplement.pdf

**Supplement:**

[Figure]

[Figure]

**Optical and Geometrical Properties of Cirrus Clouds in Amazonia Derived From 1-year of Ground-based Lidar Measurements**

Diego A. Gouveia[1], Boris Barja[1,2], Henrique M. J. Barbosa[1], Theotônio Pauliquevis[3], and Paulo Artaxo[1].

[1]Applied Physics Department. Institute of Physics, University of São Paulo (USP), São Paulo, SP, Brazil.

[2]Atmospheric Optics Group of Camagüey. Meteorological Institute of Cuba, Cuba.

[3]Department of Natural and Earth Sciences, Federal University of São Paulo, Diadema, SP, Brazil.

*Correspondence to:* Boris Barja Gonzalez (bbarja@gmail.com)

**Abstract.** For one year, from July 2011 to June 2012, a ground-based raman lidar provided atmospheric observations north of Manaus, Brazil, at an experimental site (2.89°S and 59.97°W) for long-term aerosol and cloud measurements. Upper tropospheric cirrus clouds were observed more frequently than previous reports in tropical regions. The frequency of occurrence was found to be as high as 82 % during the wet season and not lower than 55 % during the dry season. The diurnal cycle shows a minimum around local noon and maximum during late afternoon, associated with the diurnal cycle precipitation. Optical and geometrical characteristics of these cirrus clouds were derived. The mean values were $14.4 \pm 2.0$ km (top), $12.7 \pm 2.3$ km (base), $1.7 \pm 1.5$ km (thickness), and $0.36 \pm 1.20$ (cloud optical depth). Cirrus clouds were found at temperatures down to $-90$ °C and 7 % were above the tropopause base. The vertical distribution was not uniform and two cloud types were identified: (1) cloud base $>$ 14 km and optical depth ~0.02, and (2) cloud base $<$ 14 km and optical depth ~0.2. A third type, not previously reported, was identified during the wet season, between 16 and 18 km with optical depth ~0.005. The mean lidar ratio was $20.2 \pm 7.0$ sr, indicating a mixture of thick plates and long columns. However, the clouds above 14 km have a bimodal distribution during the dry season with a secondary peak at about 40 sr suggesting that thin plates are a major habit. A dependence of the lidar ratio with cloud temperature (altitude) was not found, thus indicating they are well mixed in the vertical. Cirrus clouds classified as subvisible ($\tau <$ 0.03) were 40 %, whilst 37.7 % were thin cirrus ($0.03 < \tau < 0.3$) and 22.3 % opaque cirrus ($\tau > 0.3$). Hence, not only does the central Amazon have a high frequency of cirrus clouds, but a large fraction of subvisible cirrus clouds as well. This high frequency of subvisible cirrus clouds may contaminate aerosol optical depth measured by sun-photometers and satellite sensors to an unknown extent.

**1. Introduction**

*See Mare et al 2004*
*40-60%*

[Figure]

Cirrus clouds cover on average more than  of the Earth's atmosphere, with higher fractions occurring in the Tropics, hence, are important to understanding current climate and predicting future climate (Wylie

et al. 2005, Stubenrauch et al. 2006; Nazaryan et al., 2008). Several studies emphasize the important role that cirrus clouds play in the Earth's radiation budget (i.e. Liou 1986; Lynch et al. 2002; Yang et al.

2010a). Their role is twofold. Firstly, cirrus clouds may increase warming by trapping a portion of infrared radiation emitted by the Earth/atmosphere system. Secondly, cirrus clouds could cool the atmosphere by reflecting part of the incoming solar radiation back into space. The contribution of each effect and the net effect on the radiative forcing depends strongly on cirrus cloud optical properties, altitude, vertical and horizontal coverage (Liou 1986). Therefore, understanding their properties is critical to determining their effect on the albedo and greenhouse effects (Barja and Antuña, 2011, Boucher et al.,

2013). Also, the tropical cirrus clouds could influence the vertical distribution of radiative heating in the tropical tropopause layer (e.g., Yang et al., 2010b; Lin et al., 2013). Noticeably, it has been shown that an accurate representation of the cirrus vertical structure in cloud radiative studies improved the results of these calculations (Khvorostyanov and Sassen, 2002; Hogan and Kew, 2005; Barja and Antuña, 2011).

Recent research also shows that an increase in stratospheric water vapor are linked mainly with the occurrence of cirrus clouds in the tropical tropopause layer (TTL) (Randel and Jensen, 2013). Finally, measurements of the properties of cirrus clouds at different geographical locations are of utmost importance, potentially allowing for improvements in numerical models parameterizations and, thus, reducing the uncertainties in climatic studies.

Ground-based lidars are an indispensable tool for monitoring cirrus clouds, particularly identifying optically thin and subvisible cirrus clouds (SVC) with very low optical depth, which are undetectable by cloud radars (Comstock et al., 2002) or by passive instruments (e.g., Ackerman et al., 2008). For this reason, several studies with ground-based lidars have reported the characteristics of cirrus clouds around the globe during the last decade. There are some long-term studies reporting climatologies from midlatitude (eg. Sassen and Campbell, 2001; Goldfarb et al., 2001; Giannakaki et al., 2007; Hoareau et al., 2013) and tropical regions (eg. Comstock et al., 2002; Cadet et al., 2003; Antuña and Barja, 2006;

Thorsen et al., 2011; Pandit et al., 2015). Table 1 shows an overview of these studies with different values for cirrus clouds characteristics in diverse geographical regions. There are also some short-term reports on cirrus clouds characteristics during measurement campaigns in midlatitude (e.g. Immler and Schrems,

2002a) and tropical latitudes (Immler and Schrems, 2002b, Pace et al., 2003 and references therein).

Additionally, satellite-based measurements have been used to investigate the global distribution of cirrus characteristics (eg. Nazaryan et al., 2008; Sassen et al., 2009; Sassen et al., 2009; Wang and Dessler 2012,

Jian et al., 2015). Characteristics of tropical and subtropical cirrus clouds have similar geometrical values and these values are higher than those in midlatitudes. The frequencies of occurrence of cirrus cloud types differ significantly between different locations.

Cirrus clouds measurements reports over tropical rain forests are scarce. Very few global studies with satellites instruments include these regions. Some studies focused on deep convection in the Amazonia reported cirrus clouds (eg. Machado et al., 2002; Hong et al., 2005, Wendisch et al., 2016), but no lidar measurements were used. Baars et al. (2012) focused on aerosol measurements with a ground-based

Raman lidar, but report only one cirrus cloud case between 12 km and 16 km during September 11, 2008.

Barbosa et al., (2014) describe a week of cirrus clouds measurements from 30 August to 6 September

2011 during an intensive campaign for calibration of the water vapor channel of the UV Raman lidar was

DEFINE
Subvisible
cloud 42

conducted in the ACONVEX (Aerosols, Clouds, cONVection EXperiment) site. Cirrus clouds during this period were present in 60% of the measurements. Average base and top heights were 11.5 km and

13.4 km, respectively, and average maximum backscatter occurred at 12.8 km. Most of the time, three layers of cirrus clouds were actually found.

From the above discussion, the importance of continuous and long-term observations of tropical cirrus clouds is evident. In the present study, we use one year of ground-based lidar measurements (July 2011 to

June 2012) at Manaus, Brazil to investigate the seasonal and diurnal variability of geometrical (cloud top and base altitude) and optical (cloud optical depth and lidar ratio) properties of cirrus over a tropical rain forest site. In section 2, a brief description of the Raman lidar system, dataset, processing algorithms and site are given. The results and discussion are presented in section 3. We close this paper with concluding remarks in section 4.

**2.  Instrumentation, dataset and algorithms.**

**2.1.  Site and instrument description**

The ACONVEX (Aerosols, Clouds, cONVection EXperiment) or T0e (nomenclature of the

GoAmazon2014/15 experiment sites, Martin et al. 2016) site is located upwind from Manaus-AM,

Brazil, at 2.89°S and 59.97°W, in the center of the Amazon Forest. The Atmospheric observations began in 2011 at this site, and the objective was to operate a combination of several instruments for measuring atmospheric humidity, clouds and aerosols, as well as processes which lead to convective precipitation (Barbosa et al., 2014). Figure 1 gives an overview of the location where the measurements in this study  _were collected_. As with most tropical continental sites, the diurnal cycle is strong with a late afternoon peak in precipitation (Adams et al., 2013).  The usual climatological seasons in Central Amazon are: the wet (December to April), dry (July and August), and the transitions wet-to-dry (May and June) and dry-to-wet (September to November) (Machado et al., 2004), however the definition ≲

may vary (e.g. Arraut et al., 2012, Tanaka et al., 2014). Deep convection is a characteristic of the region during both seasons, being more active during the wet season (Machado et al., 2002), when it is influenced by the intertropical convergence zone (ITCZ). As the ITCZ moves northward during the months of dry season, the convective activity decreases. Hence, it is to be expected that deep convection is the principal cirrus clouds formation mechanism in the region.

The lidar system (LR-102-U-400/HP, manufactured by Raymetrics Advanced Lidar Systems) operates in the UV, at 355 nm and has also two Raman channels for nitrogen (387 nm) and water vapor (408 nm).

The system is tilted from the zenith 5° to avoid specular reflection of horizontally oriented ice crystals. It is automatically operated 7 days a week, only being closed between 11 am and 2 pm local time (LT is

−4 UTC) to avoid the sun crossing the field of view. Detailed information about the lidar system and its characterization are given by Barbosa et al. (2014). To retrieve the particle backscatter and extinction profiles from the lidar signal, the temperature and pressure profiles were obtained from the radio soundings launched at 0 and 12 UTC from the Ponta Pelada Airport, located 28.5 km to the South (3.14°S, 59.98°W) of the experimental site.

[Figure]

**2.2. Datasets**

The lidar dataset used in the present study comprises measurements between July 2011 and June 2012. A

total of 36,597 5-minute profiles were analyzed and only 20,752 had a signal to noise ratio (SNR) higher than 3 at the characteristic altitudes of the possible cirrus clouds occurrence (between 8 km and 20 km).

Statistical tests (not shown) were conducted to obtain the lowest SNR value suitable to detect subvisible cirrus clouds, and the value 3 was selected as a threshold for obtaining a good SNR. The number of 5-min lidar profiles and number of profiles with good SNR during each month of the studied period were analyzed. July, August and September, the driest months (Figure 2) show the higher fraction of profiles with good SNR, while the wettest months have the lowest fraction of lidar profiles with good SNR (see figure S.1). The cloud fraction of low, optically thick clouds increases during this season, thereby attenuating the signal and reaching the cirrus clouds altitudes with a low SNR. The frequency was then defined as the ratio between the number of lidar profiles of 5 min with good SNR containing cirrus clouds and the total number of profiles with good SNR. This frequency does not count the number of individual clouds, but the time coverage of these clouds. Thus, the frequency of occurrence was the best estimate, for a ground-based lidar, of the fraction of time when the sky is covered with cirrus clouds of different geometric and optical characteristics.

Temperature, pressure, geopotential height, humidity and winds for the study period were obtained from the ERA Interim reanalysis (Dee et al., 2011) of European Center for Midrange Weather Forecast (ECMWF) with spatial resolution of 0.75° and temporal resolution of 6 h. This dataset was used to obtain the mean high level winds, near to the cirrus clouds habits (200 hPa). Moreover, the tropopause altitudes were obtained from vertical profiles over the site using the methodology of the World Meteorological

Organization (FCM-H3-1997). A precipitation dataset for the same period was acquired from TRMM

(Tropical Rainfall Measuring Mission) version 7 product 3B42 (Huffman et al., 2007) with 0.25° and 3 h of spatial and temporal resolution, respectively.

**2.3. Cirrus cloud detection algorithm.**

We used an automatic algorithm for the detection of cloud base, top and maximum backscattering heights, based on Barja and Aroche (2001). This algorithm assumes a monotonically decreasing intensity of the lidar signal with altitude in a clear atmosphere and searches for significant abrupt changes. These abrupt changes are marked as a possible cloud base. Examining the signal noise and the change between the possible cloud base, a true cloud base is discriminated. Then, the lowest altitude above cloud base with signal lower than that at cloud base and corresponding to a molecular gaseous atmosphere is determined as the cloud top. When more than one cloud is present in the same profile, and their top and base are separated more than 400 m, they are considered as individual clouds. Figure S.2 gives an example of the cloud detection algorithm. Barbosa et al. (2014) provide details on the fully automated algorithm, which includes discrimination of false alarm and distinguishing aerosols from thin cloud layers. After obtaining the base, top and maximum backscatter heights, the corresponding cloud temperatures is obtained from the nearest radiosonde. A detected high cloud is classified as a cirrus cloud if the layer has a temperature equal or below than −25°C. These temperatures are reached above 8 km in our experimental site almost all the time.

[Figure]

**2.4. Cirrus Cloud Optical Depth, backscattering coefficient profile and lidar ratio**
**determination.**

The attenuation of the lidar signal by cirrus clouds can be obtained using the ratio of the range corrected signal at the top and at the cloud base as in (Young, 1995):

$\frac{S(z_t)}{S(z_b)} = \frac{\beta(z_t)}{\beta(z_b)} e^{-2\int_{z_b}^{z_t}\alpha_p(z')dz'} e^{-2\int_{z_b}^{z_t}\alpha_m(z')dz'}$  (1)

where $z_b$ and $z_t$ are the base and top of cirrus clouds heights, $S(z) = P(z)z^2$ is the range corrected signal.

$\beta(z)$ and $\alpha(z)$ are the volumetric backscattering and extinction coefficients, respectively, and each is the sum of a molecular (sub index m) and a particle (sub index p) contribution. Volumetric backscattering and extinction profiles from molecules were derived following Bucholtz (1995). Assuming a negligible aerosol contribution in the atmospheric layers just below and above the cirrus clouds (Young, 1995), we can express the transmittance factor of the lidar equation due to cirrus cloud, $T^{cirrus}$, as

$T^{cirrus} = e^{-2\int_{z_b}^{z_t}\alpha_p(z')dz'} = \frac{S(z_t)}{S(z_b)} \frac{\beta(z_b)}{\beta(z_t)} e^{2\int_{z_b}^{z_t}\alpha_m(z')dz'}$  (2)

And the cirrus optical depth (for an example, see Figure S.2), $\tau^{cirrus}$, as

$\tau^{cirrus} = \int_{z_b}^{z_t}\alpha_p(z')dz' = -\frac{1}{2}\ln(T^{cirrus})$  (3)

The accuracy of this calculation depends mainly on the SNR at the cirrus cloud altitude. However, when the lidar signal is completely attenuated by the cirrus cloud (i.e. the transmission factor approaches zero) it is impossible to obtain the true values of the cirrus top altitude and optical depth. The retrievals, in these cases called apparent values, are necessarily underestimated. Tilting the system by about 5° from the zenith minimizes the effect of the specular reflection on the quasi-horizontal ice crystals.

The backscattering coefficients of cirrus clouds were determined by the Fernald-Klett-Sasano method (Fernald et al., 1972; Klett, 1981; Sasano and Nakane, 1984) for each 5-min averaged profile that has large enough SNR above the cirrus cloud, thus allowing a molecular fit. For retrieving the extinction, however, the Klett method requires a predetermined value for the lidar ratio (LR), which is the ratio between the extinction and backscattering coefficients. Then, integrating the extinction coefficient from the cloud base to cloud top, the cirrus cloud optical depth is obtained ($\tau_{Klett}^{cirrus}$). Following Chen et al.

(2002), we estimated the value of LR for every cloud by iterating over the values of LR and comparing the values of $\tau_{Klett}^{cirrus}$ with the independent value of the cirrus optical depth obtained from the transmittance method described above ($\tau^{cirrus}$). The cirrus lidar ratio is the one that minimizes the residue: $R(S) = \left(\tau_{Klett}^{cirrus} - \tau^{cirrus}\right)^2$.

The Klett method assumes single scattering, but eventually the received photons could have been scattered by other particles several times before reaching the telescope. This effect, named multiple scattering, increases the laser transmittance and decreases the real extinction coefficient values. Thus, a correction is needed in our calculation of cirrus optical depth. As explained by Chen et al. (2002), for thin clouds it is possible to neglect the multiple scattering effect, nevertheless, we used their proposed correction for all cirrus clouds detected:

$\eta = \frac{\tau^{cirrus}}{e^{\tau^{cirrus}}-1};$    $\tau_{corrected}^{cirrus} = \frac{\tau^{cirrus}}{\eta}$  (4)

[Figure]

[Figure]

[Figure]

*[handwritten: ℔ = new paragraph]*

*[handwritten: what is the  absolute rate of cloud occurence?]*

## 3. Results and discussion.

### 3.1. Frequency of cirrus cloud occurrence.

*[handwritten margin: These are relative numbers]*

A total of 13,946 lidar profiles were measured with the presence of cirrus clouds, representing a frequency of occurrence of 67 % of the total number of profiles with good SNR. Figure 2 shows the monthly frequency of occurrence of cirrus clouds in central Amazônia from July 2011 to June 2012,

There is a well-defined annual cycle, with maximum values during the months of November,

December and March, reaching approximately 85 %, and minimum value in August during the dry season, but with frequencies no lower than a rather high 50 %. In tropical regions, the main mechanisms of cirrus clouds formation are deep convection, large-scale lifting of moist layers, orographic lifting over mountain slopes and "cold trapping" near the tropopause (Sassen et al, 2002). Deep convective clouds generate cirrus clouds while winds in the upper troposphere removes ice crystals of the top of the large convective column, generating the anvil cloud. This cloud remains even after the deep convection cloud dissipation and persist from 0.5 to 3 days (Seifert et al, 2007).

However, it should be noted that small local topographic effects around Manaus can  the occurrence and intensity of deep convection (Fitzjarrald et al. 2008; Adams et al. 2015). The frequency of deep convection during the rainy season is higher than the dry season, related with the seasonal change of the (ITCZ). The boxplot in Figure 2 show the variability of the daily frequency of occurrence for each month. There is a high dispersion of the daily frequencies, maximum dispersion in

August and lowest in November. The monthly cirrus cloud frequency follows the same seasonal pattern as the accumulated precipitation (Figure 2, green line), maximums during the wet months and minimums during dry months. For that reason, we divided the study period in wet (January, February, March and

April), dry (June, July, August and September) and transition (May, October, November and December)

periods, based on the accumulated precipitation in each month. The average monthly precipitation in each season during the observation period was 314 mm, 114 mm and 206 mm respectively.

*[handwritten margin: you just said this]*

The mean wind field on the typical cirrus cloud occurrence altitude (200 hPa) and the precipitation spatial distribution during the dry and wet months from July 2011 to June 2012 are shown in the Figure 3.

During wet months (Figure 3, lower panel), the site is inside the South American Monsoon region with great deep convection activity and associated rain ranging from 8 to 14 mm/day on average. Winds at

200 hPa blow from the southeast with about 20 m/s thus allowing the advection of cirrus clouds produced over other parts of the South Atlantic Convergence Zone. As the tropical cirrus can be transported by advection thousands of kilometers (Fortuin et al., 2007), we speculate that during the wet period, the cirrus clouds observed in central Amazonia are a mixture of locally produced and clouds transported by advection from other regions. During the dry period, the  convective activity moved to the north over

Colombia and Venezuela and the 200 hPa circulation is reversed. Hence, we speculate based on the high level circulation and precipitation, that a great contribution to the cirrus clouds observed during the dry months is the advection from the other regions.

*[handwritten: ☆ This is almost implied this. The lidar is attenuated otherwise. You are always looking at transport.]*

[Figure]

The diurnal cycle of the frequency of cirrus clouds is shown in the Figure 4 for the overall period and different seasons. All curves exhibit a similar pattern with minimum frequency occurrence values around

10 and 14 LT hours. Maximum values are found between 17 and 18 LT, in late afternoon, when values are slightly higher than in the morning. This diurnal variation of cirrus cloud occurrence follows the diurnal cycle of convection and precipitation documented in the literature (e.g. Machado et al., 2002;

Silva et al., 2011, Adams et al. 2013). Figure 4 shows also the diurnal cycle of precipitation for wet and dry months during the study period, averaged over an area of 2° x 2° centered on the experimental site.

The maximum of the cycle occurs between 13 and 18 LT, both in dry and wet months, similar to Adams et al. (2013). The occurrence of the maximum precipitation in the afternoon coincides with the increase in the cirrus frequency in all seasons.

A larger difference between the maximum and minimum values of the cirrus frequency for the dry months is visible in Figure 4. This can be understood by observing the maximum precipitation rates during this period, six times lower than those of the wet months, and the upper level circulation (Figure 4)

that indicates the long range advection. When the frequency of deep convection is greater, close to the site, the cirrus clouds are long-lived and more evenly distributed during the day, which does not occur during the dry months.

**3.2. Geometrical, optical and microphysical properties of cirrus clouds.**

Table 2 shows the statistics of the properties of cirrus clouds during the study year and different seasons.

The overall mean values for the cloud base altitude is 12.7 ± 2.3 km, cloud top is 14.4 ± 2.0 km and geometrical thickness is 1.7 ± 1.5 km. The mean value of the cloud maximum backscattering altitude is

13.2 ± 2.3 km, where the mean temperature is −58 ± 17 °C. The differences between the mean values of the geometrical properties in different seasons are statistically significant. The frequency of occurrence of all cirrus clouds throughout the year was 67 % of the time measurements with good SNR. The seasonal behavior discussed previously is also evident, with higher values of frequency of occurrence during the wet months (82 %) and lower during dry months (55 %). The mean values of the geometrical characteristics are similar, only the thickness is slightly different with 1.9 km (1.6 km) for wet (dry)

months. Mean COD values are 0.46 and 0.27 for the wet and dry months, respectively. Although the similarities between the mean values of the characteristics of all cirrus clouds during both seasons there are statistically significant differences between the mean values of the geometrical properties in the different seasons.

Our mean values are similar to those reported by Seifert et al. (2007) in the Maldives (4.1 °N, 73.3 °E):

11.9 ± 1.6 km (base), 13.7 ± 1.4 km (top), 1.8 ± 1.0 km (thickness), 12.8 ± 1.4 km (max. backscatter) and

−58 ± 11 °C (temperature at max. backscatter). Reports from subtropical regions also show similar values. Cadet et al. (2003) report for the Reunion Island (21°S, 55°E) cirrus cloud base and top altitudes of 11 km and 14 km, respectively. Antuña and Barja (2006) report to subtropical experimental site (21.4°

N, 77.9° W) cirrus cloud base and top altitudes of 11.63 km and 13.77 km, respectively. On the other hand, Sassen and Campbell (2001) show mean values for midlatitude cirrus cloud base/top of 8.79

km/11.2 km, lower as expected than tropical cirrus and an average geometrical thickness of 1.81 km.

Some cirrus clouds characteristics reported around the globe are shown in Table 1 for comparison. What

stands out is that our measurements over the Amazon show high frequency of occurrence of subvisible cirrus clouds similar or higher than previously reported from ground-based measurements in the Tropics.

The geometrical characteristics of the detected cirrus clouds were examined by means of normalized histograms. Figure 5 shows the results for cloud base and top height, thickness and the corresponding optical depth.

Histograms for the wet and dry months reveal differences. The cirrus clouds top altitude distribution (figure 5b), for instance, shows two peaks in the wet months, one centered in 14.25 km and second centered in 17.75 km, with a local minimum centered in 15.25 km and 16.25 km. On the other hand, for dry months, there is only one peak centered at 15.75 km. The local minimum during wet months occurs at

15.25 km, where a higher value near to maximum is found during dry months.

For the cloud base (figure 5a), the maximum frequency is in an interval of altitudes around 12.25 km.

Similar value of frequency is found in other peak centered at 14.75 km, with local minimum in 14.25 km.

There is a local maximum centered in 16.25 km during wet months. For the dry months, this last peak disappears, but the other two peaks remains with higher frequency values and with local minimum in

13.75 km.  These results suggest different cirrus types with different origins: cirrus formed directly by anvil outflows from cumulonimbus clouds through local convection; in situ formation from slow large scale air ascent; or possible advection from other convective regions. Comstock et al. (2002) proposed two different types of cirrus clouds at Nauru Island in the tropical western Pacific with oceanic conditions: one type (laminar thin cirrus) with cloud base altitude above 15 km and the other (geometrically thicker and more structured cirrus) with base altitude below this value, with different characteristics. Liu and Zipser (2005) used TRMM Precipitation Radar (PR) dataset to trace the deep convection and precipitation throughout the tropical zone, including oceans and continents. The authors showed that only 1.38 % and 0.1% of tropical convective systems, and consequently their generated cirrus clouds reached 14 km and 16.8 km of altitude, respectively. Hence, they suggested that those clouds with bases about 14 km are the thick anvil type cirrus, and the higher, thin cirrus have their bases above this altitude during the entire study period.

Considering these previous results, we suggest that the highest peaks in wet months and the single peak in dry months in cloud base and top histograms have the contribution of cirrus clouds formed far from the site and were transported from large distances. The clouds generated by convective systems can persist in the atmosphere from hours to days if they are slowly lifted (Ackerman et al., 1988; Seifert et al., 2007).

Thus, these clouds that ascended and were horizontally transported by long distances are, in general, optically and geometrically thinner and found in upper troposphere and tropical tropopause layer. Our results indeed indicate that these clouds are optically thin. This could be also the reason for which the geometrical thicknesses and optical depth are lower in the dry months see Figure 5 c and d, respectively.

We can see that the distribution of the geometrical thickness below 2 km and optical depth below 0.1 in dry months is above the distribution for the wet months.

From the cloud base altitude histogram (Figure 5a), one can note the high values for the frequency of occurrence of cloud base heights between 8.5 km and 9.5 km during wet season. This peak is the second most frequent after the principal one centered at 12.25 km and 14.75 km. This secondary peak is a result of using the cloud-base temperature of −25 C as a criterion for defining cirrus clouds. For this altitude,

there is possibly a fraction of mixed phase clouds that are counted inadvertently. The most reliable way of identifying the cloud phase is by measuring the depolarization caused by backscattered light, not available in the present study.

Figure 5d shows the normalized histogram of the cirrus clouds optical depth (COD) for the studied period and just the wet and dry seasons. In this case, the apparent COD values (explained in section 2) were excluded. This histogram shows how the frequency decrease with increases in COD. Moreover, during dry months, the cirrus clouds are optically thinner than during wet months.

A more in-depth analysis of the vertical distribution of cirrus clouds reveals features of different cirrus types, and its relation with COD becomes apparent. Figure 6 shows two-dimensional histograms of cloud optical depth and cirrus cloud top (upper row) and cloud base (lower row) for the wet months (left column) and dry months (right column). During the wet months, there is more dispersion of the values than in the dry months, which we speculate might be associated with a larger variability in the outflow altitude from deep convective clouds. The cloud-base distributions (Figure 6c, d) clearly show that the higher values of COD correspond to lower cloud base, whereas the lower values correspond to higher cloud base. The almost linear decrease is steeper for wet months (Figure 6c) than dry months (Figure 6d).

Hence, cirrus with the same cloud base altitude are more optically thick during that period. There are two maxima during the dry months, suggesting two types of cirrus clouds, those with bases below and those with bases above 13.75 km. The low altitude type has a cloud base at about 12.25 km and COD 0.20, while the high altitude ones, have a base at 14.75 km and subvisible optical depths of 0.01. During wet season, the two groups are much less pronounced and have higher COD. These cirrus clouds types were previously reported for the tropical region by Comstock et al. (2002) and Pace et al. (2003). However, the altitude that separates these two types of clouds over the Amazon (13.75 km) is lower than that reported over Nauru Island in the tropical western Pacific (15 km) by Comstock et al. (2002) and over Mahé Island in the tropical Indian Ocean (14.50 km) by Pace et al. (2003). Moreover, we also identified another group of subvisible clouds with very high cloud base (16.25 km) during the wet months, which is likely above the tropopause.

In the case of cloud top, the relation with COD is not clear. During the dry months (Figure 6b), almost all cirrus clouds tops, regardless of their COD, are found around 16.75 km. During the wet months, the cloud tops are spread from 13 km to 16 km, but all COD values occur at all altitudes. To investigate the role of the tropopause capping on the cirrus vertical development, its altitude was calculated from the ERA

Interim dataset (see section 2). The tropopause mean altitudes during the wet, transition and dry periods are 16.5 ± 0.2 km, 16.3 ± 0.3 and 15.9 ± 0.4, respectively.

Figure 7 shows the distribution of the distance from the cloud top and bottom to the tropopause. About

7 % (22 %) of the detected cirrus clouds have cloud base (top) above the tropopause during the wet season, and 6 % (17 %) during the dry season. Most of the cirrus clouds tops are found right below the tropopause inversion (see figure S.3a and S.3b), except during the wet season when they are found from -

3 km to +0.5 km. The presence of the cirrus clouds in the tropical tropopause layer is the consequence of the deep and strong convection in the Amazonian region, reported previously by Liu and Zipser (2009).

Their vertical distribution can then be understood as following. During the wet season, the intensity of deep convection in central Amazonia (as measured with convective available potential energy, water

vapor convergence, cloud top temperatures) can vary (Machado et al., 2002; Adams et al., 2009, 2013,

2015). Moreover, the tropopause is higher during the wet season (figure S.3c). Hence the cloud tops can be found from 13 km to 18 km, and cloud bases from 9 km to 18 km (figure 6). During the dry season, deep convection is found primarily north of the equator (figure 3), hence the cirrus clouds measured at

Manaus are mostly those transported over long distances by the prevailing winds (figure 3). As the cirrus produced northward around the tropopause do not last long, as they cannot be adiabatically lifted (Jensen et al., 1996), they do not reach the measurement site and there is only one maximum near 15 km in the distribution of cloud tops. During these dry months over the Amazon, however, precipitation is still about

100 mm per month. Hence, there is a second type of cirrus clouds (Figure 7a and 6d), which those are produced nearby, and hence are lower and optically thicker.

The statistical characteristics of cirrus clouds above and below 14 km are shown in the Table 2. Mean values of the properties are different for these cloud types. Cirrus clouds above 14 km are geometrical and optically thinner than clouds below 14 km. There are statistically significant differences between the properties of these two cirrus clouds types and between seasons. Also, there is a seasonal behaviour of the of these cloud types. During wet months the cirrus clouds are higher and optical and geometrically thicker than during the dry months.

The classification of cirrus clouds following Sassen and Cho (1992) shows that 40.0 % of the cirrus clouds measured in our experimental site are subvisible ($\tau < 0.03$), 37.7 % are thin cirrus ($0.03 < \tau < 0.3$)

and 22.3 % are opaque cirrus ($\tau > 0.3$). Table 2 shows these values for each season. subvisible cirrus clouds have the highest (lower) fraction during dry (wet) months. Opaque clouds have the highest (lower)

fraction during wet (dry) months, which is expected as there is a dominance of newly generated clouds by deep convection columns. This large fraction of optically thin and subvisible cirrus clouds over the

Amazon present a challenge for using passive remote sensing from space, such as MODIS. As mentioned by Ackerman et al. (2010), thin cirrus clouds are difficult to detect because of insufficient contrast with the surface radiance. MODIS only detects cirrus with optical depth higher than 0.2 (Ackerman et al.,

2008). Therefore, the MODIS's cloud-mask does not include 71 % of cirrus clouds over the Amazon, and likewise, their estimation of aerosol optical depth might be contaminated with these thin cirrus. Aerosol optical depth measurements from AERONET can also be contaminated with thin cirrus clouds. Chew et al. (2011), for instance, estimated a contamination of about 0.034 to 0.060 in AERONET AOD in

Singapore, where the cirrus frequency of occurrence is about 34%. Therefore, in our region with much higher cirrus frequency, the AERONET AOD might be more contaminated. Exactly how much contamination from thin cirrus there might be in MODIS and AERONET aerosol products over the

Amazon will be the subject of a forthcoming study.

These different types of cirrus clouds measured in central Amazonia, with different formation mechanisms, optical depths and altitude range are expect to be composed of ice crystals of different shapes. One way to gain information is to compute the ratio of the backscatter to the total extinction, the so-called lidar-ratio. As explained in section 2, we are able to find the average lidar-ratio for the detected cirrus clouds using an interactive approach instead of explicitly calculating the extinction from the Raman signal, which would be available only during night-time. Figure 8 shows the histograms of lidar ratio values for cirrus clouds during dry and wet months. The cirrus clouds were divided in three categories,

following our previous discussion: those clouds with base above 14 km, top below 14 km and those with
top (base) above (below) 14 km. In all case, the most frequent lidar ratios are between 16 sr and 20 sr.
There are notable differences only for the distributions for higher clouds (base above 14 km) during dry
months, when we observed two types of cirrus (Figure 6). For dry months, there is a large frequency of
occurrence of cirrus clouds with lidar ratios around 40 sr. According to the study of Sassen et al. (1989),
cirrus clouds composed of thick plates, long columns and thin plates would have lidar ratio values around
11.6 sr, 26.3 sr and 38.5 sr, respectively. Hence, during wet months there is the predominant mixture of
thick plates and long columns for all clouds. During dry months, the cirrus clouds that are entirely above
14 km have an important contribution of thin plates. These are long-range transported cirrus, thus the
aged ice crystals, will tend to become thinner during the transport.
The mean value of 20.2 ± 7.0 sr is obtained for the whole period and varying less than 1.5 sr for different
season months. Pace et al., (2003) showed a distribution to the inverse value of lidar ratio similar to that
presented here. They found a mean value of lidar ratio of 19.6 sr for the tropical site of Mahé, Seychelles.
Seifert et al.(2007), also for tropical regions report values near to 32 sr. Platt and Diley, (1984) reported
the value of 18.2 sr with an error of 20%. The value of the lidar ratio may vary greatly depending on the
altitude and composition of cirrus clouds (Goldfarb et al., 2011). For the other latitudes, there are
differences between the lidar ratio values examples given in Table 1.
After the analysis of the properties of the cirrus clouds, it is interesting to examine the behavior of the
variable with the temperature. Figure 9 show the dependence of the geometrical thickness, optical depth
and lidar ratios with the cirrus clouds temperature. The plots show temperature uniform intervals of
2.5 °C, and the variables with their mean and standard deviation for each corresponding interval. The
upper and middle panels contain the dependence of the geometrical thickness and optical depth with
cloud base temperature, respectively. We can see both variables increase at higher temperatures. Values
nearly to 3 km of geometrical thickness and 0.9 optical depth correspond to a temperature of −25 °C,
decreasing monotonically for lower temperatures in both month's periods. Similar results are reported by
Hoareau et al. (2013) and Seifert et al. (2007).
The Lower panel in Figure 9 shows the dependence between lidar ratio with mid-level cloud temperature. A
slight increase in the lidar ratio values from 15 sr to 24 sr when the temperature decrease up to −70 °C is
showed for dry period. During the wet period, the lidar ratio values are between 15 sr and 20 sr in all
temperature intervals. Seifert et al. (2007) and Pace et al. (2003) both show the same temperature
dependence of the lidar ratio, but with different mean values of lidar ratio. This behavior is an indication
of little variation in the microphysical characteristics of observed clouds. Nevertheless, for the dry period,
the lidar ratio grows when temperatures are below −75 °C. These temperature intervals correspond to the
clouds above 14 km discussed previously. These clouds above 14 km have different ice crystals shapes
concluded from the analysis of the right panel from Figure 9.

**4. Conclusions.**

The ACONVEX site started in 2011 with the goal of continuously monitoring climate relevant cloud
properties in central Amazonia. The ground based lidar measurements from July 2011 to June 2012 were
used to investigate the geometrical and optical properties of cirrus clouds in the region. An algorithm was

*[Handwritten margin notes:]*
*How did you derive the uncertainty. Standard deviation?*

*See Garnier et al 2015 AMT*

*why mid-level* *you're likely attenuated*

[Figure]

[Figure]

*Relative greeny when you had to divert SNR*

developed to search through this dataset with high vertical and temporal resolution and to automatically find the clouds, calculate the particle backscatter, and derive the optical depth and lidar-ratio. The frequency of occurrence during the observation period was 67 %, which is higher than all previous reports in the literature for other tropical regions. This frequency reached 82 % during the wet months (January,

February, March and April), but decreased to 55 % during the dry months (June, July, August, and

September). The analysis of high-level circulation and precipitation during the dry months indicate that advection from the northern regions is likely the main source of these cirrus. Whilst during the wet period, there was a mixture of locally produced and advected clouds. However, the diurnal cycle of the frequency of cirrus clouds showed a minimum around 12h LT and maximum around 18h LT, following the diurnal cycle of the precipitation for both seasons.    *SNR?*

The geometrical, optical and microphysical characteristics of cirrus clouds measured in the present study were consistent and in agreement with other reports from tropical regions. The mean values were

$12.7 \pm 2.3$ km (base), $14.4 \pm 2.0$ km (top), $1.7 \pm 1.5$ km (thick), 0.36 (optical depth) and 20.2 sr (lidar ratio). With the exception of the optical depth and lidar ratio, these mean values are similar to those found during the wet, transition and dry periods. Cirrus clouds were found at temperatures up to $-90$ °C and 7 %

of the cirrus were above the tropopause level or in the tropical tropopause layer. The role of these clouds in wetting or drying the stratosphere was left for another study.

By simultaneously analyzing cloud altitude and COD, it was found that cirrus clouds during the dry months are optically thinner and lower in altitude than those during the wet period. Moreover, the higher values of COD correspond to lower cloud base, whereas the lowest values, to higher cloud base. The almost linear decrease is steeper for wet months than dry months, hence, cirrus with the same cloud base altitude are more optically thick during wet season. The statistical distribution of altitude and COD

suggested the presence of two cloud types as expected. The first is located above 14 km with COD ~

0.02, and the second type at lower altitudes with COD ~ 0.2. A third type, not previously reported, was identified during the wet season, between 16 and 18 km with COD ~ 0.005. Cirrus clouds above 14 km were geometrically and optically thinner than those below, but have higher lidar ratios.

For the first time, the lidar ratio of cirrus clouds was obtained for this region. The mean lidar ratio was

$20.2 \pm 7.0$ sr, indicating a mixture of thick plates and long columns ice crystals, in agreement with other reports from the tropical regions. The statistical distribution of lidar ratios measured in the different seasons is the same, and they also do no vary with the temperature (altitude) of the cirrus clouds, indicating that these clouds are well mixed in the vertical. It was observed, however, that the distribution of the lidar ratio for clouds above 14 km during dry months shows a secondary peak around 40 sr, suggesting a different crystal shape like thin plates. From all cirrus clouds observed, 40 % were classified as subvisible (COD < 0.03), 38 % were as thin (0.03 < COD < 0.3) and 22 % as opaque (COD > 0.3).

During the dry months, the suvisible cirrus clouds reached a maximum of 46 %, while opaque cirrus has their maximum during wet months. These values are characteristic for our region and slightly different from measurements in other tropical regions. The central Amazon has a high frequency of cirrus clouds in general, and a large fraction of subvisible cirrus clouds. Therefore, the aerosol optical depth determined by sun-photometers and satellite based sensor in this region might be contaminated with the COD of these

thin clouds. Future work must be conducted in order to evaluate how large this contamination might be
over the Amazon.

**5. Acknowledgements**

We thank the researcher David K Adams from UNAM for reviews and valuable comments to improve the
paper content. We acknowledge the financial support from CAPES project A016_2013 on the program
Science without Frontiers, FAPESP Research Program on Global Climate Change under research grants
2008/58100-1, 2009/15235-8, 2012/16100-1, 2013/50510-5, and 2013/05014-0. Maintenance and
operation of the instruments at the experimental site would not have been possible without the
institutional support from EMBRAPA. We thank INPA, The Brazilian Institute for Research in Amazonia
and the LBA Central office for logistical support. Special thanks to Marcelo Rossi, Victor Souza and
Jocivaldo Souza at Embrapa, and to Ruth Araujo, Roberta Souza, Bruno Takeshi and Glauber Cirino from
LBA.

[Figure]

Tables:

Table 1. Summary of some recent cirrus clouds studies based on at least a few months of ground-based lidar observations in the tropics and mid-latitudes. The first columns show the period of study and laser wavelength (nm) for each site location, for which more than one study might be available. The cirrus characteristics are those reported by the different authors, which might include: base and top height (km), thickness (km), base and top temperature (°C), frequency of occurrence (%) and lidar-ratio (sr).

| Measurement site | Location | Period of study | Wave length [nm] | Height [km] Base | Height [km] Top | Height [km] Thick. | Average values Temp. [°C] Base | Average values Temp. [°C] Top | Frequency [%] SVC | Frequency [%] Thin | LR [sr] | |
|---|---|---|---|---|---|---|---|---|---|---|---|---|
| Salt Lake City, Utah, USA | 40.8°N, 111.8°W | 1986 to 1996 | 694 | 8.8 | 11.2 | 1.8 | -34.4 | -53.9 | 50 | - | | Sassen and Campbel (2001) |
| Haute Prov., France | 43.9°N, 5.7°E | 1997 to 2007 | 532/1 064 | 9.3 | 10.7 | 1.4 | | | 38 | | 18.2 | Goldfarb et al. (2001) Hoareau et al. (2013) |
| Thessaloniki, Greece | 40.6°N, 22.9°E | 2000 to 2006 | 355/5 32 | 8.6 | 11.7 | 2.7 | -38 | -65 | | 57 | 30 | Giannakaki et al. (2007) |
| Seoul, South Korea | 37°N, 127°E | 2006 to 2009 | 532/1 064 | 8.8 | 10.6 | | | | | | 20 | Kim et al. (2009) |
| Buenos Aires, Argentina | 34.6 °S, 58.5 °W | 2001 to 2005 | 532 | 9.6 | 11.8 | 2.4 | | -64.5 | | | | Lakkis et al.(2008) |
| Reunion Island | 21°S, 55°E | 1996 to 2001 | 532 | 11 | 14 | | | | 65 | | 18.3 | Cadet et al. (2003) |
| Camagüey, Cuba | 21.4° N, 77.9° W | 1993 to 1998 | 532 | 11.6 | 13.8 | | | | 25 | | 10 | Antuña and Barja, (2006) |
| Gadanki, India | 13.5 N, 79.2 E | 1998 to 2013 | 532 | 13.0 | 15.3 | 2.3 | | -65 | 52 | 36 | 25 | Pandit et al., (2015) |
| Hulule. Maldives | 4.1°N, 73.3°E | 1999, 2000 | 532 | 11.9 | 13.7 | 1.8 | -50 | -65 | 15 | 49 | 32 | Seifert et al. (2007) |
| Mahe´, Seychelles | 4.4 °S, 55.3 °E | Feb-Mar 1999 | 532 | | | 0.2-2.0 | | | | | 19 | Pace et al. (2003) |
| Nauru Island | 0.5 °S, 166.9 °E | Apr-Nov 1999 | 532 | -14 | -16 | | | | | | | Comstock et al. (2002) |

[Figure]

Table 2. Mean cirrus cloud properties and standard deviation in parenthesis for all cirrus clouds, for cirrus clouds above and below 14 km and cirrus clouds with base below and top above 14 km. These cloud properties are also informed to total time of observation, wet, transition and dry seasons

*Relative?*

| *All Cirrus Clouds* | Total | Wet | Transition | Dry |
|---|---|---|---|---|
| Frequency of Occurrence [%] | 67.2 | 82.2 | 79.4 | 55.5 |
| Base Altitude [km] | 12.7 (2.3) | 12.7 (2.6) | 12.5 (2.4) | 12.8 (2.1) |
| Top Altitude [km] | 14.4 (2.0) | 14.5 (2.2) | 14.3 (2.2) | 14.4 (1.8) |
| Thickness [km] | 1.7 (1.5) | 1.9 (1.6) | 1.8 (1.4) | 1.6 (1.4) |
| Cloud Optical Depth | 0.36 (1.20) | 0.46 (1.49) | 0.38 (1.22) | 0.27 (0.90) |
| Max Backscatter Altitude [km] | 13.2 (2.3) | 13.2 (2.5) | 13.0 (2.4) | 13.3 (2.0) |
| Temperature Max. Back. Alt. [°C] | -58.1 (16.9) | -58.0 (18.2) | -56.3 (18.6) | -59.1 (14.8) |
| Lidar Ratio [sr] | 20.2 (7.0) | 18.5 (6.5) | 19.4 (6.3) | 21.3 (7.2) |
| Subvisual Cirrus [%] | 40.0 | 36.4 | 33.9 | 46.0 |
| Thin Cirrus [%] | 37.7 | 37.0 | 41.7 | 36.2 |
| Opaque Cirrus [%] | 22.3 | 26.6 | 24.4 | 17.7 |
| | | | | |
| *Cirrus Clouds with Base > 14 km* | | | | |
| Fraction of all cirrus [%] | 31.8 | 32.0 | 28.7 | 33.2 |
| Base Altitude [km] | 15.4 (1.0) | 15.7 (1.1) | 15.5 (1.0) | 15.2 (0.8) |
| Top Altitude [km] | 16.2 (0.9) | 16.6 (0.9) | 16.4 (0.9) | 15.9 (0.8) |
| Thickness [km] | 0.8 (0.6) | 1.0 (0.8) | 1.0 (0.7) | 0.7 (0.4) |
| Cloud Optical Depth | 0.03 (0.06) | 0.04 (0.07) | 0.04 (0.07) | 0.02 (0.04) |
| Lidar Ratio [sr] | 22.9 (9.5) | 20.2 (8.7) | 21.7 (9.0) | 25.5 (9.7) |
| | | | | |
| *Cirrus Clouds with Top < 14km* | | | | |
| Fraction of all cirrus [%] | 38.0 | 38.8 | 41.9 | 35.3 |
| Base Altitude [km] | 10.8 (1.5) | 10.7 (1.5) | 10.5 (1.5) | 11.2 (1.3) |
| Top Altitude [km] | 12.3 (1.4) | 12.3 (1.4) | 12.1 (1.3) | 12.5 (1.3) |
| Thickness [km] | 1.5 (1.2) | 1.6 (1.2) | 1.6 (1.2) | 1.3 (1.0) |
| Cloud Optical Depth | 0.50 (1.70) | 0.67 (2.09) | 0.55 (1.64) | 0.32 (1.28) |
| Lidar Ratio [sr] | 20.0 (6.7) | 19.3 (7.7) | 18.9 (6.2) | 20.5 (6.4) |
| | | | | |
| *Cirrus Clouds with Base< 14km and Top>14 km* | | | | |
| Fraction of all cirrus [%] | 30.2 | 29.2 | 29.3 | 31.5 |
| Base Altitude [km] | 12.2 (1.4) | 11.2 (1.5) | 12.3 (1.3) | 12.3 (1.3) |
| Top Altitude [km] | 15.2 (0.9) | 15.3 (1.0) | 15.3 (0.9) | 15.0 (0.7) |
| Thickness [km] | 3.0 (1.7) | 3.3 (1.9) | 2.9 (1.5) | 2.8 (1.5) |
| Cloud Optical Depth | 0.55 (0.99) | 0.70 (1.19) | 0.49 (1.01) | 0.47 (0.78) |
| Lidar Ratio [sr] | 19.7 (6.3) | 18.0 (5.5) | 19.1 (5.6) | 20.9 (6.7) |

[Figure]

[Figure]

Figures:

[Figure]

Figure 1. Location of the experimental site (2.89°S 59.97°W) is shown, 30 km upwind from downtown Manaus-AM, Brazil.

[Figure]

Figure 2. Monthly frequency of occurrence of cirrus clouds from July 2011 to June 2012 (blue line). Red dashes (black x) in the boxplots are the median (mean) of the daily frequency in each month. The edges of the boxes are the 25th and 75th percentiles, and the whiskers extend to the most extreme daily values. Accumulated rainfall is shown in green on the right axis. Data is from TRMM 3B42 version 7.

[Figure]

[Figure]

[Figure]

Figure 3. Mean precipitation (colors, mm/day) from the TRMM 3B42 version 7 and mean wind field (vectors, m/s) at 200 hPa from ECMWF ERA Interim reanalysis are shown for the average dry months (JJAS) and wet months (JFMA), between July/2011 and June/2012. The experimental site location is marked with a black dot.

[Figure]

Figure 4. Diel cycles of the hourly frequency of occurrence of cirrus clouds are shown for the annual, wet, transition and dry periods. Mean precipitation rate (mm/h) over an area of 2°× 2°centered in the site is shown in dashed lines for the Dry (+) and wet (□) periods. Data is from TRMM version 7.

[Figure]

[Figure]

[Figure]

Figure 5. Panels show the normalized histograms of (a) cirrus cloud base, (b) top, (c) geometrical thickness, and (d) optical depth, for the overall period (black), wet season (JFMA, red) and dry season (JJAS, blue).

[Figure]

Figure 6. Two-dimensional histograms of cirrus cloud top (top) and cloud base (bottom) with optical depth during the wet (left) and dry (right) months are shown.

[Figure]

[Figure]

[Figure]

Figure 7. Normalized histograms of the distance of the tropopause to the cirrus base (left) and top (right) are shown for overall period (black) and each season (colors). Negative values mean that clouds are below tropopause. The average tropopause altitude was $16.2\pm0.4$ km.

[Figure]

Figure 8. Normalized histograms of the lidar ratio for the wet (left) and dry (right) months are shown for all clouds (black) and clouds with base above 14 km (red), top below 14 km (blue) and cloud with top (base) above (below) 14 km (green).

[Figure]

[Figure]

Figure 9. Dependence of the geometrical thickness, cloud optical depth and lidar ratio with temperature. Temperature are shown in 2.5 °C intervals and the other variables with their mean and standard deviation in each temperature interval.

---

## Author Comment (AC1) · 28 Sep 2016

**Answer to Anonymous referee #1**

We would like to thank anonymous referee #1 for the careful reading of the manuscript and the useful comments and recommendations. We accepted all suggestions. Below you will find our replies and short descriptions of the changes we've made in the text. Referee comments are in red and start with **"R:"** and our replies are in black and start with **"A:".** Original manuscript text is shown in **blue**, with new text highlighted in yellow.

We would like to warn referee #1, however that we are still working on our data analysis to accommodate all suggestions from both referees and hence the final numbers might still change. Referee #2 urged us to change our definition of cirrus clouds according to (Campbell et al., 2015). More important, however, is the request by Referee #1 to do a full multiple-scattering correction, instead of the approximate correction we have now. The new manuscript will be uploaded as soon as we finish all these changes.

R1: The authors should be less aggressive when speculating on their statistics and reading into differences that are not statistically significant.

A: We agree with the recommendation. We have reviewed our statistical analysis; particularly we have calculated the statistical significance before making strong statements.

R2: Much of the paper is spent discussing season and diurnal differences in the cirrus statistics. However, little attenuation is paid to whether these differences are actually statistically significant. Some figures/tables give the standard deviations, but little discussion of them is given in the text leaving the reader to determine significance themselves. In Table 2, it appears that non of the statistics differ significantly: i.e. one cannot say that the cirrus differ at all from season to season. Similarly in Figure 2, the box plot reveals that there is no significant seasonal cycle in frequency of occurrence either. In other figures where histograms are given, a statistical test should be applied to ensure difference among distributions are statistical significant before they are discussed. It is only appropriate/worthwhile to discuss differences that are statically significant.

A: We agree that it only makes sense to discuss differences that are statistically significant, and we have indeed tried to do that. It seems, however, that the captions and text discussion about figure 2 and table 2 were not well explained, leading the reviewer to misinterpret their content.

Table 2 shows the values of the mean and of the standard deviation of the sample (for individual 5-min observations). When comparing mean values, however, we should use the standard deviation of the mean, which is roughly the standard deviation of the sample divided by square-root(N). As shown in Figure S.1 (supplement), there are about 1500 good profiles per month, or ~6000 per season. However, as multiple layers are considered as different clouds (if spaced by more than 400m), the number of cloud layers detected in any season is about ~10000, hence the standard deviation of the mean values are about 1/100th of the values shown in brackets in table 2. That is why the differences of seasonal mean values are statistically significant for most parameters shown.

We agree, however, that we did not give enough information, and we apologize particularly for not having included the values of N for each column in Table 2. To resolve that issue, we moved the information from Fig S.1 into table 2, and we also included the number of cloud layers detected. We have also included an * (asterisk) to indicate when the difference of mean values between the dry and wet seasons is statistically significant with a 95% confidence level. The caption of Table 2 now reads:

Table 2. Mean cirrus cloud properties and standard deviation of the sample (in parenthesis) for all cirrus clouds, for cirrus clouds above and below 14 km and cirrus clouds with base below and top above 14 km are shown. Values are given to the total time of observation, as well as the wet, transition and dry seasons. Statistics are based on 5-min observations. The number of clouds layers detected was used to calculate the standard deviation of the mean, and then determine if the seasonal differences (wet-dry) are statistically significant to the 95% confidence level (indicated as *).

One important point to note is that table 2 gives statistical properties (mean and std) based on 5-min profiles, and hence the frequency of occurrence is a single number without an associated standard deviation. The box-plot in Figure 2, on the other hand, was calculated from daily averages as was discussed in the text and in the figure caption. For instance, the caption reads: *"..Red dashes (black x) in the boxplots are the median (mean) of the daily frequency in each month…"*. Hence, the box-plot indicates the day-to-day variability in

each month. For example, during March 2012, we never observed any single day with less than 52% or more than 95% frequency of occurrence. In fact, 50% of the daily averages are within the interval 79-89%. As noted before, to compare the mean values (in this case of the daily occurrences) one should use the standard deviation of the mean, not of the sample. And when comparing the whole distribution, one should also note that even if two distributions have the same mean values, it doesn't mean they are the same. In this sense, the boxes in Figure 2 show that Jul and Aug 2011 have the same distribution and mean values, while Jan and Jun 2012 have close medians but different distribution.

However, we agree that the text/captions/figures could be modified to make this point more clear. Hence, in Figure 2, we added the standard deviation of the mean as an error bar centered at the mean daily frequency (cross) and we created a secondary panel to show the precipitation. The caption of Fig. 2 now reads:

Figure 2. The upper panel shows the monthly frequency of occurrence of cirrus clouds from July 2011 to June 2012 (blue line) for all data in each month. The boxplot is for the daily frequencies and hence show the day-to-day variability. Red dash, black cross (x) and error-bar indicate the median, the mean and the standard deviation of the mean, for the daily frequencies in each month, respectively. The edges of the boxes are the 25th and 75th percentiles, and the whiskers extend to the most extreme daily values. Accumulated rainfall is shown in the lower panel based on data from TRMM 3B42 version 7.

The text around lines 207-210 was modified accordingly:

The boxplot in Figure 2a show the variability of the daily frequency of occurrence for each month. There is a high day-to-day variation (i.e. dispersion of the daily frequencies), which is maximum in May-Aug and lowest in Nov-Apr. The mean monthly cirrus cloud frequency follows the same seasonal pattern as the accumulated precipitation (Figure 2b), with values during the wet months higher by a statistical significant amount than those during dry months (notice the small standard deviation of the mean despite the high variability).

R3: I would also caution the authors against extrapolating too much from their relatively limited data. An example of this is using the lidar ratio to infer the ice crystal habit. The lidar ratio alone cannot be used to identify the ice crystal habit since it also depends on the particle orientation relative to the laser beam. In addition, theoretical studies of ice crystal phase functions vary wildly so there is no real consensus on what the lidar ratio even is for different ice crystals.

A: We thank the reviewer for pointing out this issue. After his/her comments, we revised the literature once again and found, as he/she mentioned, no consensus about what the LR should be for different crystal habits. We have hence removed any reference to actual crystal shapes. However, we kept the argument that the bimodal distribution of the LR for some seasons is an indicative of mixture of different shapes, although we can't tell which habit from our limited dataset.

R4: The authors note that this ground-based site is unique compared to others reported in previous work, yet rely heavily on previous work to explain their results. The paper would be greatly enhanced by making a more quantitative effort the explain their data. For example, instead of speculating on the sources of moisture for the cirrus in different seasons, a more convincing approach would be to run back trajectories to show the reader where the air came from.

A: We thank the reviewer for pointing this out. We note, however, that the first intent of our paper was do document the diurnal and seasonal cycles of geometrical and optical properties of cirrus clouds in the central Amazon. This is a region known for having an important link with climate, but also known for the lack of continuous long-term observations. This is the reason why most papers rely on satellite-based observations, which are mostly polar-orbit and hence, have low time resolution. We said that "our site is unique" because it is allowing us, finally, to perform long-term observations with high temporal resolution. The obvious drawback is that we don't have the spatial coverage of the satellites.

Although our initial intention wasn't the identification of the sources of the cirrus clouds, we agree that giving a more quantitative explanation would strengthen our paper. Following the referee's suggestion, we did back-trajectory analysis using Hysplit forced by GDAS winds (1deg resolution), starting every 6h from 14.5 km over the site during the dry season period. Each of the 480 back-trajectories were integrated for 7 days. Figure below shows the result of this analysis. In the top panel, we show the individual trajectories just for the 0:00 of each day and there are so many lines that clutter the plot. The lower panel shows the trajectory density, i.e., the number of trajectories in a point divided by the total number of trajectories (a number [0-1]). In this case we used a log-scale because the density will obviously be much higher closer to the trajectory

start point. The result is quite interesting as it reveals that many trajectories actually don't follow the average wind pattern (fig. 3 in the manuscript, top panel). On the other hand, many trajectories come from Colombia and Venezuela, exactly where precipitation from deep convection is found (also shown in fig. 3, top), and some even reach towards the ITCZ, far to the east. This comparison could be improved if we select only the trajectories starting at times when we detected a cirrus clouds (yet do be done).

[Figure]

Figure C.1 – Hysplit 7day backward trajectories starting 14.5km above the site every 6h for the four months of the dry season. It should be compared to the top panel of figure 3 in the manuscript.

Although this trajectory analysis (suggested by referee #1) indeed gives further evidence that our cirrus clouds originate from deep convection, it is not a quantitative evidence (as referee #1 wanted). Alternative ways of having a quantitatively analysis would be to run the back trajectories for each cloud layer detected and use GOES images to locate deep convective cells, and then calculate the distance between each trajectories and the surrounding precipitation (as a function of backward time). This is a huge effort and, we believe, deserves its own paper. Another possibility would be to do that, but just for one case study in each season (e.g. as Fourtin et al., 2007JGR). We are not sure, however, how representative and quantitative that could be.

If the reviewer has other suggestions, besides the back-trajectories already shown above, we would be glad to try. Or if the reviewer / editor think the plots above are already enough, we would be happy to include this discussion in the manuscript.

R5: Is there reason the authors don't use the nitrogen signal to retrieve extinction? Not doing so doesn't completely discount the data presented, but it does devalue it somewhat since this paper is just another in a long-line of elastic lidar cirrus studies.

A: Raman inelastic scattering has a cross-section of about 1/1000[th] of the elastic one. For our system, that weak signal is only discernible from the background during nighttime, thus not being appropriate for our study, which wanted to investigate the diurnal cycle. Moreover, the Raman method involves calculating the derivative of the signal, which gives rise to large uncertainties in cases of such low signal-to-noise ratios as typically found at cirrus altitudes. That noise could be reduced by vertical smoothing or by time averaging,

but our clouds are not homogenous in time in space, and the averaging could compromise the results.

We should also mention that our experience with simulated signals have shown us that we can obtain a very precise and accurate LR with Chen's method. We did this analysis of simulated cloudy lidar profiles to evaluate the accuracy of the transmittance method but did not include in the manuscript, or in the supplement (but see discussion following your next question). We believe that these simulations and the fact that Raman cannot be done during daytime justify our method of choice.

Finally, we also noticed that this argument was given only at the results section (lines 398-400). We have thus added the following paragraph in section 2.4, at line 181:

We use Chen et al. (2002) instead of the Raman method (Ansman et al., 2002) because our instrument can only detect the nitrogen Raman channel during nighttime. Moreover, the Raman results are very noisy even during nighttime and by analyzing simulated lidar profiles (supplement material) we found that a precise and accurate cirrus LR can be obtained with Chen's method.

R6: In addition, the transmission method is really only accurate for mid-range optical depths. Too thin and there isn't enough transmission signal to get a reliable optical depth. Too thick and there isn't enough molecular signal above the cloud. I encourage the authors to go beyond just checking the SNR above/below the cloud when doing the optical depth retrieval and to fully derive the uncertainty in the optical depth values they report. Figures 5 and 6 show optical depths down to 0.001, which I expect to be extremely uncertain when using the transmission method to retrieval optical depth.

A: We agree with the reviewer's point of view and, in fact, we have already calculated the uncertainty in all optical depth values that we have obtained. However, the plots and tables shown in the manuscript are always for averages over a huge amount of profiles, and hence we choose to report the standard deviation of the mean instead of the errors in individual retrievals.

We also agree that the COD uncertainty is very large if obtained for a single profile with COD = 0.001. In fact, we have done an extensive simulation study to validate the methods we use, which was not included in the manuscript. For COD = $10^{-3}$, the relative error in a single retrieval is 120% for S/N = 50 and 1150% for S/N = 3, both large but not enough to change the cirrus category (e.g. from sub-visual to thin). Moreover, averaging over N profiles reduces this uncertainty by a factor of square-root of N. In our study, we analyzed about 37k 5-min profiles, where 21k had S/N > 3 at 12km and in 14k of these we found a cirrus cloud. Thus, the error in the mean cloud optical depth reported in Table 1, or in the histograms in figures 5 and 6, is indeed much lower, typically below 20% even for S/N = 3.

We were planning to have a separate manuscript on AMT about the accuracy and precision of the transmission method for the retrieval of COD and LR from elastic lidars. However, as both referees have questioned about this, we believe that some of that needs to be included in the supplement material. We will consult the editor to see if he/she agrees with this approach.

R7: The treatment and discussion of multiple scattering could be improved. Although, not explicitly stated, I'm guessing the authors use Eq. (10) from Chen et al. (2002) where eta depends on the optical depth of the cloud layer. I'd would encourage against using this equation. Chen et al. provide no physical justification for this equation and the values for larger optical depths quickly approach the wide angle scattering limit of eta=0.5 which is unrealistic for the geometry of a ground based lidar. In addition, for optical depth greater than about 1.2, eta<0.5 which is unphysical. The authors should also keep in mind that the shorter wavelength of 355nm (compared to 532nm as is used in Chen et al. 2002 and many other studies) means much stronger forward scattering and therefore larger amounts of multiple scattering. Typical extinction biases could range from 5-30% and sometimes even larger (see Thorsen and Fu, JTECH 2015 Fig. 13). I would suggest the authors make clear to the reader that their optical depth may contain significant biases due to multiple scattering unless some type of explicit treatment of multiple scattering is performed.

A: We thank the referee for explaining the limitations of the correction proposed by Chen. To appropriately account for the multiple-scattering, we reviewed the work of Platt (1981) and Wandinger (1998) and finally decided to apply a full treatment following the model of Hogan (2008). Our preliminary results are indicating a change of the LR from ~19 sr to ~25 sr. That means that Chen's correction was indeed not valid for our case! Looking at table 1, where we compared our results with some available in the literature, it seems that

Goldfarb et al. (2001), Pace et al. (2003), Cadet et al. (2003), Hoareau et al. (2013), and Pandit et al. (2015) also did the same mistake as we did.

However, reprocessing all the dataset with this more complex algorithm for multiple-scattering is taking much longer than anticipated. Hence, as we mentioned in the beginning, an updated version of the manuscript will be uploaded only after finishing these calculations.

References:

Hogan, R. J., 2008: Fast lidar and radar multiple-scattering models: Part 1: Quasi-small-angle scattering using the photon variance-covariance method. J. Atmos. Sci., 65, 3621-3635.
Platt, C. M. R., 1981: Remote sounding of high clouds. III: Monte Carlo calculations of multiple scattered lidar returns. J. Atmos. Sci., 38, 156–167
Wandinger, U.: Multiple-scattering influence on extinction- and backscatter-coefficient measurements with Raman and high-spectral-resolution lidars, Appl. Optics, 37, 417–427, 1998.

---

## Author Comment (AC2) · 30 Sep 2016

**Answer to referee #2 James Campbell**

Dear J. Campbell:

Thanks a lot for your useful comments, technical notes, editing recommendations and questions. We appreciate it very much; it will definitely improve our manuscript. Below you will find our replies and short descriptions of the changes we've made in the text. Your comments are in red and start with **"R:"** and our replies are in black and start with **"A:"**. Original manuscript text is shown in **blue**, with new text highlighted in yellow.

We would like to warn you, however that we are still working on our data analysis to accommodate all suggestions from both referees and hence the final numbers might still change. You urged us to change our definition of cirrus clouds, but what is being more challenging is the request by Referee #1 to do a full multiple-scattering correction, instead of the approximate correction we have now. The new manuscript will be uploaded as soon as we finish all these changes.

R1: My primary scientific concern relates to the definition of "cirrus" clouds in the manuscript. (…) In a recent paper that I authored (Campbell et al. 2015), we went to significant length to demonstrate a practical and viable definition for cirrus clouds in autonomous long-term datasets like this one, and in particular for those that lack a polarized backscatter measurement. (…) This absolutely has to be revised. I recognize that this is a serious request, and I raise this point very respectful of the work that has been put into the manuscript, the statistics and the analysis. However, I question every number you have in here, again respectfully, because of such a simple and non-physical definition applied for discriminating these clouds.

I would respect if the authors were to disagree with our conclusions/recommendations in Campbell et al. (2015). But, in response, they'd better come up with a physically- based reason for doing so. Cloud top temperatures of -37°C makes physical sense for the class of clouds that we call cirrus. It's a practical and defendable threshold.

A: We agree with the reviewer that cirrus is a phenomenological classification based on surface visual observations. We also agree that such classification (and alternatives to make it more practical) has been debated in the literature. For instance, the -25 degC threshold that we applied has been used in previous papers (e.g. SEIFERT et al., 2007; GOLDFARB et al., 2011). However, we don't agree that this criterion allows our numbers to be, although respectfully, strongly questioned. In the tropics, our threshold corresponds to a minimum cloud-base altitude of about 8 km. Likewise, the -37 degC cloud-top threshold suggested by the reviewer corresponds to roughly 11 km. The cloud-top histogram in figure 5 shows that less than 6% of what we've called cirrus would not fit the reviewer's suggested criteria. Therefore, our numbers cannot be that far off from what would be obtained following Campbell et al (2015).

Nonetheless, a physically based definition such as proposed by Campbell et al (2015) is indeed preferred. We will use this definition in the new version of the manuscript. We are already reprocessing all our data.

R2: Its unclear what the authors are saying about the presence of SNR > 3 in the upper troposphere with respect to cloud observation. Do they mean clear-sky? Or, do they mean within particulate scattering layers?

A: We apologize for not having stated this clearly in the manuscript. What we meant is that the molecular lidar signal just below the cirrus cloud-base should have a SNR of at least 3, in a single bin of 7.5m (our raw resolution). For 30m bins, for instance, the SNR will increase to about 6, and one should remember that the molecular fitting involves many points, which further reduce the noise. For the typical cirrus optical depths, SNR>3 means that the laser was not attenuated and hence we will most likely still have good enough SNR above the cloud top, which is needed for the retrieval of the optical depth and lidar ratio by the transmittance method. We also have to evaluate the SNR in cirrus-free profiles in order to count all the profiles in which cirrus could have been detected if they were present. This is necessary to compute frequency of occurrence correctly.

The first paragraph of section 2.2 was modified as follows around lines 115-118:

A total of 36,597 5-minute profiles were analyzed and only 20,752 had a signal to noise ratio (SNR) higher than 3, for a single 7.5 m bin just below the cirrus base. Given the typical cirrus cloud optical depths, this threshold means we will also have a SNR at cloud top that is good enough for estimating the optical depth with the transmittance method (see section 2.4). Statistical tests with the transmittance method based on simulations for various SNR, COD and cloud thickness (not shown) were conducted to obtain the SNR threshold. The number of 5-min lidar profiles and number of profiles with good SNR during each month of the studied period were analyzed.

R3: Its unclear how the authors define the tropopause, and thus accommodate the potential for resolving the bottom/top of the tropopause transition layer, in Section 2. This hurts the discussion later on where context is necessary for understanding where the clouds are with respect to this boundary.

A: We apologize for not having stated this clearly. As we wrote in the manuscript, we are using the WMO definition (International Meteorological Vocabulary, 1966). In this technical document the Tropopause is defined as: "*the boundary between the troposphere and the stratosphere, where an abrupt change in lapse rate usually occurs. It is defined as the lowest level at which the lapse rate decreases to 2°C/km or less, provided that the average lapse rate between this level and all higher levels within 2 km does not exceed 2°C/km*".

One should be careful when applying this definition, however, as the number of vertical levels in the sounding, reanalysis or model data might be too coarse. To overcome this issue, we follow the methodology suggested by the National Meteorological Center (McCalla, 1981). The lapse-rate is assumed to vary linearly with pressure, and the exact altitude where $\Gamma=2°C/km$ (i.e. the tropopause) is found by linearly interpolating between the closest available pressure levels.

We modified section 2.2 around lines 130-147 to better explain the calculation of the tropopause altitude:

This dataset was used to obtain the mean high level winds, near to the cirrus clouds habits (200 hPa). The tropopause altitudes were obtained from vertical profiles over the site using the definition of the World Meteorological Organization (IMV WMO, 1966), i.e. "*the lowest level at which the lapse rate decreases to 2°C/km or less, provided that the average lapse rate between this level and all higher levels within 2 km does not exceed 2°C/km*". We further assumed the lapse rate to vary linearly with pressure (McCalla, 1981), and the exact altitude where $\Gamma=2°C/km$ (i.e. the tropopause) was found by linearly interpolating between the closest available pressure levels. A precipitation dataset for the same period was acquired from TRMM (Tropical Rainfall Measuring Mission) version 7 product 3B42 (Huffman et al., 2007) with 0.25° and 3 h of spatial and temporal resolution, respectively.

And we include these two references:

International meteorological vocabulary. WMO, No. 182. TP. 91. Geneva (Secretariat of the World Meteorological Organization) 1966. Pp. xvi, 276. Sw. fr. 40. Q.J.R. Meteorol. Soc., 93: 148. doi:10.1002/qj.49709339524

McCalla, C., 1981: Objective Determination of the Tropopause Using WMO Operational Definitions, Office Note 246, U.S. Department of Commerce, NOAA, NWS, NMC, 18pp, October 1981.

R4: Since the sample size is stated to relative to the ability to measure SNR > 3 in the upper troposphere, all of the samples appear to be relative occurrence frequencies and not absolute ones. This is HIGHLY confusing. There is no way that you're resolving an absolute cloud frequency of 67%. In a new paper that we have in Early Online Release in JAMC (Campbell et al. 2016), we show in a year's worth of MPLNET observations at Greenbelt, MD an absolute frequency near 16%, which owes to the attenuation of the beam from low-level clouds and undersampling of the upper troposphere. There are multiple places in the narrative where serious confusion arises and the speculative discussion becomes meaningless because of this confusion.

A: We do not agree with the referee in this point.

We know, based on satellite studies that the cirrus absolute frequency is much higher than the 16% found by the referee for Greenbelt. For instance, based on Calipso and CloudSat, Sassen et al., JGR 2001 show for US east coast a frequency about 25% (fig.1 of that paper). They also show a frequency about 50% at our site in the Amazon (3S, 60W). The only way one could have such a low frequency, as suggested by

the reviewer, is if one divides the total number of cirrus detected by the total number of possible observations (i.e. including low-level clouds, etc…). We argue, however, that such number would not have a physical meaning. It would just reflect your sampling issues.

On the other hand, it is very usual to report the cloud occurrence the way we do (e.g. Erika Kienast-Sjögren et al. 2016; Nazaryan et al., 2008; and references therein). There is a broad but valid assumption behind, which is justifying this approach. The lifetime of cirrus is much longer than for other clouds. At the same time, the presence of cirrus clouds in the sky is rather independent of low-level water clouds that can fully attenuate the laser beam. Hence, you can estimate the absolute cirrus frequency simply by dividing: the number of lidar profiles with a cirrus, by the number of profiles where you could have detected a cirrus cloud.

We agree this is not the true cirrus frequency, but it is the best estimate one can make. Besides, our cirrus frequencies are in agreement with values obtained from CALIPSO, if we consider the same time of the satellite overpass. See, for instance, the values reported by NAZARYAN et al. 2008 or SASSEN et al 2008 and compare with the values in our paper.

We want to give a very naïve example, and we do so very respectfully with the aim of making our point very clear. The approach we follow is the same as doing a pool to figure out who will win the next election for president. You take a small sample of 1000 people from a population of 150 million voters and you can still tell the outcome of the election (given that your sample was randomly selected). But please note that we don't have to worry about the "random selection" in the case of cirrus frequency because: 1) the presence of cirrus and low-level clouds are independent (and also independent from our sampling failures); and 2) we sampled 37k profiles of 5-min, which is 1/3 of the maximum possible number of profiles during 1 year.

But how do we know which profiles we could have detected a cirrus (if they were there)? We do that by looking at the SNR at the typical cirrus altitudes and knowing the efficiency of our algorithm as a function of the SNR. Based on the analysis of simulated profiles (GOUVEIA, 2014, Msc Thesis, U. of Sao Paulo), we know that our algorithm can detect 99% of cirrus clouds with COD > 0.005 if the SNR is at least 3 below cloud base. With this strategy, for example, profiles with low water clouds that kill the laser beam are not added in the denominator. The total number of profiles measured in each month and the number of profiles with good SNR was shown in figure S.1 in the supplement and discussed it in the text:

Lines 119-125
July, August and September, the driest months show the higher fraction of profiles with good SNR, while the wettest months have the lowest fraction of lidar profiles with good SNR (see figure S.1). The cloud fraction of low, optically thick clouds increases during this season, thereby attenuating the signal and reaching the cirrus clouds altitudes with a low SNR. The frequency was then defined as the ratio between the number of lidar profiles of 5 min with good SNR containing cirrus clouds and the total number of profiles with good SNR.

R5: Speaking of this issue, nothing is said of the work of Thorsen et al. (2011) and Protat et al. (2014) and undersampling issues relating to ground-based profiling, attenuation, and the relative cloud samples that we have to analyze. This is a serious weakness that leads to three other points of concern.

Now that the reviewer mentioned Thorsen and Protat, we believe to have understood his concerns. We probably did not explain very well how we count the profiles for our statistics and that might have lead to his confusion here and in the previous comment.

As we explained above, properly counting how many profiles you have in the denominator of your cirrus cloud fraction is the key point (think of the election pool). You cannot include attenuated profiles otherwise you will introduce a bias. You also cannot average over a long period of time by simply averaging your data, unless it is uniformly distributed.

All of these points are considered in our approach. For instance, how do we calculate the year average if we measured a different amount of days in each month (different sample sizes)? We just average the fractions of each month (weighted by the number of days), and the fraction in each month was calculated including in the denominator only the profiles for which you could have detected a cirrus. If our sampling

varied too much within a month, we could've broken it up into weeks. The same strategy is applied when we calculate the diurnal cycle. The cirrus fraction in a given hour-bin is the number of profiles with cirrus in that hour divided by the number of profiles for which we could have detected a cirrus in that hour. Therefore, we can still have a good estimative of the true diurnal cycle even if we have different sample sizes for each hour.

To our understanding, our approach (election pool) is the same as used by Thorsen et al JGR (2011) and Protat et al JAMC (2014), however, they've called it "conditional sampling". See, for instance, what Protat says in section 3 of his paper:

> "*Fortunately, conditional sampling (for instance excluding profiles where low-level obscuration occurs, as in Thorsen et al. 2011) can be carefully designed for sake of model and satellite product evaluation using data collected at the ground-based sites.*"

To make our approach more clear, we will modify section 2.2, removing the discussion about the SNR. That will be included in a new section called "sampling issues" where we will explain how we applied the conditional sampling of Thorsen (2011) and Protat (2014). That will be a summary of our replies to your comments R4 and R5.

R6: It is discussed that the lowest cloud observations occur around solar noon (10-12 LT). This leads me to believe that your instrument is suffering from issues with SNR from the bright background, even at 355 nm. Whereas it is introduced that this is potentially a real artifact, I see no reason to take such a claim at face value. As I cannot evaluate your algorithm or its performance, and with the practical understanding that you are willing to deal with cloud samples in the algorithm at an SNR as low as 3, I cannot help but conclude that you're dealing with sampling issues due to background noise.

We thank the reviewer for carefully looking at all details of our results. We should say, however, that we also have looked into this minimum around solar noon to be sure that it was not a problem with the solar background. Our conclusion is that it is real for the reasons below.

About the SNR of 3 -- We should emphasize that this is for a single bin of 7.5m in the molecular range below the cloud base, as explained in the reply to comment R2. The SNR of the cloud it self is, of course, always much larger than that. Besides, the molecular fitting (below and above the cloud) works as an averaging procedure and hence the effective molecular SNR is also much larger.

About the algorithm performance -- We have done an extensive simulation study to validate the methods we use, which was not included in the original submission. We were planning to have a separate manuscript on AMT about the accuracy and precision of our cloud detection algorithm and of the transmission method for the retrieval of COD and LR from elastic lidars. However, as both referees have questioned about this, we believe that some of that needs to be included in the supplement material. We will consult the editor to see if he/she agrees with this approach.

In the simulations, we varied the cloud thickness (from 15m to 4.5km), the cloud extinction coefficient (from 0.02 to 0.1 km$^{-1}$) and the SNR from 3 to 50. We verified that our cloud detection algorithm can identify 99% of clouds with COD > 0.005 if the molecular SNR is at least 3 below cloud base. This is evidence that we are not suffering from SNR issues from the bright background.

The second evidence is that other studies have also reported a minimum in the cirrus occurrence around noon. Hong et al JGR (2006) used the TOGA radar during the TRMM-WETAMC campaign in the Amazon, and also the PR and VIRS instruments onboard of TRMM. They showed that the diurnal cycle of thick anvils (hence no SNR x bright Background issue) has minimum around 8-12LT. This is similar to what we found and fits perfectly with the diurnal march of tropical convection (Machado JGR 2002). There are examples in other regions as well. Throsen et al JGR (2013) used ARM data from SGP together with CALIPSO and showed that the thinnest cirrus occurs around 12h LT. Liu et al Adv. Atmos. Sci. (2015) used an MPL in southeastern China and showed the diurnal cycle of total cloud fraction also has a minimum around noon.

The third evidence is the diurnal cycle separated for sub-visual, thin and thick cirrus (see below). The left panel

uses the column total COD, hence the sum of the 3 curves will give exactly the black line in fig. 4 of the paper. In the right panel, we used the layer COD and hence the sum might be more than the total occurrence (because of multiple cirrus layers). There is a well marked diurnal cycle for the thick cirrus clouds. These clouds have COD > 0.3, hence there can't be any artifacts from SNR versus bright background issues. Their maximum occurs just after the maximum of precipitation and leads to the conclusion that they are actually formed from detrainment of the anvils, what fits nicely with our argument in the paper. The figure also shows that the maximum of thin cirrus occurs about 12h after the peak in thick cirrus. The COD of these clouds is way larger than the detection limit of our algorithm, and hence their diurnal cycle cannot be an artifact of low SNR during daytime. Together, thin and thick cirrus accounts for 60% of the total amount of cirrus and they both have a minimum around noon. The sub-visual cirrus (individual layers) show a prevalence during night-time. We are currently running more simulations to be sure that this is not an artifact. However, if there is one, it would be that we are missing day-time SVC and hence, that their true fraction is even larger than 40%.

Last but not least, we thank the reviewer again for raising this point. This forced us to further explore our dataset producing the figure/analysis below. We will modify the manuscript to make our argument more clear and self-evident, following this discussion.

Column COD                                          COD of individual layers

[Figure]

Fig – Diurnal cycle of cirrus frequency for each cirrus type for the year average. The sum of the three curves gives exactly the black curve shown in figure 4 of the manuscript.

R7: Furthermore, all of the speculation about the transport of clouds vs. near-source convective generation is very weak. The authors are forgetting that if the clouds are being generated at/near or on top of them, the lidar will not be profiling the clouds. You are *always* dealing with transport of some kind, as such. I recognize what they are trying to say, but recommend they be much more circumspect about how they are delineating source/transport with respect to the limited information that they have.

First of all let us clarify that when we talk about transported cirrus clouds we are specifically talking about long-range transport (Fortuin et al 2007). By the way, this was explained in the manuscript, at lines 256-258:

As the tropical cirrus can be transported by advection thousands of kilometers (Fortuin et al., 2007), we speculate that during the wet period, the cirrus clouds observed in central Amazonia are a mixture of locally produced and clouds transported by advection from other regions.

Nonetheless, we agree with the reviewer that our discussion about the sources of the cirrus we measured is weak, which was also noted by reviewer #1. He/she suggested that we could use back-trajectories to give further quantitative evidence that our cirrus clouds originate from deep convection. Hence, we did back-trajectory analysis using Hysplit forced by GDAS winds (1deg resolution), starting every 6h from 14.5 km over the site during the dry season period. Each of the 480 back-trajectories were integrated for 7 days. Figure below shows the result of this analysis. In the top panel, we show the individual trajectories just for the 0:00 of each day and there are so many lines that clutter the

plot. The lower panel shows the trajectory density, i.e., the number of trajectories in a point divided by the total number of trajectories (a number [0-1]). In this case we used a log-scale because the density will obviously be much higher closer to the trajectory start point. The result is quite interesting as it reveals that many trajectories actually don't follow the average wind pattern (fig. 3 in the manuscript, top panel). On the other hand, many trajectories come from Colombia and Venezuela, exactly where precipitation from deep convection is found (also shown in fig. 3, top), and some even reach towards the ITCZ, far to the east. This comparison could be improved if we select only the trajectories starting at times when we detected a cirrus clouds (yet do be done).

[Figure]

Figure C.1 – Hysplit 7day backward trajectories starting 14.5km above the site every 6h for the four months of the dry season. It should be compared to the top panel of figure 3 in the manuscript.

A way we could make this analysis more quantitative would be to run the back trajectories for each cloud layer detected and use GOES images to locate deep convective cells, and then calculate the distance between each trajectories and the surrounding precipitation (as a function of backward time). This is a huge effort and, we believe, deserves its own paper. Another possibility would be to do that, but just for one case study in each season (e.g. as Fourtin et al., 2007JGR). This, however, would not be very representative of the full dataset.

If the reviewer thinks these plots/analysis are interesting, we would be happy to extend it to the other seasons and include this discussion in the manuscript.

R8: The distribution of clouds as a function of COD also relates to sample bias and attenuation effects.

It is unclear which figure the reviewer is refereeing to. Moreover, we believe that after our reply to comment R6, it is now well explained how we are counting the profiles and how the method takes into account (and corrects for) sample bias and attenuation effects.

R8 (cont.) Yes, there is an exponential distribution of cirrus cloud occurrence with respect to COD (again, see what we have in Campbell et al. 2015). However, the distributions that you have with respect to subvisible, optically-thin and opaque clouds is absolutely not consistent with other studies.

We do not agree with the reviewer. Our distribution is very consistent with other studies! We have listed 7 papers in table 1, including one co-authored by the reviewer, which report distributions similar to ours. The fraction of SVC from these studies varies from 15 to 65% (but also vary the latitude), and we have found 40% of SVC. To mention the specific values: 15% (Seifert et al., 2007), 25% (Antuna and Barja, 2006), 38% (Goldfarb et al, 2001; Hoareau et al., 2013), 50% (Sassen and Campbel, 2000), 52% (Pandit et al., 2015), and 65% (Cadet et al., 2013).

R8 (cont.) There should rougly be a 50-60%/40-50% distribution between transluscent and opaque clouds. In Campbell et al. (2016), we see a very similar distribution as yours that we fully attribute to sampling bias. I see no reason to think this sample is not subject to the same effects.

We believe that after our reply to comment R6, it is now well explained how we are counting the profiles and how the method takes into account (and corrects for) sample bias and attenuation effects. Moreover, the proportion of 60%/40% mentioned by the reviewer cannot be taken as absolute. Firstly, it will not be the same in different locations. Particularly, there is no physical reason why it would be same over the Amazon (i.e. cirrus formed by deep convection, year precip > 2200m) and Greenbelt (frontal systems, year precip < 1100mm).

Secondly, the proportion will be different depending on the algorithm. Let's say the same dataset is analyzed by two different cloud-detection algorithms, one that can see clouds with very low COD (e.g. down to 0.001) and another that can detect only COD > 0.01. Of course, the amount of SVC detected will be very different, and hence the proportion of transluscent / opaque will be different!

R9: Although there is a point where the authors show a correlation between COD and cloud base, cloud base is a nearly useless parameter for such vigorous study. As myriad Sassen papers discuss and describe, cloud top is the most important layer because this is where cirrus cloud nucleate, grow and begin falling. Cloud base, as such, is redundant. Its simply the boundary where evaporation/sublimation is complete in falling crystals. So much effort in the narrative is spent on cloud base and drawing physical correlation, where it seems to have no physical meaning. Cloud top should be the focal point

We thank the reviewer for the suggestion. We will change the manuscript to focus on the cloud top.

R10: As such, there is absolutely no physical basis for evaluting lidar ratio versus mid- cloud temperature. It makes absolutely no physical sense. Now, I recognize that the CALIPSO team has done this very thing with their analyses. I don't agree with them either. But, they are dealing with a downward looking dataset, at least, and this offers other challenges that the authors are not dealing with in the zenith. Whereas I would accept if the authors referenced Garnier et al. (2015) and wanted to leave this as is, I still wouldn't think that it made much physical sense. In particular, as with CALIPSO, you're never actually going to know for certain what the mid-cloud temperature is (or unfortunately the cloud top temperature is) because of attenuation. For CALIOP, this is actually a bigger issues, since they can attenuate working downward with clouds that ground-based lidars would likely never reach. But, the comment still remains. I recommend sticking with what you can physically interpret, and particle effective size and habit are likelier in the long run to relate with available water vapor and temperature found at cloud top than somewhere within the cloud

We thank the reviewer for the suggestion. We will change the manuscript to evaluate the LR as a function of the cloud top temperature.

R11: No uncertainty analysis is provided for the lidar ratio analysis. This concerns me, again, because of the low SNR environments that you claim to be working with. As such, its unclear to me that you can actually develop meaningful correlative relationships, like Garnier, with a relatively low number of cases that the SNR would be sufficient and uncertainty suppressed. The uncertainty term presented appears to me to be a standard deviation, which again seems misrepresentative in context.

In the paper, we did not show any individual retrieved quantity. The tables and figures show average values, sometimes for the whole year, or season, or hours in a day. Hence, the uncertainties of individual retrieval are not given. Depending on the discussion, we reported either the error in the mean value, or the variability of the values.

However, we recognize that we should better explain how we evaluated the uncertainties in the LR and COD obtained by combination of the transmittance and Klett methods. Indeed, this was a request made be reviewer #1 as well.

To make it clear, we have calculated the uncertainties in the optical depth and LR. That comes from a simulation study we performed to access the accuracy and precision of our algorithms. In the simulations, we varied the cloud thickness (from 15m to 4.5km), the cloud extinction coefficient (from 0.02 to 0.1 $km^{-1}$) and the SNR from 3 to 50. Even for SNR=3, the difference between the true and retrieved LR was < 5 sr for COD = 0.01.

More details about the simulation can be found in the file attached with this answer. This is a draft of the material that we are preparing for the supplement.

R12: Please see my note about how you interpreted Chew et al. (2011). Its not correct. 34% of Level 2 AERONET observations were found biased by unscreened cirrus.

Thanks for pointing that out. We changed that in the manuscript.

R13: I recognize that this is a lot of stuff. I offer this with full respect to what you are trying to do, because its in my direct interest working so many years with MPLNET to see this sort of work get published. I present these thoughts in detail with the sincere hope of helping resolve what I believe to be significant scientific shortcomings in the narrative

We really appreciate your suggestions and, particularly, the time you dedicated for doing such a careful review of our manuscript. Your constructive criticism helped a lot to improve our work.

ATTACHED DOCUMENT – Hand written notes with many suggestions for improving the manuscript text.

We thank the referee for carefully reading. The suggestions for improving the English writing will definitely make the paper easier to follow. We have accepted all suggestions and made the changes in the manuscript.

**TO BE INCLUDED IN THE SUPPLEMENT MATERIAL**

**S.1 Simulations for evaluating the retrieval methods**

It is important to know the uncertainties in the retrieved cloud optical depth and lidar ratio, particularly because we are using the transmission method (Chen et al., 2002), which becomes very sensitive to signal noise for low optical depths. To estimate the effect of random signal noise in our retrievals and evaluate the errors for different COD, we did numerical simulations of lidar profiles having cirrus clouds with fixed LR of 20 sr. Cloud-base was fixed at 12 km and eight cloud-thickness were simulated: 15, 30, 45, 90, 150, 450, 1200, and 4500 m. For the cloud extinction coefficient, two values were simulated: 0.02 and 0.1 km$^{-1}$, thus the COD ranged from 3 x 10$^{-4}$ to 0.45. Random noise following a Poisson distribution was added to the simulated photon-count signal to get signal to noise ratios of 50, 10, 5 and 3 in a single bin just below the cloud base. For each combination of COD and S/N, 100 simulations were performed. The simulated profiles were processed with the same algorithm used for atmospheric data. Therefore, we can evaluate the uncertainty in the COD and LR as a function of the S/N by calculating the mean, the standard deviation, and the standard deviation of the mean over these 100 realizations. The standard deviation will give how the signal random noise might affect the retrievals, while the mean and the standard deviation of the mean will show if the retrieved values converge to the expected values, after many observations.

[Figure]

**Figure S1**: Example of background and range corrected signals (top) and the corresponding S/N ratio (bottom) are shown. The blue curve is a measured lidar profile with 5-min temporal average and 7.5 m vertical resolution, while the red curve is a simulated profile with similar S/N ratio at cloud base and a cirrus cloud of optical depth 0.12. The black curve is the same measured profile but with a 5-bin vertical binning (37.5 m), and thus a higher S/N ratio.

Figure S1 shows an example of a measured profile with 5 min average and original 7.5 m vertical resolution, from some day in July 2011. The system shows a good performance. Typical S/N ratio for the molecular backscatter at 12 km of altitude, for this temporal and spatial resolution, varies from 6 to 20, depending on the presence of low clouds and the solar background. This S/N ratio can be improved, for instance, by reducing the vertical resolution as shown in the lower panel (black curve).

**S1.1 Uncertainty in the retrieval of COD**

As discussed in section 2.4, for the calculation of the optical depth with the transmittance method, it is necessary to fit the molecular part of the signal below and above the cloud. Considering a large region for those fits, in our case 1 to 10 km, helps to reduce the effect of the noise. The difference between the two fits gives the cirrus transmittance and the optical depth is half the natural logarithm of that value (eq. 2 and 3). Figure S2 shows the mean COD and the standard deviation of that mean value, for the 100 simulations. These results show that the magnitude of absolute mean error (mean COD – truth) is independent of the true COD. The root mean square error (RMSE) is $2.5 \times 10^{-3}$, for S/N = 3, and only $2.3 \times 10^{-4}$, for S/N = 50. That is for the averages over 100 simulations, for single profiles it is 10 times larger. The relative error is smaller for large COD values. This error is less than 20% for COD > 0.005 and S/N = 3, and less than 6% for COD > $4.5 \times 10^{-4}$ (minimum value) and S/N = 50 (largest value). We note that even for very low S/N ratio and small COD the method still find a mean value compatible with the true COD.

[Figure]

**Figure S2**: COD calculated by the transmittance method as a function of the true COD for different S/N ratios. The error bars are the standard deviation of the mean values. The absolute differences (lower panel) are all compatible with zero (i.e. mean calculated COD is compatible with true COD).

The cloud optical depth can also be calculated by integrating the extinction coefficient obtained with the Klett method, however an a-priori LR is required. We use this method in two cases. First, when there is more than one cloud layer. In this situation, the transmittance method gives the total cloud optical depth (all layers combined) and that is used to obtain an average LR (all layers combined), the same way as explained in section 2.4. The extinction profile from the Klett method (with that average LR) is then used to divide the total optical depth into contributions from each layer. The second case is when the interactive method described in section 2.4 fails to converge, i.e., when it cannot find a reasonable LR value that makes the Klett inversion give the same optical depth as the transmission method. This happens for XX % of our profiles with clouds and they have very low optical depth, about YY, and only ZZ m of thickness. These profiles are usually those near the edges of the clouds. In these cases, the cloud optical depth is obtained with the Klett method by assuming a LR equal to the average value obtained from all the other profiles (i.e. the ones when we could determine the LR, for the current version of the manuscript it is about 20 sr).

Figure S3 shows the COD obtained by this method in the best scenario, i.e. when imposing the true LR for the simulations (20 sr). We can see that the Klett method is much less sensitive to the S/N ratio. The RMSE is $2.8 \times 10^{-4}$, for S/N = 3, and $5.5 \times 10^{-5}$, for S/N = 50, both much smaller than the mean errors obtained with the transmittance method, but also closer to each other. As for the transmittance method, the Klett method also finds a mean value compatible with the true COD even for very low S/N ratios and small CODs.

[Figure]

**Figure S3**: COD calculated by the Klett method, assuming the true LR = 20 sr, as a function of the true COD for different S/N ratios. The error bars are the standard deviation of the mean values. The absolute differences (lower panel) are all compatible with zero (i.e. mean calculated COD is compatible with true COD).

It should be noted, however, that a wrong guess about the LR would bias the retrieved CODs obtained with the Klett method. To quantify that effect, we applied the Klett method assuming a LR value 50% higher and lower than the true value (i.e. 10 and 30 sr) and S/N of 50 (so that it can be disregarded). The result is shown in figure S4 together with the result for the transmission method with S/N ratio of 10. It is clear that the COD retrieved by the Klett method is only as good as the estimative of the LR.

[Figure]

**Figure S4**: COD calculated by the Klett method, for LR = 10, 20 (true) and 30 sr, is shown as a function of the true COD for S/N = 50. Points in green are the transmittance method for S/N = 10. The error bars are the standard deviation of the mean values.

Figure S5 shows the relative errors for both methods. This is defined as RMSE/True_COD. As expected, the lower the S/N ratio the higher the error. In the worst case, i.e. S/N = 3, the relative error from a single retrieval using the transmittance method is below 20% only for COD > 0.1. For S/N = 10, this limit is COD > 0.025. This error, however, is random and fluctuates around zero as shown previously. By averaging over 100 profiles (right panel, Fig. S5), the relative errors decrease by a factor of 10. Under these circumstances, i.e. with many profiles, or if the S/N is high or if the COD is not very small, it is advantageous to use the transmittance method because it does not depend on an a-priori LR. In our study, we analyzed about 37k 5-min profiles, where 21k had S/N > 3 at 12km and in 14k of these we found a cirrus cloud. Thus, the error in the mean cloud optical depth reported in Table 1 is indeed much lower than shown in the right panel of Fig. S5.

As expected, the relative errors for the Klett method with the true LR are always smaller than those from the transmittance method for the same COD (Fig. S5, compare the respective lines with triangles and circles). However, there is a large uncertainty from the value of choice for the LR. The dashed black line shows the relative error from choosing a LR of 10 or 30 sr. The induced bias in the retrieved COD is proportional to the change in LR, hence the relative error is approximately constant. The 50% change in the LR translates in a relative error of about 50% for small COD, and 30% for large COD.

[Figure]

**Figure S5**: Relative error (in %) in determining the COD for both methods as a function of the true COD for the different signal to noise ratios are shown.

**S1.2 Uncertainty in the retrieval of LR**

As explained in section 2.4, the LR is estimated by a minimization procedure in which the LR is allowed to vary from 2 to 50 sr. The optimal LR is the one making the cloud optical depth from the Klett algorithm equal to that from the transmittance method. Typically, we are able to estimate the LR for clouds with COD > 0.01, which is about 91% of our observations. Below that threshold, the COD is not very sensitive to changes in the LR and the method does not converge. That is why we estimate the COD, in these cases, with the Klett method and a fixed LR = 20sr.

Figure S6 shows the results of the LR estimated for the same simulated profiles used for the evaluation of the COD retrievals but only for the cases when the LR algorithm converged. When the S/N is low or when the COD is small, there is a tendency of overestimating the LR (all deviations are positive). However, all retrieved values are still compatible with the true value (t-score < 3) and the maximum deviation is just 4.7 sr. Moreover, the variability of the calculated LR (standard deviation, i.e. 10 times the error bars in Fig. S6) decreases with increasing COD. For S/N of 5, it is 12 sr for COD = 0.02 and 3 sr for COD = 0.45. The variability also decreases with increasing S/N ratio. This is shown in the histograms in figure S7, for COD > 0.02 and S/N of 5 and 10. The variability was reduced from 5 to 2.5 sr, respectively. For these cases with somewhat larger COD, it is clear that there is no bias in the mean retrieved LR.

[Figure]

**Figure S6**: LR calculated from the combination of the transmittance and Klett methods, as a function of the true COD for different S/N ratios. The error bars are the standard deviation of the mean values. The absolute differences (lower panel) are all compatible with zero (i.e. mean calculated LR is compatible with true LR).

[Figure]

**Figure S7**: Histograms of LR calculated from the combination of the transmittance and Klett methods for S/N = 5 (left) and S/N = 10 (right) are shown just for COD > 0.02.

---

## Author Response (AR1)

We would like to thank the editor for accepting our manuscript for consideration in ACP, based our responses in the online discussion. As we already gave a very detailed response to all questions and comments from both reviewers, here we give only updated replies to the most important questions and to the handwritten comments annotated in the scanned PDF by one of the reviewers.

Reviewer comments are in red and start with **"R:"** and our replies are in black and start with **"A:".** Original manuscript text is shown in **blue**, with new text highlighted in yellow.

With this response, we are also attaching 1) an annotated version of the manuscript indicating all changes; and 2) the updated manuscript.

===================================
**Most important concerns from reviewer #1**

R4: The paper would be greatly enhanced by making a more quantitative effort the explain their data. For example, instead of speculating on the sources of moisture for the cirrus in different seasons, a more convincing approach would be to run back trajectories to show the reader where the air came from.

A: We agree that giving a more quantitative explanation would strengthen our paper. Following the referee's suggestion, we did 24-h back-trajectory analysis using Hysplit forced by GDAS winds (0.5 deg resolution), starting from the time and altitude of each detected cirrus layer. Below is the updated figure 3 in the manuscript. The result is quite interesting as it reveals that many trajectories actually don't follow the average wind pattern. On the other hand, many trajectories come from Colombia and Venezuela, exactly where precipitation from deep convection is found (dry season). During the wet season, most trajectories point to the near-by convection. More discussion is given in section 3 of the updated manuscript.

As we mentioned before, this trajectory analysis (suggested by referee #1) indeed gives further evidence that our cirrus clouds originate from deep convection but it is not a quantitative evidence.

[Figure]

Figure 3. Left panels show mean precipitation (colors, mm/day) from the TRMM 3B42 version 7 and mean wind field (vectors, m/s) at 150 hPa from ECMWF ERA Interim reanalysis. Right panels show 24 h back trajectories of

air masses arriving at the site at the time and altitude that cirrus layers were detected. Back trajectories were computed using HYSPLIT model with 0.5° resolution winds from GDAS/NOAA. Results are shown separately for the dry (JJAS, top) and wet months (JFMA, bottom). The experimental site location is indicated in all panels.

R5: Is there reason the authors don't use the nitrogen signal to retrieve extinction? Not doing so doesn't completely discount the data presented, but it does devalue it somewhat since this paper is just another in a long-line of elastic lidar cirrus studies.

A: For our system, the weak Raman signal is only discernible from the background during nighttime, thus not being appropriate for our study, which wanted to investigate the diurnal cycle as well. Moreover, the Raman method involves calculating the derivative of the signal, which gives rise to large uncertainties in cases of such low signal-to-noise ratios as typically found at cirrus altitudes.

Besides, our experience with simulated signals have shown us that we can obtain a very precise and accurate LR with Chen's method. We did this analysis of simulated cloudy lidar profiles to evaluate the accuracy of the transmittance method but did not include in the manuscript, or in the supplement (but see discussion following your next question). We believe that these simulations and the fact that Raman methodology can in our case not be applied during daytime justify our method of choice.

R6: In addition, the transmission method is really only accurate for mid-range optical depths. Too thin and there isn't enough transmission signal to get a reliable optical depth. Too thick and there isn't enough molecular signal above the cloud. I encourage the authors to go beyond just checking the SNR above/below the cloud when doing the optical depth retrieval and to fully derive the uncertainty in the optical depth values they report. Figures 5 and 6 show optical depths down to 0.001, which I expect to be extremely uncertain when using the transmission method to retrieval optical depth.

A: We agree with the reviewer's point of view and, in fact, we have already calculated the uncertainty in all optical depth values that we have obtained. However, the plots and tables shown in the manuscript are always for averages over a huge amount of profiles, and hence we choose to report the standard deviation of the mean instead of the errors in individual retrievals.

We also agree that the COD uncertainty is very large if obtained for a single profile with COD = 0.001. In fact, we have done an extensive simulation study to validate the methods we use, which is now included as a supplement to the manuscript. For COD = $10^{-3}$, the relative error in a single retrieval is 120% for S/N = 50 and 1150% for SNR = 3, both large but not enough to change the cirrus category (e.g. from sub-visual to thin). Moreover, averaging over N profiles reduces this uncertainty by a factor of square-root of N. In our study, we analyzed about 37k 5-min profiles, where 21k had S/N > 3 at 12km and in 14k of these we found a cirrus cloud. Thus, the error in the mean cloud optical depth reported in Table 1, or in the histograms in figures 5 and 6, is indeed much lower, typically below 20% even for S/N = 3.

R7: The treatment and discussion of multiple scattering could be improved. Although, not explicitly stated, I'm guessing the authors use Eq. (10) from Chen et al. (2002) where eta depends on the optical depth of the cloud layer. I'd would encourage against using this equation. Chen et al. provide no physical justification (…)

A: We thank the referee for explaining the limitations of the correction proposed by Chen. To appropriately account for the multiple-scattering, we reviewed the work of Platt (1981) and Wandinger (1998) and finally decided to apply a full treatment following the model of Hogan (2008). Our new results show a change of the LR from ~16.8 sr to ~23.6 sr, while Chen's formula was giving 20.2 sr. That means that Chen's correction was indeed not valid for our case!

Reprocessing all the dataset with this more complex algorithm for multiple-scattering took much longer than anticipated and it is the main reason for the delay in submitting the updated manuscript for ACP. Nonetheless, we thank the reviewer for pushing us to do the necessary corrections.

===================================
**Most important concerns from reviewer #2**

R1: My primary scientific concern relates to the definition of "cirrus" clouds in the manuscript. (…) In a recent paper that I authored (Campbell et al. 2015), we went to significant length to demonstrate a practical and viable definition for cirrus clouds in autonomous long-term datasets like this one, and in particular for those that lack a polarized backscatter measurement. (…)

A: We agreed with the reviewer and changed our definition to follow Campbell et al (2015). As we alerted during the online discussion, however, the amount of clouds that we no longer consider to be cirrus is less than 6%. Therefore, the numbers in the update manuscript are not so different after all. Nonetheless, a physically based definition is indeed preferred, and we thank the reviewer for the suggestion.

R2: Its unclear what the authors are saying about the presence of SNR > 3 in the upper troposphere with respect to cloud observation. Do they mean clear-sky? Or, do they mean within particulate scattering layers?

R4: Since the sample size is stated to relative to the ability to measure SNR > 3 in the upper troposphere, all of the samples appear to be relative occurrence frequencies and not absolute ones. This is HIGHLY confusing. There is no way that you're resolving an absolute cloud frequency of 67%. (…) which owes to the attenuation of the beam from low-level clouds and undersampling of the upper troposphere. (…)

R5: Speaking of this issue, nothing is said of the work of Thorsen et al. (2011) and Protat et al. (2014) and undersampling issues relating to ground-based profiling, attenuation, and the relative cloud samples that we have to analyze. This is a serious weakness that leads to three other points of concern.

A: These three questions were basically about the same issue. We apologize for not having stated it clearly in the original manuscript. As we promised in the online discussion, we modified section 2.2, removing the discussion about the SNR, and we created a new section called "frequency of occurrence and sampling issues". This is shown below.

**2.4 Frequency of Occurrence and Sampling Issues**

In a simplified manner, the frequency of occurrence would just be the ratio of the number of profiles with cirrus clouds to the total number of profiles. However, while one might be sure when a cirrus was detected in a given profile, there is no certainty of its presence when the profile has a low SNR or when there is no measurements. Sampling cirrus clouds with a ground-based profiling instrument might be problematic, particularly for the calculation of the frequency of occurrence, due to the obscuration by lower clouds or availability of measurements, which might introduce sampling biases (Thorsen et al., 2011).

To avoid these sampling issues, we use an approach similar to the conditional sampling proposed by Thorsen et al. (2011) and Protat et al. (2014). Firstly, we recognize that the presence of cirrus clouds is rather independent of low-level water clouds that can fully attenuate the laser beam, and independent of instrumental issues that might restrict measurement time. Hence, the best estimate of the true frequency of occurrence is the ratio of the number of profiles with cirrus, by the number of profiles where cirrus could have been detected.

These good profiles are identified as follows. The noise in each clear-sky segment of each 5-min lidar profile is evaluated. Profiles are selected if a clear-sky signal to noise ratio (SNR) higher than 3 is found at the typical cirrus altitude, and considering 7.5 m vertical resolution. This threshold was obtained from a performance evaluation of the detection algorithm and transmittance methods based on simulations for various SNR, COD and cloud thickness (not shown). What we found was that our algorithm can detect 99% of cirrus clouds with COD > 0.005 if the SNR is at least 3 below cloud base. In other words, given the typical cirrus cloud optical depths, the used threshold implies in a good enough SNR at cloud top for applying the transmittance method.

From the analysis of the available profiles, 20,752 were found to satisfy these criteria. July, August and September, the driest months, show the higher fraction of profiles with good SNR, while the wettest months have the lowest fraction of lidar profiles with good SNR (see figure S.1). To avoid introducing biases from the different sample sizes in different months, the frequency of occurrence for the year is calculated as the average frequency of

occurrence for each season. The frequency for each season, in turn, is calculated from the frequency of each month. Finally, the frequency for each month is calculated by averaging over the diel cycle. This is preferred than calculating the daily averages because there are more profiles with good SNR during night versus daytime.

R6: It is discussed that the lowest cloud observations occur around solar noon (10-12 LT). This leads me to believe that your instrument is suffering from issues with SNR from the bright background, even at 355 nm. Whereas it is introduced that this is potentially a real artifact, I see no reason to take such a claim at face value.

We thank the reviewer for carefully looking at all details of our results. We should say, however, that we also have looked into this minimum around solar noon to be sure that it was not a problem with the solar background. Below is the new figure 4, and the accompanying text that explain why the minimum around noon is real.

To verify that the lower cirrus cloud cover around noon was not related to a decrease in SNR and, hence, a decrease in detection efficiency, we analyzed the frequency of occurrence for different cirrus types (following Sassen and Cho, 1992). Opaque (COD > 0.3), thin (0.3 > COD > 0.03) and sub-visual cirrus (SVC) clouds (COD < 0.03) were considered. Their diurnal variation is shown in Figure 4. The larger amplitude, during both dry and wet seasons, is actually of the frequency of occurrence of opaque cirrus. During the dry (wet) season, it increases from less than 5 % (20 %) to about 30 % (50 %) in the hours following the precipitation maximum, 15 h to 19 h LT. The second larger diurnal variation is of the frequency of thin cirrus, which decreases after the sunrise from 30 % (50 %) to 20 % (30 %) during the dry (wet) season, and increase again during night time, when the opaque cirrus are dissipating. The SVC, which detection could be biased by lower SNR, do not show a clear diurnal cycle. Hence, the diurnal cycle of the frequency of occurrence of cirrus clouds in central Amazonia comes from the diurnal cycles of opaque and thin cirrus, which have a large enough COD not to be missed by the detection algorithm, even for lower SNR.

[Figure]

Figure 4. In the top panel, diel cycles of the hourly frequency of occurrence of cirrus clouds are shown for the annual, wet (JFMA), transition (ONDM) and dry (JJAS) periods. In the middle, panels show the same but for SVC, thin and opaque cirrus clouds during the dry (JJAS) and wet seasons. Lower panel shows mean observed

precipitation rate (mm/h) from TRMM version 7 over an area of 2° × 2° centered on the site for the dry (+) and wet periods.

================================
**Hand written notes by Reviewer #2**

**R:** L. 33 – Not 30% . See Mace et al, 2007

**A:** Mace et al GRL 2007 used CloudSat and his paper is about hydrometeors in general, and just for one boreal summer. He says that the global average occurrence of any type of clouds is 50.6%, hence it is not possible that cirrus alone represent 40-60% as suggested by the reviewer. Sassen et al. JGR 2008 used 1-yr of Calipso + CloudSat data and found that global average cirrus coverage is 16.7%, which compares well with ISCCP data (13.2 to 19%). In a follow up study, Sassen et al JGR 2009 showed that *"about 35% of this cirrus coverage occurred within ±15 latitude and 56% within ±30 latitude of the equator"*.

We have modified the manuscript that now reads:

Clouds cover on average about 50 % of the Earth´s atmosphere (Mace et al., 2007) and cirrus alone cover 16.7 % (Sassen et al., 2008), with higher fractions occurring in the Tropics (Sassen et al., 2009), hence cirrus are important to understanding current climate and predicting future climate (Wylie et al. 2005, Stubenrauch et al. 2006; Nazaryan et al., 2008).

Mace, G. G., R. Marchand, Q. Zhang, and G. Stephens (2007), Global hydrometeor occurrence as observed by CloudSat: Initial observations from summer 2006, Geophys. Res. Lett., 34, L09808, doi:10.1029/2006GL029017.

Sassen, K., Z. Wang, and D. Liu (2008), Global distribution of cirrus clouds from CloudSat/Cloud-Aerosol Lidar and Infrared Pathfinder Satellite Observations (CALIPSO) measurements, J. Geophys. Res., 113, D00A12, doi:10.1029/2008JD009972.

Sassen, K., Z. Wang, and D. Liu (2009), Cirrus clouds and deep convection in the tropics: Insights from CALIPSO and CloudSat, J. Geophys. Res., 114, D00H06, doi:10.1029/2009JD011916.

**R:** L53 Define Sassen and Cho 92

A: To avoid making the definition of thin and subvisible cirrus clouds too early, we modified the text that now reads:

Ground-based lidars are an indispensable tool for monitoring cirrus clouds, particularly those cirrus clouds with very low optical depth, which are undetectable by cloud radars (Comstock et al., 2002) or by passive instruments (e.g., Ackerman et al., 2008).

**R:** L128 – 135 What accommodation did you use for the TTL?

**A:** As explained in our response during the online discussion, we calculated the TTL altitude using WMO definition and the ERA Interim temperature profiles. This paragraph now reads:

Temperature, pressure, geopotential height, humidity and winds for the study period were obtained from the ERA Interim reanalysis (Dee et al., 2011) of European Center for Midrange Weather Forecast (ECMWF) with spatial resolution of 0.75° and temporal resolution of 6 h. The tropopause altitudes were obtained from ERA Interim temperature profiles over the site using the definition of the World Meteorological Organization (IMV WMO, 1966), i.e. *"the lowest level at which the lapse rate decreases to 2°C/km or less, provided that the average lapse rate between this level and all higher levels within 2 km does not exceed 2°C/km"*. We further assumed the lapse rate to vary linearly with pressure (McCalla, 1981), and the exact altitude where Γ=2°C/km (i.e. the tropopause) was found by linearly interpolating between the closest available pressure levels. A precipitation dataset (…)

**R:** L 189 – What is the absolute rate of cloud occurrence?

**A:** The absolute rate (i.e. number of profiles with cirrus divided by the total number of profiles measured) does not have any physical meaning (although readers may think so) and we refrain from giving that information in the paper. This absolute rate is known to be biased by low clouds, number of hours of observation, etc… (see for instance Thorsen et al, 2011 and Protat et al., 2014).

The numbers we gave at lines 193-194 are not relative numbers, as suggested by the reviewer. These are the best estimative of the true cirrus frequency. These are obtained by first excluding all profiles that do not have good clear-sky SNR at typical cirrus altitudes. We then divide the number of cirrus profiles by the total remaining. This is exactly the "conditional sampling" suggested by Thorsen et al (2011), mentioned by the reviewer.

**R:** L 246 – "mean temperature"'is defined how? Base? Top?

**A:** In this case, we are giving the temperature at the altitude of the maximum backscatter. We have modified the text to make it more clear:

The mean value of the cloud maximum backscattering altitude is 13.2 ± 2.3 km and the corresponding temperature is −58 ± 17 °C.

**R:** L 248 – Relative frequency

**A:** As explained above and in the interactive discussion, absolute frequencies are just wrong and should not be reported. Moreover, our frequencies are not relative. We used the "conditional sampling" approach and, thus, they are the best estimate of the true cirrus frequency.

**R:** L 331 – Nucleating level

**A:** We thank the reviewer for the explanation. We have add it to this paragraph as follows:

As expected, there is no obvious relation of cloud top and COD. The nucleating level is independent of the vertical extent of the cloud (i.e. COD). During the dry months (…)

**R:** L361 Consistent with Campbell et al 2016

**A:** These lines now read:

The classification of cirrus clouds following Sassen and Cho (1992) shows that 40.0 % of the cirrus clouds measured in our experimental site are subvisible (τ < 0.03), 37.7 % are thin cirrus (0.03 < τ < 0.3) and 22.3 % are opaque cirrus (τ > 0.3), which are consistent with Campbell et al. (2016). Table 2 shows (…)

**R:** L395 How did you derive the uncertainty?

**A:** Sorry for not explaining this clearly. In this case, we are just reporting the average value and the standard deviation of the distribution. The new text reads:

A mean value of 20.2 ± 7.0 (std) sr was obtained for the whole period and it varies less than 1.5 sr for different seasons. Pace et al (…)

**R:** L411 Why mid-level, you're likely attenuated

**A:** As suggested by the reviewer, we avoided analyzing base or mid-cloud temperatures. In the updated version, everything is related to cirrus top temperatures.

**R:** L425 Relative frequency where you had sufficient SNR

**A:** As discussed in the online discussion, our methodology of counting the profiles (and hence the cirrus frequency) is exactly the "conditional sampling" suggested by Thorsen et al (2011), mentioned by the reviewer. Therefore, the reported frequencies are the best estimative of the true cirrus frequency.

**R:** L432 SNR?

**A:** In our reply in the online discussion, we have shown a plot of the diurnal cycle of cirrus frequency for the different categories of cirrus. This is now the updated figure 4, which was shown above. As opaque cirrus has a very large COD (>0.3), which is well above our detection limit, the SNR cannot affect its detection. Therefore, the opaque (or even thin) cirrus marked diurnal cycle cannot be not an artifact of the sampling or of the lower SNR around noon.

[revised manuscript text omitted]

Unknown

---

## Referee Report (RR1)

[referee-annotated manuscript omitted]

---

## Referee Report (RR2)

The authors have made significant improvements to the manuscript. However, it remains unclear how two of my major comments on the initial draft where addressed.

(1) Statistical significance still doesn't quite appear to be assessed correctly. The text is still vague on this, but it appears that the authors are only checking if the means +/· the standard errors overlap. Just because the errors bars do not overlap, does not mean the difference is significant. An actual statistical test (e.g. 2 sample t-test) needs to be used to assess differences. For example, in Table 3 for "All Layers" the lidar ratio differences are not statistically significant at 95% confidence (p=0.16). The authors should revisit the statistical test used and/or better explain what is being done in the text.

Explain what is being uone in the text. (2) I comment the authors for undertaking a full treating of multiple scattering. But, it is not clear how exactly the authors performed the correction. In the text, the authors simply state that they "perform a full treatment of multiple scattering following the model of Hogan (2008)". But the Hogan model is a forward model: i.e. it requires inputs of the true (single scattering) backscatter/extinction and from that computes the measured (single and multiple scattering) signal. Therefore, what is retrieved from the lidar cannot be directly inputted into the Hogan model to get the multiple scattering effects. I suggest that the authors elaborate more on how they correct for multiple scattering.

---

## Author Response (AR2)

We thank the two reviewers for carefully reading our manuscript and for the important suggestions to improve it. Below we give detailed responses to all questions and comments from both reviewers. Reviewer comments are in red and start with **"R:"** and our replies are in black and start with **"A:"**. Original manuscript text is shown in blue, with new text highlighted in yellow.

With this response, we are also attaching 1) an annotated version of the manuscript indicating all changes; and 2) the updated manuscript.

=====================================
**Reply to Reviewer #1**

**R:** (1) Statistical significance still doesn't quite appear to be assessed correctly. The text is still vague on this, but it appears that the authors are only checking if the means +/- the standard errors overlap. Just because the errors bars do not overlap, does not mean the difference is significant. An actual statistical test (e.g. 2 sample t-test) needs to be used to assess differences. For example, in Table 3 for "All Layers" the lidar ratio differences are not statistically significant at 95% confidence (p=0.16). The authors should revisit the statistical test used and/or better explain what is being done in the text.

**A:** We apologize for not making this clearer in the text, but the statistical test we performed was actually a two-sample t-test between the two independent data sets (wet and dry) and not just comparing if means +/- the standard errors overlap as the Reviewer thought. Fortunately, the reviewer found the only typo in the table. The correct value for the lidar ratio in the dry season considering all layers is 24.4 sr and not 22.4 sr, as it was written. The typo probably occurred when we were transcribing the data from Matlab into the Word table.

We fixed this typo in the table and reviewed all other values. We could provide t-scores of this comparison by adding a new column in the table, but we felt that this would further pollute the table and would not bring relevant information. Nonetheless, for the reviewer benefit, we included t-scores in the two tables attached to this response. Please note that because of the high number of degrees of freedom, a 95% confidence is already found for t > 1.645. The new captions for tables 2 and 3 are:

Table 2. Summary of column-integrated statistics for the total time of observation, as well as for the wet, transition and dry seasons. Frequency of occurrence is calculated using a conditional sampling to avoid biases (session 2.4). Mean cirrus cloud properties and standard deviation of the sample (in parenthesis) are shown. The standard deviations of the mean were calculated and used to determine if seasonal differences (wet-dry) of the mean values are statistically significant to the 95% confidence level (indicated as *) using a 2-sample t-test. Geometrical properties are not given because most cloud profiles have more than one layer of cirrus. Lidar ratio is calculated as a column average.

Table 3. Summary of layer-statistics for the total time of observation, as well as for the wet, transition and dry seasons. Mean cirrus cloud properties and standard deviation of the sample (in parenthesis) are shown. The standard deviations of the mean were calculated and used to determine if seasonal differences (wet-dry) are statistically significant to the 95% confidence level (indicated as *) using a 2-sample t-test. Lidar ratio is calculated as a column average.

**R:** (2) I commend the authors for undertaking a full treating of multiple scattering. But, it is not clear how exactly the authors performed the correction. In the text, the authors simply state that they "perform a full treatment of multiple scattering following the model of Hogan (2008)". But the Hogan model is a forward model: i.e. it requires inputs of the true (single scattering) backscatter/extinction and from that computes the measured (single and multiple scattering) signal. Therefore, what is retrieved from the lidar cannot be directly inputted into the Hogan model to get the multiple scattering effects. I suggest that the authors elaborate more on how they correct for multiple scattering.

In fact, the Hogan model is a forward model and thus the multiple scatter correction is done iteratively until the simulated multiple-scattered signal by Hogan's model converges to the true signal measured with the lidar. In the original Hogan model paper (Hogan, 2006 APPLIED OPTICS) he says that the purpose of the model is to be accurate and fast enough to be used iteratively and be used in a variational retrieval approach. Delanoe and Hogan (2008 JGR) have exemplified in detail this iterative process for ice clouds using the Hogan model, but this iterative method solution has already been used previously using Monte Carlo calculations of the effects of multiple scattering (e.g., Reichardt, 2006 APPLIED OPTICS)

The correction is done using a parameterization of ice crystals effective radius and the uncorrected extinction coefficient profile (iteration 0) to simulate a first SS/MS correction. This first correction is then applied to the original lidar profile and then we recalculate the extinction coefficient profile with the first correction (iteration 1). We re-run the model with the first correction of the extinction coefficient and the result calculated by the model is closer to the original lidar signal. We repeat this process (iteration 2,3,4 ...) until the signal calculated by the model converges to the original signal measured by the lidar. In general, with 2 or 3 iterations the result is already close to the corrected value, but we iterate 5 times. This iterative process was applied in all our profiles, which took a considerable computational time, even being the Hogan model accurate and fast.

To make this information clearer, the paragraph has been modified and now it is:

(…) For this reason, we refrain from applying empirical correction formulas (e.g. such as eq. 10 in Chen et al., 2002), and  instead perform a full treatment of multiple scattering following the model of Hogan (2008) . The correction is found iteratively, similar to Seifert et al. (2007) and Kienast-Sjögren et al. (2016). The forward model is initialized with the originally retrieved, uncorrected extinction profile, and the model output is used to correct the extinction profile iteratively, until it converges. In our case, we assumed the effective radius of ice crystals to vary with temperature according to a climatology of aircraft measurements of tropical cirrus data (Krämer et al., 2016a, 2016b), which includes the recent ACRIDICON field campaign with the German aircraft HALO in the Amazon region (Wendisch et al., 2016). The full treatment (…)

===================================
**Reply to Reviewer #2**

We would like to thank reviewer # 2 J. Campbell for his comments, editing recommendations and questions. All of his textual corrections were accepted and corrected in the manuscript. Below are the replies to the other comments.

**R:** L. 123 – Consecutive?

**A:** Yes, consecutive. We do 1-min data acquisition. For this paper, however, each profile is the result of a continuous 5 min acquisition (i.e. time average), which represents 3000 laser shots at the repetition rate of 10Hz of our laser (a Quantel CFR-400).

We have modified the text to make this clear:

The lidar dataset used in the present study comprises measurements recorded between July 2011 and June 2012, which were temporally averaged into 5-min profiles (3000 laser shoots at 10 Hz). A total of 36,597 profiles were analyzed, corresponding roughly to 1/3 of the maximum possible number of profiles during 1 year.

**R:** L. 228 – Worries me slightly, but small concern. May want to check out Heymsfield et al, 2014 (JAS) to see how they compare?

**A:** I first came across the work of Heymsfield et al, 2014 (H14) through Campbell et al, 2016-JAMC and Lolli et al, 2016-ACPD. For temperatures below -60° C (usually> 12.5 km for our region) the parameterization of the effective radius of H14 and that provided by Kramer et al., 2016ab (K16) in tropical regions are very close, with a mean effective radius between 10 and 30 μm. For temperatures higher than -50°C, H14 correlates better with the K16 result for mid-latitudes (also with Wang and Sassen, 2002 in min-Lat.), which is on average larger when compared to tropical regions. Because the work of K16 also included very recent measurements in the Amazon region in 2014 (field campaign with German aircraft, HALO, for an overview see Wendisch et al. BAMS, January 2016), we chose to use K16.

We have modified the text to include the reference to the HALO campaign in the Amazon, and also to clarify to reviewer #1 how we did the multiple-scattering correction:

(…) For this reason, we refrain from applying empirical correction formulas (e.g. such as eq. 10 in Chen et al., 2002), and we instead perform a full treatment of multiple scattering following the model of Hogan (2008) that was used by. The correction is found iteratively, similar to Seifert et al. (2007) and Kienast-Sjögren et al. (2016). The forward model is initialized with the originally retrieved, uncorrected extinction profile, and the model output is used to correct the extinction profile iteratively, until it converges. In our case, we assumed the effective radius of ice crystals to vary with temperature according to a climatology of aircraft measurements of tropical cirrus data (Krämer et al., 2016a, 2016b), which includes the recent ACRIDICON field campaign with the German aircraft HALO in the Amazon region (Wendisch et al., 2016). The full treatment (…)

**R:** L. 306 – Average? Wow, that is high! Frequency absolute or relative?

**A:** When we speak of frequency of occurrence we are referring to the best estimate of the true frequency at which cirrus clouds are present over the measurement site, and is calculated considering all possible problems related to the extinction of the laser pulses by low clouds or other acquisition problems (as discussed in the new section 2.4, where we explain the conditional sampling proposed by Thorsen et al (2011) and Protat et al. (2014)). To keep the information complete, we also reported the number of profiles attenuated by low clouds or by something else at low levels (.i.e. and hence not used) in Table 2. This was also mentioned in this section.

**R:** L. 313 – repeated?

**A:** No. In the first paragraph we were considering the average value of this property when considering the entire column (integrated), which can contain several distinct clouds layers at the same time (data presented in table 2, where we also inform the average number of layers observed). This is interesting for possible comparison with non-profiling instruments (like sun photometers), because these will measure, for example, the optical depth of the whole integrated column.

The second paragraph is when we start to consider the properties of each cloud layer separately. As described in the text, distinct layers are considered when there is at least a column of 500 m of cloud-free air between them. The mean values for these distinct layers are shown in table 3.

To avoid this confusion, we rewrite the paragraph and make that clearer. Now it reads:

Table 2 shows column-integrated statistics of the properties of cirrus clouds during the one-year observational period, also  distinguished by season.  Column-integrated COD varies from $0.25 \pm 0.45$ in the dry season to $0.47 \pm 0.65$ in the wet season. The frequency of occurrence of opaque, thin and SVC column-integrated COD is 11.8 % (31.3 %), 23.9 % (37.9 %) and 23.3 % (18.3 %) respectively in the dry (wet) season. The maximum backscattering altitude does not show a seasonal cycle, and is on average $13.4 \pm 2.0$ km (or $-60 \pm 15$ °C). The average number of simultaneous layers of cirrus present in each cloudy profile is 1.4 (1.25 during the dry, and 1.62 during the wet season), and hence geometrical properties, in a column-integrated sense, are not discussed.
As cirrus at different altitudes might have different origins or microphysical properties, it is more important to analyze the statistics based on each layer detected, as shown in Table 3. The overall mean value for the cloud layer base altitude is $12.9 \pm 2.2$ km, for the cloud layer top altitude, $14.3 \pm 1.9$ km, and for the cloud layer geometrical thickness, $1.4 \pm 1.1$ km. The mean value of the cloud layer maximum backscattering altitude is $13.6 \pm 2.0$ km. (…)

**R:** L. 346 – What about TTL Cirrus?

From our observations alone we cannot rule out that layers observed above the tropopause are not TTL cirrus. These can either be formed "*from ice detrainment from convective towers or from in situ formation in supersaturated regions created by large- to mesoscale uplifts*" (e.g. Wang and Dessler, 2012). Garret et al. (2004) mention that these thin tropopause cirrus "*originate as stratiform pileus clouds that form near the tropopause ahead of vigorous convective uplift".* They then "*hypothesize that the pileus are penetrated by the convection, moistened through turbulent mixing, and once the convection subsides, they are sustained by radiative cooling due to the presence of the anvil layer beneath".* It should be noted, however, that until nowadays "*the respective importance of homogenous and heterogeneous nucleation remains unclear, although in situ observations suggest that both are active in the TTL" (Jensen et al., 2013; Cziczo et al., 2013).* Finally, it is still unknown the *"role of different waves with different scales on cirrus processes"* (Kim and Alexander, 2015).

Because of this difficulty, we have until now refrained from talking about TTL, although we suspect that the thin cirrus we observed above the tropopause base are exactly that. We have mentioned convection overshooting and detrainment, which we could observe (or infer), but not about the possible in-situ formation. We note, however, that the necessary large-scale uplifts in the Amazon are easily found around mesoscale convective systems (MCS), which are very common feature of the Amazon hydrological cycle. Figure below gives an example of thin TTL cirrus that appears to be over an anvil cloud:

We agree with the reviewer that mentioning TTL cirrus would be important. We did not include the picture, but we changed the text around line 346 as follows:

Moreover, while the distribution of opaque cirrus peaks at 12 km height in both seasons, thin cirrus and SVC shows a bimodal distribution only in the wet season, with the highest maxima above 14 km and 16 km respectively. This is presumably associated with the overshooting convection discussed above, which occurs mostly during the wet season (Liu and Zipser, 2005). Moreover, ice detrainment directly into the tropical tropopause layer (TTL) is one of the main mechanisms of TTL cirrus formation; the other is in-situ formation by supersaturation promoted by mesoscale uplift (Cziczo et al., 2013), which can occur above tropical convective systems (Garret et al., 2004), a very common feature of the Amazon hydrological cycle.

The conclusion was changed as well:

The geometrical, and optical, and mycrophysial characteristics of cirrus clouds measured in the present study were consistent with other reports from tropical regions. The mean values were 12.9 ± 2.2 km (base), 14.3 ± 1.9 km (top), 1.4 ± 1.1 km (thickness), and 0.25 ± 0.46 (optical depth). Cirrus clouds were found at temperatures down to −90 °C and maximum backscatter altitude was 13.6 ± 2.0. 6 % (16 %) of the observed cirrus had their base (top) above the tropopause level or in the tropical tropopause layer.

By simultaneously analyzing cloud altitude and COD, it was found that cirrus clouds observed during the dry season months are optically thinner and lower in altitude than those during the wet period. The vertical distribution of frequency of occurrence is mono-modal, and 13 % of the observed cirrus had top within the TTL. During the wet season months, there is a wider range of COD for a fixed altitude, and vice-versa, which is associated with the variability in the intensity of deep convection in Amazonia. The vertical distribution of the frequency of occurrence of the detected clouds shows a bimodal distribution for thin and SV cirrus, and 19 % of the observed cirrus had top within the TTL, which are likely associated to slow mesoscale uplifting or to the remnants of overshooting convection.

And the abstract:

(…) The mean values of cirrus cloud top and base heights, cloud thickness and cloud optical depth were 14.3 ± 1.9 (std) km, 12.9 ± 2.2 km, 1.4 ± 1.1 km, and 0.25 ± 0.46, respectively. Cirrus clouds were found at temperatures down to −90 °C. 6 % of the Cirrus clouds were above the base of the tropical tropopause layer. Frequently cirrus were observed within the TTL, which are likely associated to slow mesoscale uplifting or to the remnants of overshooting convection. The vertical (…)

These are the references we mentioned:

Cziczo, D. J., Froyd, K. D., Hoose, C., Jensen, E. J., Diao, M., Zondlo, M. A., Smith, J. B., Twohy, C. H., and Murphy, D. M.: Clarifying the dominant sources and mechanisms of cirrus cloud formation., Science, 340, 1320–1324, doi:10.1126/science.1234145, 2013.

Garrett, T. J., A. J. Heymsfield, M. J. McGill, B. A. Ridley, D. G. Baumgardner, T. P. Bui, and C. R. Webster (2004), Convective generation of cirrus near the tropopause, J. Geophys. Res., 109, D21203, doi:10.1029/2004JD004952.

Jensen, E. J., Diskin, G., Lawson, R. P., Lance, S., Bui, T. P., Hlavka, D., McGill, M., Pfister, L., Toon, O. B., and Gao, R.: Ice nucleation and dehydration in the Tropical Tropopause Layer, P. Natl. Acad. Sci. USA, 110, 2041–2046, doi:10.1073/pnas.1217104110, 2013.

Kim, J.-E. and Alexander, M. J.: Direct impacts of waves on tropical cold point tropopause temperature, Geophys. Res. Lett., 42, 1584–1592, doi:10.1002/2014GL062737, 2015.

[revised manuscript text omitted]